# Interpreting Emergent Planning in Model-Free Reinforcement Learning

**Thomas Bush[1], Stephen Chung[1†], Usman Anwar[1†], Adrià Garriga-Alonso[2], David Krueger[3]**
[1]University of Cambridge, [2]FAR AI, [3]Mila, University of Montreal
28tbush@gmail.com, {mhc48,ua237,dsk30}@cam.ac.uk, adria@far.ai
[†]Equal contribution.

## Abstract

We present the first mechanistic evidence that model-free reinforcement learning agents can learn to plan. This is achieved by applying a methodology based on concept-based interpretability to a model-free agent in Sokoban – a commonly used benchmark for studying planning. Specifically, we demonstrate that DRC, a generic model-free agent introduced by Guez et al. (2019), uses learned concept representations to internally formulate plans that both predict the long-term effects of actions on the environment and influence action selection. Our methodology involves: (1) probing for planning-relevant concepts, (2) investigating plan formation within the agent's representations, and (3) verifying that discovered plans (in the agent's representations) have a causal effect on the agent's behavior through interventions. We also show that the emergence of these plans coincides with the emergence of a planning-like property: the ability to benefit from additional test-time compute. Finally, we perform a qualitative analysis of the planning algorithm learned by the agent and discover a strong resemblance to parallelized bidirectional search. Our findings advance understanding of the internal mechanisms underlying planning behavior in agents, which is important given the recent trend of emergent planning and reasoning capabilities in LLMs through RL.

## 1 Introduction

In reinforcement learning (RL), decision-time planning – that is, the capacity of selecting immediate actions to perform by predicting and evaluating the consequences of future actions – is conventionally associated with agents that possess explicit world models, like MuZero (Schrittwieser et al., 2020). This naturally raises the question: can model-free reinforcement learning agents – that is, agents which lack explicit world models – also learn to perform decision-time planning?

In prior work, Guez et al. (2019) introduced Deep Repeated ConvLSTM (DRC) agents. Despite lacking an explicit world model, DRC agents behave like they perform decision-time planning. For example, they excel at strategic domains like Sokoban, and perform better if given extra test-time compute (Guez et al., 2019; Taufeeque et al., 2024). However, this only partially answers the above question as these behaviors may not be due to internal planning but, rather, other mechanisms that generate planning-like behavior in the environments studied. In this paper, we mechanistically analyze a Sokoban-playing DRC agent and show that it is indeed internally planning. In doing so, we provide the first non-behavioral evidence that model-free RL agents can learn to internally plan.

Using concept-based interpretability (Kim et al., 2018), we provide three types of convergent evidence showing that the DRC agent has learned, and is making use of, concepts that are instrumentally useful for planning. First, we use linear probes (Alain & Bengio, 2016) to show that the agent represents specific concepts that predict the long-term effects of its actions on the environment. Then, we demonstrate that these concept representations are associated with a learned planning process by analyzing how the agent uses them to iteratively construct 'plans' at test-time. Finally, we demonstrate that these concept representations causally influence the agent's behavior as would be expected if these representations were being used for planning

To summarize, this paper makes the following contributions:

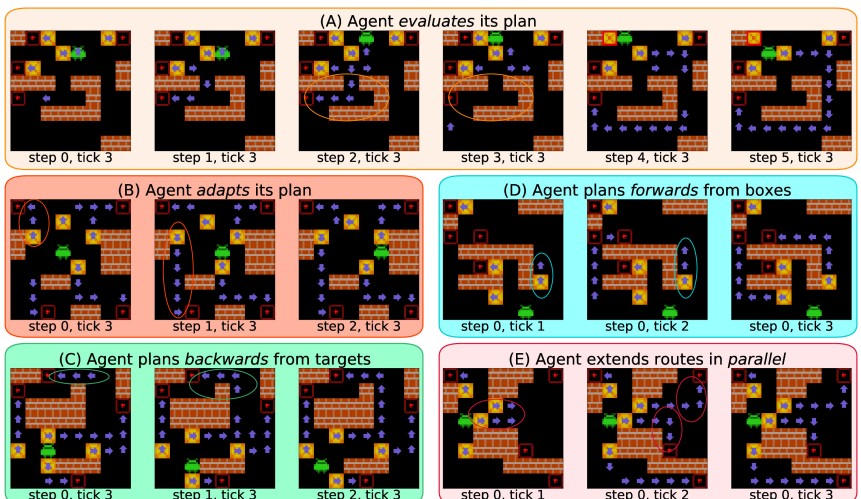

Figure 1: Examples of the DRC agent internally forming plans to push boxes to targets. A purple arrow on a square means that a linear probe decodes that the agent plans to push a box off of that square in the associated direction. No arrow on a square means that the probe decodes that agent does not plan to push a box off of that square. (A) The agent *evaluates* a naively-appealing route, concludes it is infeasible, and forms a longer alternate path. (B) The agent *adapts* its plan and changes the target it plans to push the left-most box to. (C) The agent extends part of its plan *backward* from a target. (D) The agent extends part of its plan *forward* from a box. (E) The agent extends many parts of its plan in *parallel*. We provide further examples in Appendices A.2.1-A.2.5.

- We design a procedure, based on concept-based interpretability, for determining if a model-free agent performs planning using a hypothesized set of concepts. This procedure involves (1) probing for planning-relevant concepts, (2) investigating plan formation in the agent's internal representations, and (3) verifying the causal effect of plans on the agent's behavior.

- Using this procedure, we show that, in Sokoban, a DRC agent (Guez et al., 2019) internally forms plans, and that these plans can be altered to steer the agent. We find this agent learns a planning algorithm resembling parallelized bidirectional search, which differs from commonly-used planning algorithms in RL.

This work aligns with the growing body of research demonstrating that model-free RL agents can learn to plan and even reason. For example, in this study, we show that DRC agents can learn to evaluate and revise plans. Recently, DeepSeek-R1, an LLM with reasoning capabilities primarily trained via RL, has demonstrated similar self-correction behavior in its reasoning, referred to as 'aha moments' (Guo et al., 2025). As such, we believe that understanding the mechanisms behind these emergent capabilities in RL agents is highly important.

## 2 BACKGROUND

### 2.1 PLANNING IN REINFORCEMENT LEARNING

*Planning* has many meanings in RL, encompassing algorithms utilizing environment models during training (Sutton, 1991) or at decision time (Silver et al., 2016; Chung et al., 2024a). In this work, we study whether an RL agent is specifically performing *decision-time* planning. Henceforth, we use 'planning' and 'decision-time planning' interchangeably. In past work, an agent is considered to be *planning* in this sense if it engages with an (explicit) world model to select actions associated with the best predicted long-term consequences (Hamrick et al., 2020; Chung et al., 2024a). An example is MuZero (Schrittwieser et al., 2020), which applies a planning algorithm called Monte Carlo Tree Search (Coulom, 2006) to a model of its environment to select actions associated with the best long-run consequences. Other similar agents are VPN (Oh et al., 2017), IBP (Pascanu et al., 2017), I2A (Racanière et al., 2017), MCTSNet (Guez et al., 2018b), and Thinker (Chung et al., 2024a) agents.

By definition, model-free RL agents lack an *explicit* world model. This makes it difficult to reuse past definitions of planning that presume that an explicit world model is available. Thus, for the purposes of this work, we provide a pragmatic characterization of planning that we use as a foundation for investigating whether the model-free agent studied in this paper performs planning.

We consider plans to be sequences of potential future actions. We characterize an agent as planning *if it selects actions to perform by considering plans that it formulates and evaluates based on predicted future consequences*. This is similar to how planning is understood in neuroscience (Mattar & Lengyel, 2022). It also mirrors model-based definitions of planning but relaxes the requirement for an explicit world model to the requirement that an agent predict consequences of future actions, regardless of the method used. We discuss our characterization further in Appendix E.1. For an agent to plan under our characterization, it must: (i) form plans, (ii) evaluate plans by predicting their consequences, and (iii) be influenced by these plans when acting.

## 2.2 SOKOBAN

Sokoban is an episodic, fully-observable, deterministic environment in which an agent moves around walls in an 8x8 grid to push four boxes onto four targets. When an agent moves up/down/left/right into a square containing a box, the box is pushed up/down/left/right. Sokoban levels let agents perform actions with irreversible, negative, long-run consequences (moving boxes so the puzzle is unsolvable). Sokoban is thus difficult – it is PSPACE-complete (Culberson, 1997) – and a common benchmark for studying planning (Racanière et al., 2017; Guez et al., 2019; Hamrick et al., 2020). We study a version of Sokoban where the agent observes a symbolic representation $x_t \in \mathbb{R}^{8 \times 8 \times 7}$ of the environment. For ease of inspection, all figures are presented as pixel representa-

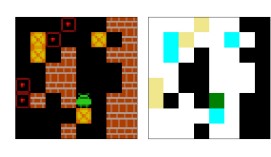

(a) Pixel    (b) Symbolic

Figure 2: Pixel and symbolic representations of a Sokoban board.

tions. Figure 2 compares these two representations. Appendix E.2 further explains this environment.

## 2.3 DEEP REPEATED CONVLSTM (DRC) AGENTS

Deep Repeated ConvLSTM (DRC) agents (Guez et al., 2019) are model-free agents based on ConvLSTMs that perform multiple computational ticks per time step. ConvLSTMs (Shi et al., 2015) are LSTMs (Hochreiter & Schmidhuber, 1997) that utilize 3D hidden states and convolutional connections. At each time step $t$, a DRC agent passes an observation $x_t$ through a convolutional encoder to generate an encoding $i_t \in \mathbb{R}^{H_0 \times W_0 \times G_0}$. This is then processed by $D$ ConvLSTM layers. At time $t$, the $d$-th ConvLSTM has a cell state $g_t^d \in \mathbb{R}^{H_d \times W_d \times G_d}$. Unlike standard recurrent networks which perform a single tick of recurrent computation per time step, DRC agents perform $N$ ticks of recurrent computation per step. Guez et al. (2019) show these internal ticks improve the performance and generalization of DRC agents. Appendix E.3 provides further architectural details.

DRC agents behave in a manner that suggests they internally engage in decision-time planning. For instance, DRC agents outperform model-based agents like MuZero (Schrittwieser et al., 2020) in Sokoban (Chung et al., 2024b), and exhibit improved performance when given extra test-time compute (Taufeeque et al., 2024). This raises a question: do DRC agents genuinely learn to internally perform planning, or is their planning-like behavior merely a result of complex learned heuristics?

In this paper, we investigate whether a Sokoban-playing DRC agent internally plans. The agent we study has $D = 3$ ConvLSTM layers and performs $N = 3$ internal ticks per step. The agent's encoder and ConvLSTMs have 32 channels ($G_d = 32$) and utilize kernels of size 3 with a single layer of input zero padding. Thus, all cell states share Sokoban's spatial dimensions ($H_d = W_d = 8$). The agent is trained for 250 million transitions on the unfiltered Boxoban training set (Guez et al., 2018a) using a similar training setup as Guez et al. (2019) as explained in Appendix E.4. Appendix E.5 shows that, consistent with Guez et al. (2019), this agent exhibits planning-like behavior.

## 2.4 CONCEPT-BASED INTERPRETABILITY

Concept-based interpretability is an approach to explaining neural network behavior that involves identifying which concepts a network internally represents (Kim et al., 2018). A concept is generally

understood as a unit of knowledge (Schut et al., 2023). In this paper, we specifically consider 'multi-class' concepts, which can formally be defined as mappings from input states (or parts of input states) to some fixed classes. That is, multi-class concepts correspond to interpretable, discrete features, and map inputs to classes of that concept. For instance, a multi-class Sokoban concept might be 'the number of empty targets'. This concept would map any observed Sokoban board $x_t$ to a class in $\{\texttt{ONE}, \texttt{TWO}, \texttt{THREE}, \texttt{FOUR}\}$ depending on the number of remaining empty targets in $x_t$.

We focus on concepts networks represent *linearly* (Mikolov et al., 2013). To check if a network linearly represents concepts, we use *linear probes*. These are linear classifiers trained to predict concept classes assigned to inputs using the associated network activations (Alain & Bengio, 2016). As linear classifiers, linear probes compute logits $l_k = w_k^T g$ for each class $k$ by projecting network activations $g \in \mathbb{R}^d$ along a class-specific vector $w_k \in \mathbb{R}^d$. Belinkov (2022) explains probes further.

# 3 METHODOLOGY

## 3.1 A PROCEDURE FOR INVESTIGATING MODEL-FREE PLANNING

In Section 2.1, we characterized planning as requiring that an agent (i) formulate plans, (ii) evaluate the consequences of these plans, and (iii) be guided by these plans when selecting actions. If an agent learns to plan, we expect planning-relevant concepts to emerge in its internal representations to meet the first condition. These concepts ought to reflect the agent's plan, and so should correspond to potential future actions, or to their likely environmental effects. Additionally, evidence of plan evaluation – such as avoiding or improving bad plans – should exist to satisfy the second condition. Lastly, to fulfill the third condition, the plan must causally influence the agent's behavior. To determine if an agent exhibits these three properties, we follow the procedure outlined below:

1. **Probe for Concept Representations.** First, we identify a group of environment-specific concepts that could be instrumentally useful for planning. We then use linear probes to establish whether these concepts are being (linearly) represented by the agent (Section 4).

2. **Investigate Plan Formation.** Next, we focus on gathering qualitative evidence of the agent forming plans based on the planning-relevant concepts probed for in the previous step, and evidence of the agent evaluating and refining these plans (Section 5).

3. **Confirm Behavioral Dependence.** Finally, we confirm that these internal plans influence the agent's behavior. For instance, we show that the agent can be steered to form and execute desired plans by intervening on plan representations within the network (Section 6).

## 3.2 PLANNING-RELEVANT CONCEPTS IN SOKOBAN

To apply this procedure, we must specify concepts we expect the agent to plan with. Sokoban has a grid-based structure with localized transition dynamics, i.e., the future state of a square is determined by the current state of its neighbors. This makes spatially local concepts (i.e., concepts related to individual or connected squares) more natural for planning than spatially global concepts (i.e., representations of the whole board). We thus claim that an agent that learns to plan in Sokoban may do so by encoding concepts localized to individual squares. We call these 'square-level' concepts. Such concepts seem natural for DRC agents as the 3D structure of ConvLSTMs allows for spatial correspondence between the Sokoban grid and agent hidden states. We focus on multi-class square-level concepts which, as explained further in Appendix E.6, map grid squares to concept classes.

We hypothesize that the agent will plan using the following square-level, multi-class concepts:

- **Agent Approach Direction** ($C_A$): For a given square, this concept encodes whether the agent will move onto the square in the future. If so, it also encodes the direction from which the agent will move onto the square the next time the agent moves onto it.

- **Box Push Direction** ($C_B$): For a given square, this concept encodes whether a box will be pushed off the square in the future. If so, it also encodes the direction in which the next box pushed off this square will be pushed.

Figure 3 illustrates the classes assigned to each square of a Sokoban board by these concepts over six transitions near the end of an episode. Both concepts map each grid square of the agent's observed

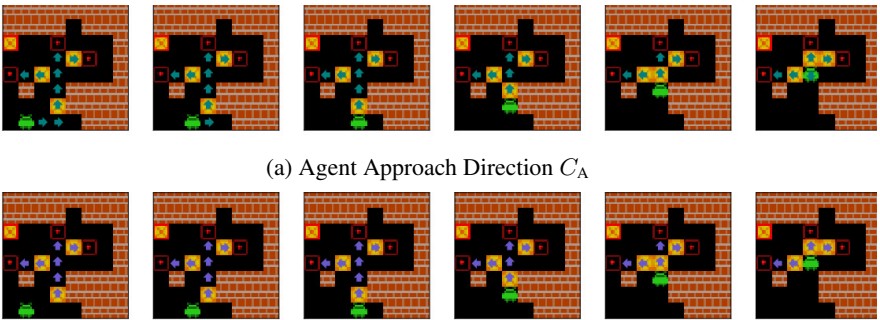

(a) Agent Approach Direction $C_A$

(b) Box Push Direction $C_B$

Figure 3: Examples of the classes assigned to the squares of a Sokoban board over 6 transitions (from left to right) by the concepts 'Agent Approach Direction' ($C_A$) and 'Box Push Direction' ($C_B$). An arrow corresponds to the assignment of the associated directional class. The lack of an arrow in a square indicates the assignment of the class NEVER.

Sokoban board to the classes {UP, DOWN, LEFT, RIGHT, NEVER}. The directional classes correspond to the agent's movement directions. If the next time the agent *steps onto a specific square*, the agent steps onto that square from the left, the concept $C_A$ would map this square to the class LEFT. If the next time the agent *pushes a box off of specific square*, the box is pushed to the left, the concept $C_B$ would map this square to the class LEFT. Finally, the class NEVER corresponds to the agent not stepping onto or pushing a box off of a square again for the remainder of the episode.

Both concepts depend on the agent's behavior: we can only determine the classes these concepts map grid squares to *after* observing the agent's behavior over the entire episode. Furthermore, as shown in Figure 3, the classes squares are mapped to will change at every transition. Once an agent steps onto a square, the classes assigned to that square will update to represent the agent's *future* interactions with that square. We investigate alternate concepts in Appendices D.4 and D.5.

## 4 PROBING FOR CONCEPT REPRESENTATIONS

We now perform the first step of our analysis: determining whether the agent internally represents the concepts that we hypothesize it uses to internally form and evaluate plans.

### 4.1 EXPERIMENT DETAILS

Specifically, we use linear probes to determine if the agent represents (a) $C_A$, the agent's future movement onto squares, and (b) $C_B$, the future directions boxes are pushed off of squares. We train linear probes that take as input the agent's cell state activations after the final of the three computational ticks performed each step. We train separate probes for the agent's three layers.

We hypothesize the agent will learn a spatial bijection between its cell state and the Sokoban grid. Thus, when predicting $C_A$ and $C_B$ at each location $(x, y)$, our probes receive as input cell state activations centered on $(x, y)$. We train both 1x1 probes (which take as input just the activations at $(x, y)$) and 3x3 probes (which take as input the 3x3 patch of activations around $(x, y)$). These probes have 160 and 1440 parameters, so are unlikely to overfit. We consider larger probes in Appendix D.3.

Each probe is trained using logistic regression with the AdamW optimizer, and five unique initialization seeds. The training dataset is generated by running the agent for 3000 episodes on levels from the Boxoban unfiltered training dataset (Guez et al., 2018a). We test probes on a test set of transitions generated by running the agent for 1000 episodes on levels from the Boxoban unfiltered validation dataset. Further probe training details are given in Appendix D.1. We compare the performance of all probes to baseline probes that receive the raw observation $x_t$ as input. This comparison aims to assess the extent to which probes' abilities to predict concept classes are due to these concepts being internally represented by the agent rather than the probes learning how to do so themselves.

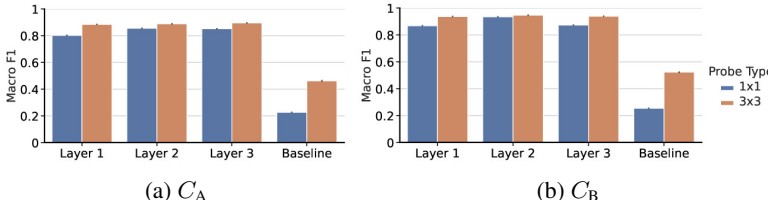

(a) $C_A$                  (b) $C_B$

Figure 4: Macro F1s achieved by probes when predicting $C_A$ and $C_B$ using the cell state at each layer, or, for the baseline probes, using the observation. Error bars show $\pm 1$ standard deviation.

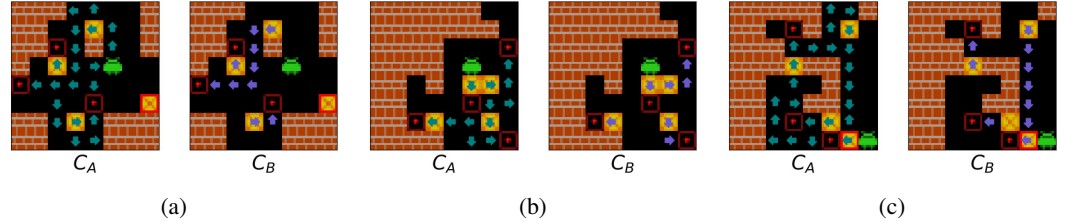

(a)              (b)            (c)

Figure 5: Examples of internal plans computed by the agent. An internal plan corresponds to the agent's combined square-level representations of $C_A$ and $C_B$. That is, an internal plan corresponds to the classes the agent represents these concepts as mapping squares of observed boards to. These internal plans are decoded from the agent's final layer cell state by a 1x1 probe. Teal and purple arrows respectively indicate the agent expects to next step on to, or push a box off, a square in the associated direction. No arrow indicates the agent does not plan to step onto, or push a box off, a square again. Further examples of internal plans are given in Figures 10, 11 and 12 in Appendix A.1.

## 4.2 RESULTS

In many Sokoban boards, the agent will never move onto, nor push a box off, a large number of squares. As a result, many squares are assigned the label NEVER for both concepts in our probing datasets, leading to class imbalance. We therefore evaluate probe performance using macro F1 scores in place of accuracy. Figure 4 shows the macro F1 scores achieved by probes trained to predict the classes assigned to Sokoban squares by $C_A$ and $C_B$. The probes that predict these concepts using the agent's cell state activations vastly outperform the baseline, implying the agent linearly represents $C_A$ and $C_B$. This aligns with past work finding linear concept representations in many different networks (Nanda et al., 2023; McGrath et al., 2022; Zou et al., 2023).

Figure 4 confirms that the agent represents square-level concepts at localized positions of its ConvLSTM cells as opposed to distributing representations across adjacent positions. This is evidenced by the minimal improvement in performance when moving from a 1x1 probe to a 3x3 probe, compared to the significant improvement in baseline performance. We thus focus on 1x1 probes for the remainder of this paper. Interestingly, Figure 4 also shows that while probes at layer 2 generally perform slightly better than probes at layer 1, there is little variation in performance across layers. This indicates that the concepts are represented across all layers. We thus hypothesize that the agent is engaged in iterative computation (Jastrzebski et al., 2018), whereby it refines plans across layers.

## 5 INVESTIGATING PLAN FORMATION

In this section, we now provide qualitative evidence that the agent forms plans by searching forward from the boxes and backward from the targets, and that the agent develops, evaluates, and adapts plans in parallel. In this section, we primarily focus on descriptive explanations of how the agent forms plans and the general shape of the plans. We defer more conclusive evidence – in the form of intervening on the agent's plan formation process to steer the agent's behavior – to the next section.

Previously, we demonstrated that the agent encodes (at least) two planning-relevant concepts: $C_A$ and $C_B$. These concepts represent predictions regarding how the agent will act when moving onto a given square in the future, and how the environment – specifically, the locations of boxes – will

be affected by these actions. We thus posit that the agent's representations of these concepts – when looked at holistically, over the entire board – will collectively constitute a plan that the agent forms and adapts. For example, in Figure 5 we visualize the agent's representations of $C_B$ and $C_A$ over entire Sokoban boards, as decoded from the agent's cell state by a 1x1 probe in different levels. Three observations can be made from Figure 5: (a) the arrows, which indicate the direction the agent expects to move or push boxes, tend to be connected and trace a path; (b) the arrows tend to connect boxes to specific targets; (c) the arrows collectively form a plan which corresponds to solving the level. In Appendix A.1 we visualize the agent's plan across layers, and show that, while the agent's plans often contains flaws (like the lack of one necessary arrow in Figure 5c), they usually consist of connected paths for the agent to follow and connected routes linking boxes and targets.

A natural question then arises: how does the agent form plans? To answer this, we direct attention to Figure 1. Figure 1 visualizes the agent's plans in terms of $C_B$ (e.g. the routes the agent plans to push boxes) over the initial steps (A-C) and internal ticks (D-E) of episodes. As can be seen in Figure 1, the agent forms plans *iteratively*. Interestingly, the agent appears to form plans iteratively by searching *forward* from boxes – as illustrated in Figure 1(C) – and *backward* from targets – as illustrated in Figure 1(D). That the agent seems to plan via bidirectional search – which is known to be especially efficient when it is applicable (Russell & Norvig, 2010) – may explain why Guez et al. (2019) found DRC agents to rival specialized planning architectures reliant on forward search. Indeed, as shown in Figure 1(E), the agent seems to utilize a form of *parallelized* bidirectional search whereby it extends multiple plans simultaneously. Appendices A.2.3, A.2.4 and A.2.5 respectively contain further instances of the agent appearing to utilize forward, backward, and parallel search.

However, recall that, in Section 2.1, we characterized planning as requiring an agent to evaluate the plans it considers. Evidence suggestive of the agent evaluating plans can be seen in Figure 1(A)-(B). Figures 1(A)-(B), show examples in which the agent appears to (1) formulate a naive plan, (2) evaluate it, and then, upon realizing that it is infeasible or could be improved, (3) adapt its plan accordingly. For instance, in Figure 1(B), the agent changes the targets it plans to push different boxes towards. This is suggestive of the agent using an *evaluative* search algorithm when forming plans. Appendices A.2.1 and A.2.2 contain further examples of the agent seeming to evaluate plans and either plan to push a box a longer route, or change which boxes it plans to push to which targets.

Further evidence of the agent planning via an iterative search algorithm can be seen in Figure 6. For Figure 6, we forced the agent to remain stationary for 5 steps prior to acting in 1000 episodes. These 5 'thinking steps' give the agent 15 internal ticks of extra test-time compute. Figure 6 reports the macro F1 when using 1x1 probes to decode $C_A$ and $C_B$ from the agent's final layer cell state at each of the 15 extra internal ticks, averaged over 1000 episodes. Clearly, the macro F1 improves with the number of ticks. Since the concepts are predictions of future behavior, we can see the predictions of our probes at any tick as being the agent's internal plan *at that tick*. We can then see the corresponding macro F1 as reflecting the quality of the agent's plan at that tick. Figure 6 shows that, as would be expected if the agent planned via an iterative search, the agent's plans iteratively improve when given extra compute. Appendix A.3.1 shows test-time plan refinement oc-

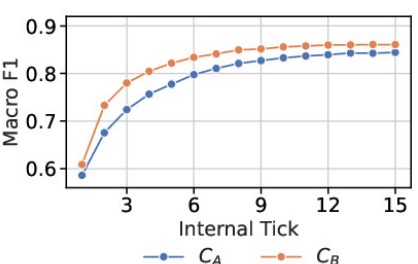

Figure 6: Macro F1 when using 1x1 probes to decode $C_A$ and $C_B$ from the agent's final layer cell state at each of the additional 15 internal ticks performed by the agent when the agent is given 5 'thinking steps', averaged over 1000 episodes.

curs at all layers. Appendix A.3.2 provides evidence that it is a consequence of the agent searching deeper. Appendix C.2 shows that this 'test-time plan refinement capability' arises early in training.

When considered with the agent's planning-like behavior, the above evidence indicates the agent uses its representations of $C_A$ and $C_B$ for search-based planning. Further evidence of this is given in Appendices A.2.6-A.2.9 which show examples of the agent planning in out-of-distribution levels, such as levels in which the agent itself is not present (Appendix A.2.6), levels with additional boxes and targets (Appendix A.2.7), and levels in which walls appear and disappear (Appendices A.2.8-A.2.9). These examples suggest the agent's ability to adapt and generalize – benefits of model-based planning Guez et al. (2019) show DRC agents possess – relate to its representations of $C_A$ and $C_B$.

|    | Layer 1 | | Layer 2 | | Layer 3 | |
| --- | --- | --- | --- | --- | --- | --- |
|    | Trained (%) | Random (%) | Trained (%) | Random (%) | Trained (%) | Random (%) |
| AS | 94.6 ($\pm$0.5) | 33.7 ($\pm$32.7) | 90.1 ($\pm$1.9) | 29.8 ($\pm$36.8) | 98.8 ($\pm$0.0) | 27.8 ($\pm$37.9) |
| BS | 56.2 ($\pm$1.4) | 31.5 ($\pm$13.9) | 72.7 ($\pm$1.1) | 30.9 ($\pm$25.8) | 80.6 ($\pm$2.4) | 4.1 ($\pm$5.4) |

Table 1: Success rates (%) when intervening on each layer using representations from trained and randomly initialized probes. AS and BS refer to 'Agent-Shortcut' and 'Box-Shortcut' interventions. Success rates are averaged over 5 interventions performed. We report $\pm 1$ standard deviations.

## 6 INVESTIGATING THE ROLE OF PLANS

So far, we have shown that the DRC agent represents $C_A$ and $C_B$ (Section 4), and that it uses these representations to form internal plans (Section 5). We now conclude our analysis by showing that these representations are causally responsible for the agent's behavior. Specifically, we: (1) use these representations to intervene on the agent to force it to form and execute specific plans, and (2) show that these representations emerge concurrently with planning-like behavior during training.

### 6.1 INTERVENING ON AGENT PLANS

First, we show we can intervene on the agent's activations to alter its behavior over entire episodes. Our interventions involve adding concept vectors learned by probes to the agent's activations to force it to represent concepts in specific ways. We then observe the causal effect of our interventions on the agent's behavior. Recall that a 1x1 probe projects activations along a vector $w_k \in \mathbb{R}^{32}$ to compute a logit for class $k$ of some multi-class concept $C$. We thus encourage the agent to represent square $(x, y)$ as class $k$ for concept $C$ by adding $w_k$ to position $(x, y)$ of the agent's cell state $g_{x,y}$:

$$g'_{x,y} \leftarrow g_{x,y} + w_k \tag{1}$$

If the agent indeed uses $C_A$ and $C_B$ for planning, altering the agent's square-level representations of these concepts ought to modify its internal plan and, subsequently, its long-term behavior.

We intervene in two sets of handcrafted levels: 'Agent-Shortcut' and 'Box-Shortcut' levels. These sets of levels are characterized by, in each level, there existing two plans: a short plan and a long plan. The plans are similar, but differ in lengths. The agent by default follows the optimal (short) plan. We show our interventions cause it to instead form and execute the suboptimal (long) plan.

In 'Agent-Shortcut' levels all boxes and targets are in one region of the board, and the agent can follow either a long or short path to this region. In these levels, we intervene using vectors learned by probes trained to predict $C_A$ to steer the agent to plan to move along the long path. Our intervention consists of two parts. We add the vector for NEVER to cell state positions on the short path. We call this the 'short-route' intervention. We also add the vector for the direction which would lead the agent to move onto the first square of the long path to the appropriate cell state position. We call this the 'directional' intervention. An Agent-Shortcut intervention is illustrated in Figure 7b.

'Box-Shortcut' levels are specially-designed levels in which three boxes are adjacent to targets and a fourth box is not. The final box can be pushed a long or short route to a target. In these levels, we intervene using vectors learned by probes trained to predict $C_B$ to steer the agent to push this box the long route. Our intervention again consists of two parts. We add the vector for NEVER to cell positions on the short route We also add the directional representation which would encourage the agent to push the box the longer route to the box's initial position. We again call these the 'short-route' and 'directional' interventions. A Box-Shortcut intervention is illustrated in Figure 8b.

We intervene on 200 levels of each type. We created 25 levels of each type and then generated 8 versions of each level by applying vertical reflection and 90°, 180°, and 270° rotations. In all levels, we repeat the 'short-route' intervention every step but repeat the 'directional' intervention only until the agent moves onto, or pushes the box off, the corresponding square.

We perform our interventions on the agent's cell state at each layer. An intervention is considered successful if it causes the agent to solve the level in the desired suboptimal way. As a baseline, we intervene using representations from randomly initialized probes. For comparability, we scale random

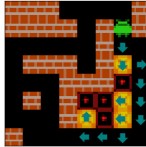 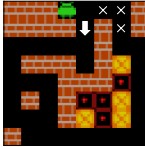 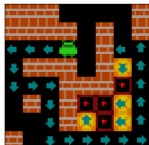

(a) Plan without intervention       (b) Intervention       (c) Plan with intervention

Figure 7: An Agent-Shortcut intervention and its effect on the agent's plan as formulated in terms of $C_A$: (a) the agent's plan after 4 steps *without* the intervention, (b) the initial state of the level and the intervention, and (c) the agent's plan after 4 steps *with* the intervention. The 'short-route' intervention adds the representation of NEVER for $C_A$ to positions with white crosses. The 'directional' intervention adds the representation of DOWN for $C_A$ to the position with the white arrow.

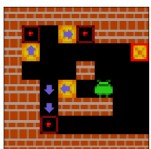 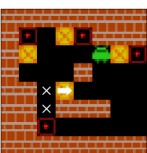 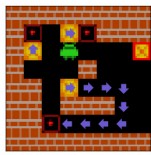

(a) Plan without intervention       (b) Intervention       (c) Plan with intervention

Figure 8: A Box-Shortcut intervention and its effect on the agent's plan as formulated in terms of $C_B$: (a) the agent's plan after 4 steps *without* the intervention, (b) the initial state of the level and the intervention, and (c) the agent's plan after 4 steps *with* the intervention. The 'short-route' intervention adds the representation of NEVER for $C_B$ to positions with white crosses. The 'directional' intervention adds the representation of RIGHT for $C_B$ to the position with the white arrow.

probe representations so that the norms of both the random and trained probes are similar. Success rates are averaged over interventions performed with five independently trained or initialized probes.

Table 1 shows intervention success rates. At all layers, Agent-Shortcut interventions are successful. While the success rate of Box-Shortcut interventions is lower, it remains high relative to the baseline of interventions using random probes. These results indicate that the agent's representations of $C_A$ and $C_B$ influence its behavior in the way that would be expected if it used them for planning. Figures 7 and 8 provide examples of the effect of interventions on the agent's internal plans. These examples suggest the agent not only behaves differently following the interventions, but does so *due to forming a different plan*. We show more examples of interventions altering the agent's internal plans in Appendix B.1. Appendix B.2 reports success rates when using an intervention scaling factor and varying the squares intervened on. Appendix B.3 reports success rates when intervening to encourage optimal behavior in levels which the agent by default cannot solve. These extra experiments further indicate that the agent's representations of $C_A$ and $C_B$ influence its behavior as expected.

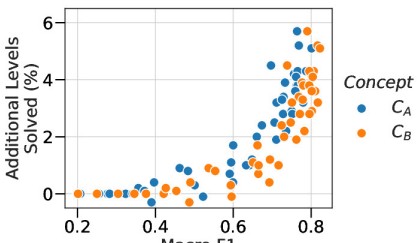

Figure 9: The relationship between the percentage of extra levels, of medium difficulty, solved when an agent is given 5 steps to 'think', and macro F1 score of probes when predicting $C_A$ (blue) and $C_B$ (orange) from the agent's final layer cell state. Each point corresponds to these quantities calculated for a single checkpoint.

## 6.2 INVESTIGATING THE EMERGENCE OF PLANNING DURING TRAINING

Finally, we show that the emergence of the agent's representations of $C_A$ and $C_B$ during training coincides with the agent beginning to exhibit planning-like behavior. This indicates that the agent indeed uses its representations of $C_A$ and $C_B$ for planning. Specifically, we show the emergence of these representations coincides with the emergence of the agent's ability to benefit from extra test-

time compute (Guez et al., 2019; Taufeeque et al., 2024). In particular, we collect checkpoints every 1 million transitions for the first 50 million transitions of training. For every checkpoint, we measure two quantities: (i) the macro F1 score of 1x1 probes trained to decode the concepts $C_A$ and $C_B$ given the agent's cell state (following the procedure described in Section 4.1), and (ii) the number of additional levels out of 1000 medium-difficulty levels from the Boxoban dataset (Guez et al., 2018a) the agent can solve when given extra test-time compute by forcing the agent to remain stationary for the first 5 steps of an episode. Figure 9 plots these quantities against each other and shows a strong correlation between them. This implies the agent only reliably begins to exhibit planning-like behavior – benefiting from extra test-time compute – once its final layer representations of $C_A$ and $C_B$ are sufficiently formed. Appendix C.3 shows that this holds for its representations of $C_A$ and $C_B$ at all layers. Appendix C.4 shows the agent begins to perform better with extra compute at a similar point in training as to when it can use this compute to refine its plans.

## 7 ADDITIONAL RESULTS

In the Appendix, we include interesting results that we lacked space to include in the main text. Appendices F provides evidence of DRC agents planning both without internal ticks, and with additional internal ticks. Appendix H provides evidence of a DRC agents planning in a different environment: Mini PacMan. Finally, Appendix G provides evidence of a ResNet (He et al., 2016) agent planning in Sokoban. However, the question of whether a generic agent can learn to plan in a generic environment remains unanswered.

## 8 RELATED WORK

Past work has investigated concept representations learned by game-playing agents (Schut et al., 2023; McGrath et al., 2022; Hammersborg & Strümke, 2022; 2023; Lovering et al., 2022; Mini et al., 2023) and language models (Li et al., 2023; Nanda et al., 2023; Karvonen, 2024; Ivanitskiy et al., 2024). While past work has focused primarily on whether networks internally represent specific concepts, we study concept representations for the broader purpose of determining if an agent possesses a capability - planning. An exception is work by Jenner et al. (2024), which finds evidence of look-ahead in a chess-playing agent, but does not investigate a wider capacity to 'plan'.

Concept-based interpretability is not the only approach to interpreting agents. An alternative is attribution-based interpretability. This involves determining – usually via saliency maps – which features in an agent's observation influence its behavior (Weitkamp et al., 2019; Iyer et al., 2018; Puri et al., 2020; Greydanus et al., 2018; Hilton et al., 2020). Attribution-based methods were not used here as they can depend on subjective interpretation (Atrey et al., 2020). Another approach, example-based interpretability, explains agent behavior by providing examples of illustrative trajectories or transitions (Rupprecht et al., 2020; Sequeira & Gervasio, 2020; Deshmukh et al., 2023; Zahavy et al., 2016). Due to not studying model internals, example-based methods were ill-suited for this paper.

Finally, this paper contributes to recent work investigating the emergence of reasoning capabilities in neural networks (Wei et al., 2022; Kojima et al., 2022; Lehnert et al., 2024; Nye et al., 2021; Wang et al., 2024). However, unlike this paper in which we provide evidence of an agent *internally* performing planning, most work thus far has focused on providing *behavioral* evidence of reasoning. An exception to this is work by Brinkmann et al. (2024) in which an algorithm learned by a transformer trained on a simple symbolic reasoning task is reverse-engineered. However, Brinkmann et al. (2024) focus on a much simpler form of reasoning than planning as considered in this paper.

## 9 FUTURE WORK

In this paper, we proposed a methodology for investigating model-free planning and used it to provide the first non-behavioral evidence of learned planning in a model-free agent. Future work may extend our investigation to other RL agents, and other environments. In particular, it would be helpful to better understand the role of different training factors, e.g., model architecture, environment dynamics in the emergence of planning.

ACKNOWLEDGMENTS

We are thankful to Erik Jenner and Joschka Braun for providing thoughtful feedback on the draft. For much of the duration of this work, TB was supported by the Cambridge Trust and Good Ventures Foundation. UA was supported by OpenPhil AI Fellowship and Vitalik Buterin Fellowship in AI Existential Safety. This work was performed using resources provided by the Cambridge Service for Data Driven Discovery (CSD3) operated by the University of Cambridge Research Computing Service (www.csd3.cam.ac.uk), provided by Dell EMC and Intel using Tier-2 funding from the Engineering and Physical Sciences Research Council (capital grant EP/T022159/1), and DiRAC funding from the Science and Technology Facilities Council (www.dirac.ac.uk).

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

# Appendix

## Table of Contents

# A   ADDITIONAL INVESTIGATIONS OF INTERNAL PLANNING

In Section 5, we provide evidence suggestive of the agent possessing a search-based internal planning mechanism. In this section, we now provide further complementary evidence regarding the agent's internal planning procedure. This section proceeds as follows:

- Appendix A.1 provides further examples of the agent's internal plan at all layers.
- Appendix A.2 provides additional examples of the agent forming plans in a manner suggestive of a search-based planning algorithm.
- Appendix A.3 provides additional investigations of the agent's ability to use extra test-time compute to improve its plans.

## A.1   FURTHER EXAMPLES OF INTERNAL PLANS

In Figure 5 we provided examples of 'internal plans' formulated by the agent. We understood the agent's internal plans to consist of its internal representations, for each square of its observed Sokoban board, of $C_A$ and $C_B$. In Figure 5, all internal plans were decoded from the agent's final layer cell state. In this section we now provide additional examples of internal plans formulated by the agent as decoded from its cell state at each layer.

Figures 10, 11 and 12 show internal plans decoded from the agent's first, second, and third layer cell states in many different levels. Specifically, Figure 10 shows the agent's internal plans at each layer at six transitions where the agent's internal plan *as decoded from its first-layer cell state corresponds to a complete plan to solve the respective level*. Similarly, Figures 11 and 12 show the agent's internal plan at each layer at six transitions where the agent's internal plan as respectively decoded from its second- and third-layer cell state correspond to a complete plan to solve the respective levels. We note that the observations we made regarding Figure 5 likewise hold here. That is, (1) the arrows tend to form connected paths, (2) the agent's plans tend to connect specific boxes to specific targets, and (3) the agent often forms complete plans to solve levels very early on in episodes.

Note, however, that the agent's plans in Figures 10, 11 and 12 often contain mistakes. This is despite the illustrated transitions being selected such that the agent's plan is correct in at least one layer. A few things can be noted about these mistakes. First, the agent's plans for box movements contain, on average, far fewer mistakes than the agent's plans for its own movements. Second, the mistakes in the agent's plan for its own movements are usually minor and consist of e.g. a few arrows being wrong, but the overall 'shape' of the plan being correct. Third, the agent's mistakes when planning its own movements in the examples tend to be mistakes regarding how it can move when not pushing boxes. We think these observations suggest that the agent is primarily planning by constructing plans in terms of $C_B$ connecting boxes and targets, and then augmenting these plans with planned agent movements where needed.

At a high-level, we suspect that mistakes in the agent's plan are best seen as relating to intermediate steps of the agent's internal planning process. First, this is because many mistakes seems to be plans that the agent considers on its way to arriving at its final plan. This is because mistakes are almost always fixed in future transitions. Second, some mistakes seem to be temporarily added to the agent's otherwise-correct plan at specific layers. We believe these mistakes potentially relate to the fact that, as part of its planning process, the agent sometimes considers variations on its plan.

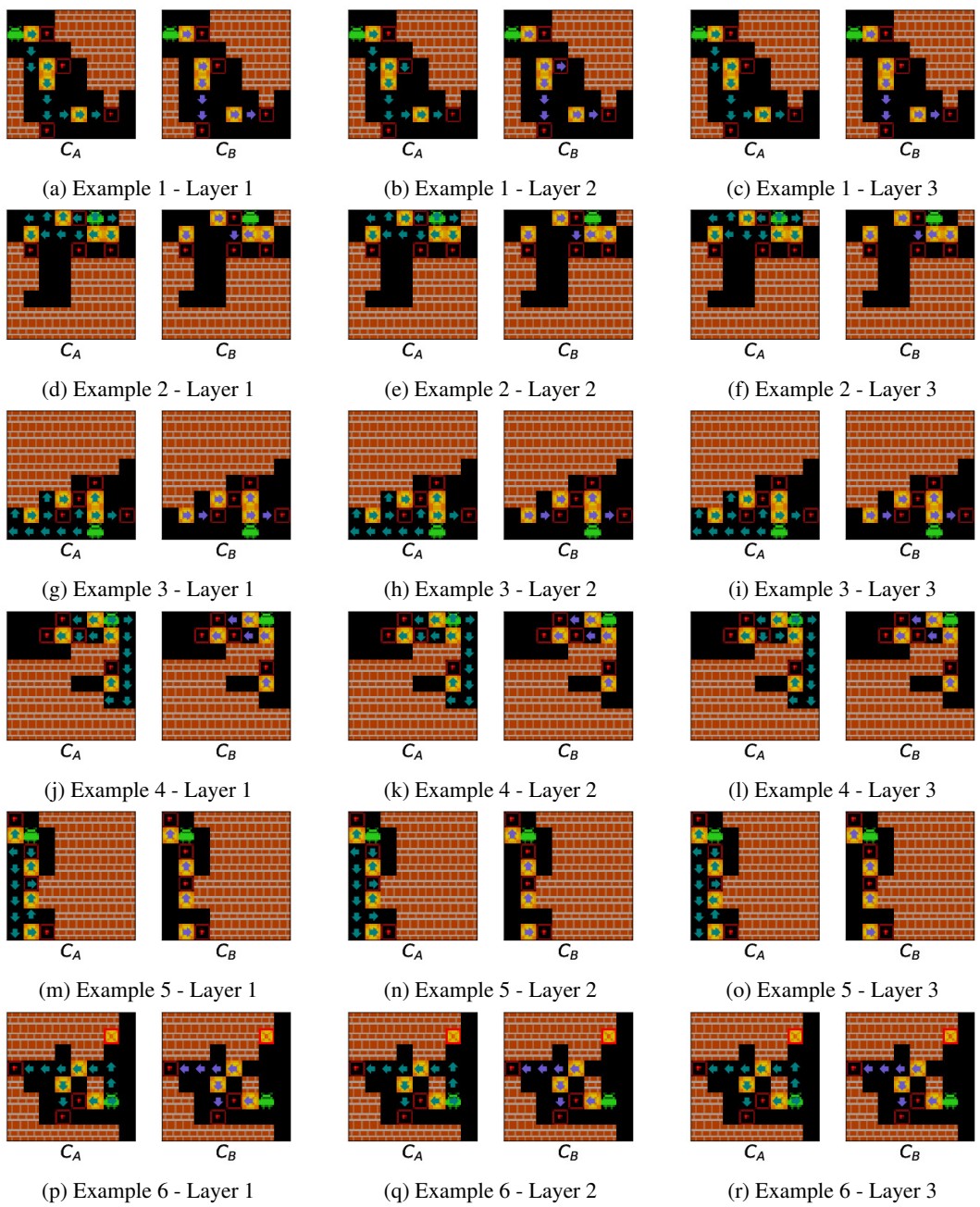

Figure 10: Internal plans computed by the agent in 6 examples levels as decoded from the agent's first layer cell state (10a, 10d, 10g, 10j, 10m, and 10p), second layer cell state (10b, 10e, 10h, 10k, 10n, and 10q) or third layer cell state (10c, 10f, 10i, 10l, 10o, and 10r) by a 1x1 probe. Teal arrows denote squares that the agent expects to next step onto from the associated direction. Blue arrows denote squares that the agent expects to push a box off in the corresponding direction. These examples are taken from the first transition of the respective episode in which the agent's plan as decoded from its **first-layer cell state** corresponds to a complete plan to solve the level.

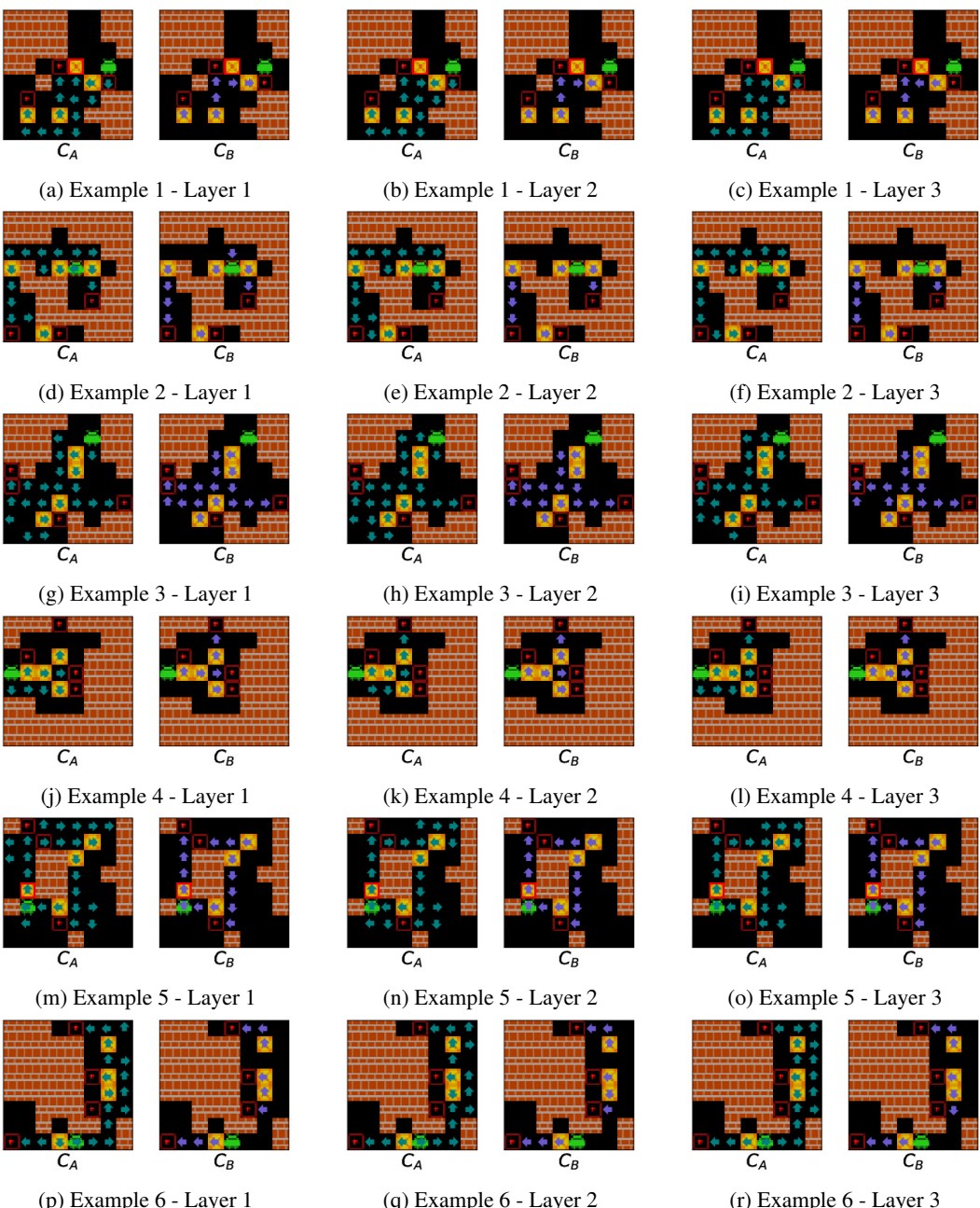

Figure 11: Internal plans computed by the agent in 6 examples levels as decoded from the agent's first layer cell state (11a, 11d, 11g, 11j, 11m, and 11p), second layer cell state (11b, 11e, 11h, 11k, 11n, and 11q) or third layer cell state (11c, 11f, 11i, 11l, 11o, and 11r) by a 1x1 probe. Teal arrows denote squares that the agent expects to next step onto from the associated direction. Blue arrows denote squares that the agent expects to push a box off in the corresponding direction. These examples are taken from the first transition of the respective episode in which the agent's plan as decoded from its **second-layer cell state** corresponds to a complete plan to solve the level.

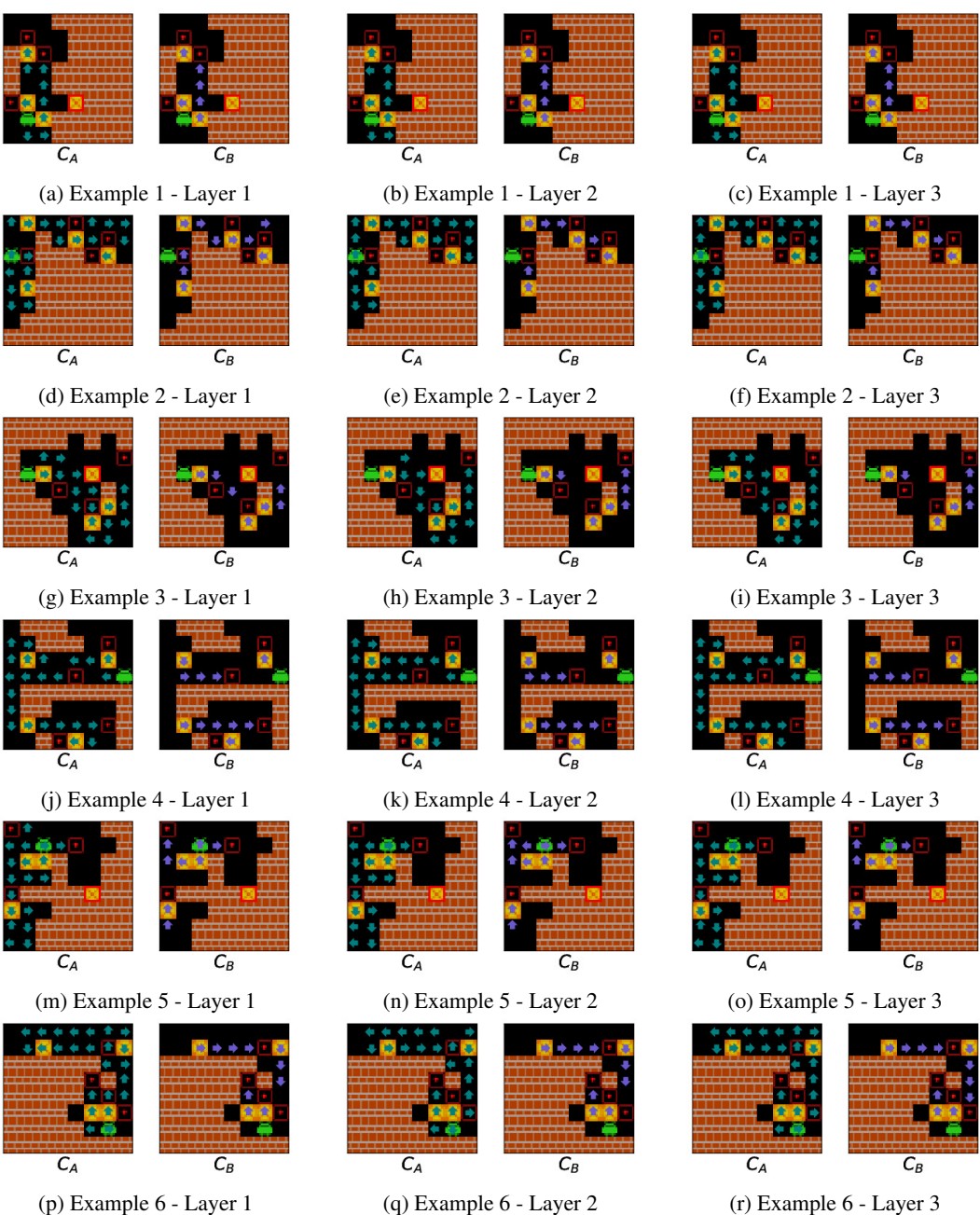

Figure 12: Internal plans computed by the agent in 6 examples levels as decoded from the agent's first layer cell state (12a, 12d, 12g, 12j, 12m, and 12p), second layer cell state (12b, 12e, 12h, 12k, 12n, and 12q) or third layer cell state (12c, 12f, 12i, 12l, 12o, and 12r) by a 1x1 probe. Teal arrows denote squares that the agent expects to next step onto from the associated direction. Blue arrows denote squares that the agent expects to push a box off in the corresponding direction. These examples are taken from the first transition of the respective episode in which the agent's plan as decoded from its **final-layer cell state** corresponds to a complete plan to solve the level.

## A.2 FURTHER EXAMPLES OF INTERNAL PLAN FORMATION

In this section, we provide additional examples of the agent seeming to use its implicit learned environment model to internally form plans via an evaluative search. As in the main paper, we focus on the agent's internal plan as formulated in terms of its representations of $C_B$. That is, we focus on the routes the agent expects to push boxes (e.g. 'box plans') rather than the paths the agent expects to follow (e.g. 'agent plans'). This is for two reasons. First, we expect 'box plans' to be determinative of 'agent plans' as the primary difficulty of Sokoban regards box movements. Second, 'box plans' are easier to visually inspect as 'agent plans' often intersect such that it is usually ambiguous to tell *why* a square has been added to an 'agent plan'. All examples use handcrafted Sokoban levels designed to allow for clear plan visualization.

We begin by providing additional examples of the plan formation 'motifs' shown in Figure 1. These motifs are re-occurring themes in the agent's internal plan formation process. In each of the following five sub-sections we present five examples of the agent formulating internal plans in a way that demonstrates one of the aforementioned motifs. These motifs are:

- Evaluative Planning - modifying plans based on feasibility (Section A.2.1)
- Adaptive Planning - modifying plans based on conflicts (Section A.2.2)
- Forward Planning - forming plans by searching forward from boxes (Section A.2.3)
- Backward Planning - forming plans by searching backward from targets (Section A.2.4)
- Parallel Planning - forming multiple plans in parallel (Section A.2.5)

However, these motifs are not the only evidence of evaluative, search-based planning exhibited by the agent's internal plan formation process. Specifically, evidence of this can also be seen when inspecting the manner in which the agent forms plans when confronted with various forms of out-of-distribution Sokoban levels. As such, we provide examples of the agent successfully formulating internal plans under various types of distribution shift. That is, we show examples of:

- Blind Planning - planning in levels in which the agent itself is not present (Section A.2.6)
- Generalized Planning - planning in levels with extra boxes and targets (Section A.2.7)
- Blocked-Route Planning - planning in levels in which additional walls appear at later time steps that block obvious routes to push boxes (Section A.2.8)
- New-Route Planning - planning in levels in which walls disappear at later time steps such that improved routes become available (Section A.2.9)

Finally, in Section A.2.10, we will conclude by discussing the implications of the agent's apparent learned search-based planning process.

### A.2.1 EVALUATIVE PLANNING

Under our characterization of 'planning' in Section 2.1, planning requires that an agent evaluates its plans. That is, merely formulating internal plans is not sufficient to view an agent as engaging in planning. Instead, we understand 'planning' as requiring that an agent arrive at an internal plan by means of evaluating different possible plans. This is because evaluating plans (e.g. in terms of their effects on the environment) allows the agent to formulate plans that it predicts will lead to good consequences.

When visualizing the agent's plan formation process, we see evidence indicative of the agent implementing some form of evaluative search-based planning. For instance, we see evidence of the agent iteratively constructing planned routes to push boxes to targets, then evaluating these routes. Specifically, the agent seems capable of evaluating routes and determining whether they are feasible. We call this *evaluative* planning.

Figure 13 shows the development of agent's internal plan in five episodes in which it performs evaluative planning of this sort. For instance, in Figure 13c, the agent's initial internal plan (i.e. at the first computational tick) involves the agent pushing the upper-left box down through a corridor to the center-most target. On the face of it, this plan is appealing. This is because it is a 'short' plan in terms of the number of squares it would involve pushing a box across. However, this plan

is infeasible. This is because the corridor is structured such that, upon pushing the box down into it, the agent would block the corridor off. This would then prevent the agent from (a) navigating to the left of the box as would be required to push it right along the corridor to the target, and (b) navigating to the lower-left box and target. Over subsequent ticks, the agent appears to evaluate its plan and realize this. In response, the agent then construct an alternative plan for the upper-left box. While less naively appealing – it is a longer plan in terms of the number of squares the box would need to be pushed – this plan would allow the agent to solve the level. The ability of the agent to recognize and avoid such 'bad' plans implies that the agent has learned to evaluate plans as required by our characterization of planning.

### A.2.2 ADAPTIVE PLANNING

Determining whether a single route would allow the agent to solve a level is not the only type of evaluation that the agent appears to perform when it formulates its internal plans. Additionally, the agent also appears to adapt its plans when conflicts arise between its sub-plans. That is, in cases when the agent's internal plan involves pushing two boxes to the same target, the agent adapts its plan by planning for one of these boxes to be pushed top an alternative target. This suggests that the agent is predicting and evaluating the consequences of its actions. We call this form of evaluation *adaptive* planning.

Instances of the agent performing adaptive planning when formulating its internal plans can be seen in Figure 14. Figure 14 shows the development of agent's internal plan over the initial steps of five episodes in which it performs adaptive planning. For examples, consider the way in which the agent's plans develop in Figure 14c. During the initial three computational ticks, the agent plans to push two separate boxes –i.e. the lower-left box and the upper-left box – to the left-most target. Importantly, however, in this level the upper-left box *must* be pushed to this target. This is because the left-most target is the only target that the upper-left box can feasibly be pushed to (e.g. if the lower-left box was pushed onto the left-most target, the level would be unsolvable). Over the fourth and fifth computational ticks, the agent appears to realize this and form an alternate plan for the lower-left target (e.g. a plan to push it to one of the top-right targets).

We take the ability of the agent to adaptively plan in this fashion as evidence that, during planning, the agent is capable of predicting and evaluating the (relevant) consequences of its actions. That is, the agent not only internally represents plans formulated in terms of the consequences of its actions on box locations (i.e. its internal plans formulated in terms of $C_B$), but also has learned that a consequence of following one of these plans and pushing a box onto a target is to 'fill' this target. This suggests that the agent has learned that the effect of pushing a box onto a target is to stop other boxes being able to be pushed onto that same target.

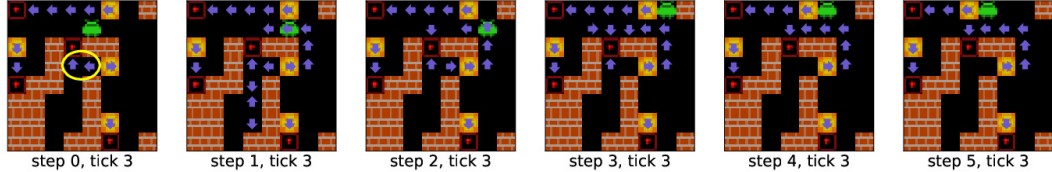

(a) The agent initially plans to push a box through the circled corridor. However, the agent appears to realize that this is infeasible as it cannot push the box up onto the target as required by this plan. It then forms a longer plan that involves pushing the box a longer route to the same target.

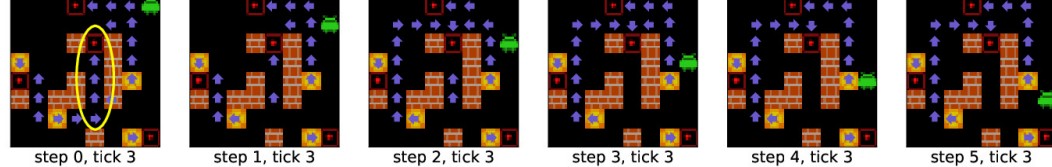

(b) The agent initially plans to push a box up through the circled corridor. However, over subsequent time steps, the agent appears to realize that this is infeasible as it cannot get under the box at the corridor entrance as it would need to in order to push this box up through the corridor. It then modifies its plan so that this box is instead pushed a longer route that avoids the corridor.

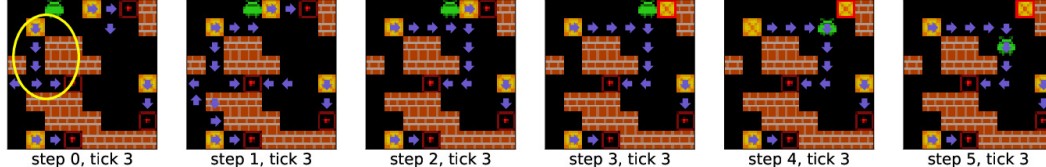

(c) The agent initially plans to push a box down to a target along the circled route. However, over subsequent computational ticks, the agent realizes that this plan would prevent it from solving the level as it would prevent it from ever reaching the lower-left box and target. The agent then modifies its plan so that this box is instead pushed a longer route avoiding the corridor.

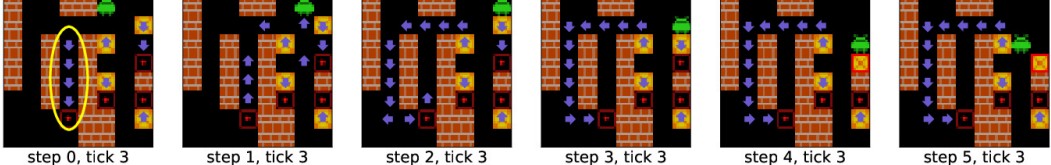

(d) At the initial time step, the agent plans to push a box down through the circled corridor. However, the agent realizes that this is not possible as it cannot get above the box at the corridor entrance as would be required in order to push it down. The agent then updates its plan so that this box is pushed through the further corridor.

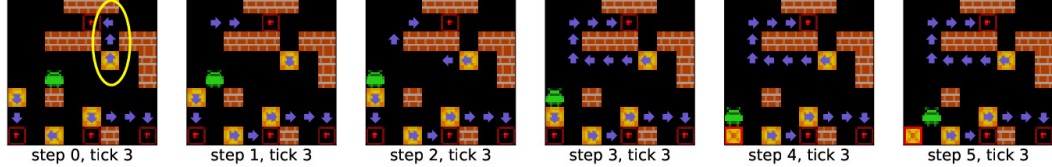

(e) After the first computational tick, the agent plans to push a box along the circled route (e.g. the shortest route to the respective target). However, at the following tick, the agent appears to realizes that this is not possible as the agent would be unable to push the box left as required. The agent then updates its plan so that this box is instead pushed a longer route to the target.

Figure 13: Examples of episodes in which the agent's internal plan initially includes planned routes that would not lead to the agent solving the level. In all these examples, the agent realizes that part of its plan is infeasible and updates its plan accordingly. Blue arrows represent the direction that the agent plans to next push a box off of each square. Yellow circles highlight parts of the agent's plan that are infeasible and later removed. The plans are decoded from the agent's cell state at its first (13a), second (13b and 13c) and third (13d and 13e) layer by a 1x1 probe. The plans are decoded from the agent's cell states at either the final computational tick of the first six steps of episodes (13a, 13b and 13d), or at each computational tick of the first two steps of episodes (13c and 13e).

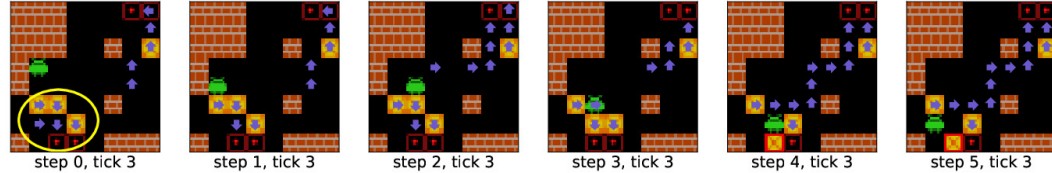

(a) The agent initially plans to push two boxes to the lower left target. However, at later time steps, the agent alters its plan to instead push one of these boxes to the top-right target.

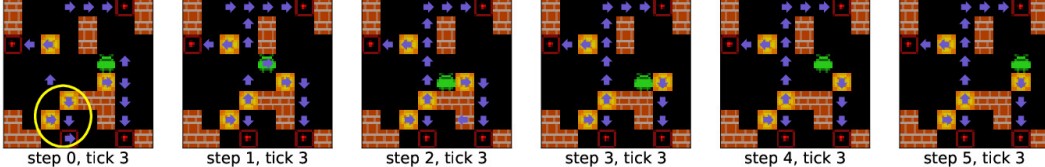

(b) The agent again initially plans to push two boxes to the two lower right targets. Again, at a later time step, the agent adapts its plan to instead push one of these boxes to the top-right target.

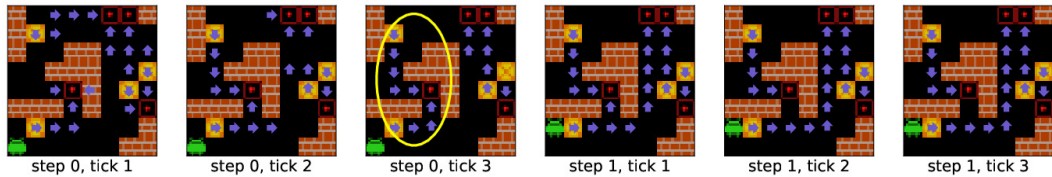

(c) During the initial 3 ticks, the agent plans to push both the lower-left and upper-left boxes to the left-most target. However, the agent appears to realize that the top-left box *must* be pushed to this target (i.e. it is the only target that it is feasible to push the top-left box to). The agent then adapts its plan to instead push the lower-left box to one of the top-most targets.

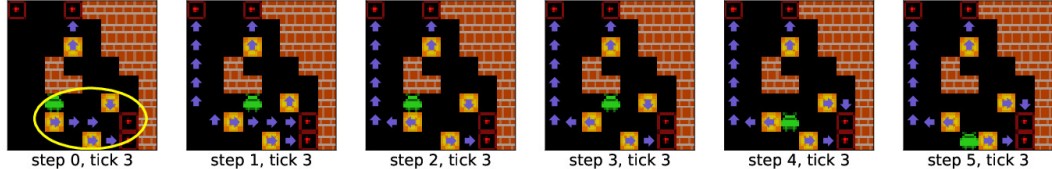

(d) Over the first two time steps, the agent considers pushing the lower-left box forward to one of the lower-right targets. However, this generates a conflict. The agent appears to realize this and then connect this box to a plan it has constructed that links this box to the top-left target.

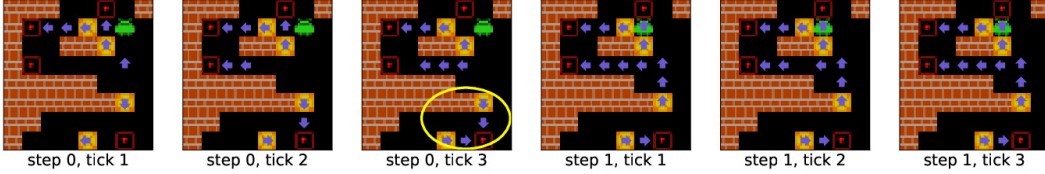

(e) After the third tick, the agent plans to push two boxes to the lower-right target. However, the agent seems to realize that the lowest box *must* be pushed to this target (e.g. the only target the lowest box can feasibly be pushed to is the lower-right target). At the fourth tick, the agent hence plans to instead push the other box that it has associated with the lower-right target to an alternate target.

Figure 14: Examples of episodes in which the agent initially plans to push multiple boxes to the same target before modifying its plan to push one of these boxes to an alternate target. Blue arrows represent the direction that the agent plans to next push a box off of each square. Yellow circles highlight parts of the agent's plan that involve pushing two boxes to the same target. The plans are decoded from the agent's cell state at its first (14a), second (14b and 14c) and third (14d and 14e) layer by a 1x1 probe. The plans are decoded from the agent's cell states at either the final computational tick of the first six steps of episodes (14a, 14b and 14d), or at each computational tick of the first two steps of episodes (14c and 14e).

### A.2.3 FORWARD PLANNING

In the previous two sections, we described forms of evaluation the agent appears to perform as part of its iterative plan-construction process. However, this leaves open the question of how the agent iteratively constructs plans. As explained in Section 5, the agent appears to do so by using some form of learned, iterative search process. In this section, we now describe one major form of iterative, search-based plan-construction the agent performs: planning *forward* from boxes.

Specifically, one of the (two) primary ways the agent appears to construct its internal plans is by iteratively extending its internal plans forward from boxes. That is, the agent seems to frequently 'initialize' plans at box locations, and then iteratively extend these plans forwards towards targets over the early computational ticks of episodes. This is a form of forward search. Algorithms based on forward search – for instance, Monte Carlo Tree Search (Coulom, 2006) – are the predominant form of planning algorithms currently used in model-based planning agents. It is hence notable that the agent appears to have learned a planning algorithm that (partially) relies on forward search.

Instances of the agent constructing its internal plans by iteratively searching forward from boxes can be seen in Figure 15. Figure 15 shows the development of agent's internal plan over the initial steps of five episodes in which the agent constructs part of its internal plan by planning forward from boxes towards targets. As a specific example, consider Figure 15a: the agent iteratively constructs a plan to push the bottom-right box to the top-right target by planning forward from the bottom-right box. Notice that the part of its plan that the agent iteratively constructs forward from this box end up connecting with a partial plan that the agent has constructed by iteratively searching backward from the top-right target. This will be discussed in the following section.

### A.2.4 BACKWARD PLANNING

Iteratively searching forward from boxes is not the only form of search-based planning that the agent appears to engage in. Additionally, the agent appears to search backwards from targets to boxes. That is, the agent will frequently initialize the end of a plan (i.e. by initializing a plan that ends at a specific target), and then iteratively search backwards towards boxes. This is a form of backward search. We hence refer to this as *backwards* planning.

Examples of the agent constructing its internal plans by iteratively searching backwards from targets can be seen in Figure 16. Figure 16 shows the development of the agent's internal plan over the initial steps of five episodes in which the agent constructs part of its internal plan by iteratively planning backwards from targets towards boxes. For instance, consider the manner in which the agent forms its internal plan in the episode shown in Figure 16e. Between the first and fifth tick in this episode, the agent iteratively extends a planned route backwards from the lower-right target to the lower-right box.

Backward search was long studied in the context of planning in 'classic' RL (Moore & Atkeson, 1993). However, whilst some recent work has investigated methods relating to backward planning (Goyal et al., 2018; Van Hasselt et al., 2019; Lee et al., 2019), backward-facing planning is significantly less popular than forward-facing planning in modern model-based RL agents. The fact that the agent has learned to (partially) rely upon backwards planning is, therefore, interesting.

### A.2.5 PARALLEL PLANNING

Finally, the agent appears to be capable of extending multiple plans in parallel over a single computational tick. We refer to this as *parallel* planning. Examples of the agent constructing its internal plans in parallel can be seen in Figure 17. Figure 17 shows the development of the agent's internal plan over the initial steps of five episodes in which the agent utilizes parallel planning. Parallel planning is not something that is common in standard planning algorithms. This is because, unlike the DRC agent that can plan by applying convolution operations to its spatially-extended cell states, standard planning algorithms must extend a single node at a time. We hypothesize that this parallel planning is learned by the agent to further increase the efficiency with which it plans.

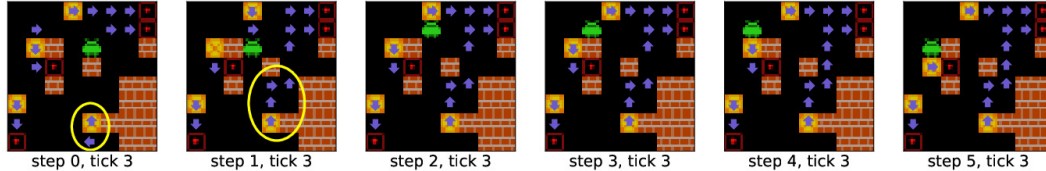

(a) The agent iteratively extends part of its plan forward from the bottom-right box. The agent extends this part of its plan forward until it connects to a part of the agent's plan that connects it to the top-right target.

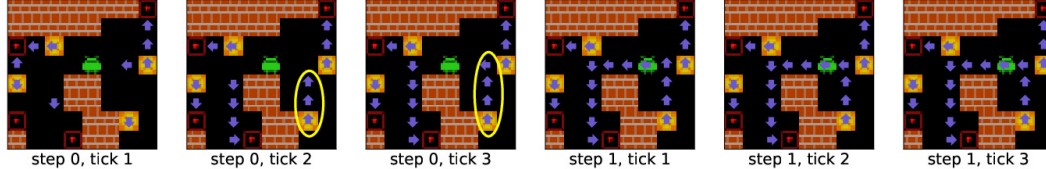

(b) The agent constructs part of its plan by iteratively searching forward from the bottom-right box. The agent searches forward until this part of its plan connects to a part of the agent's plan that connects this box to the bottom-left target.

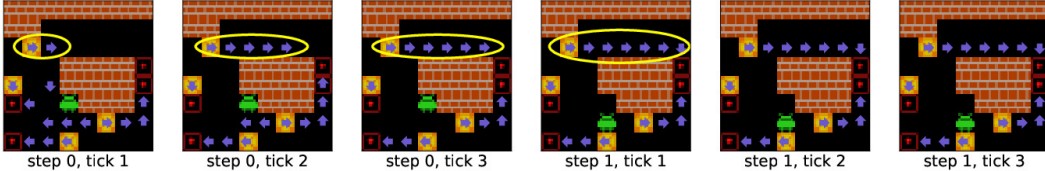

(c) The agent formulates a planned route to push the top-most box by extending its plan forward from this box to the top-right target.

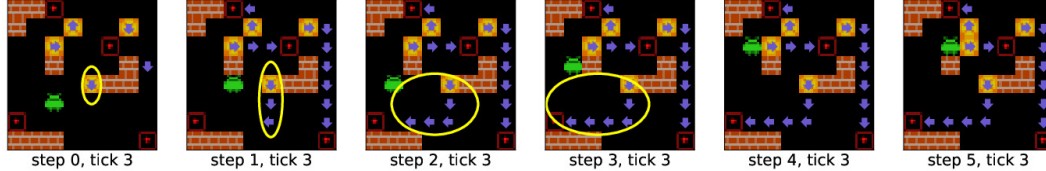

(d) The agent forms a plan for the bottom-most box by searching forward from this box to the lower-left target.

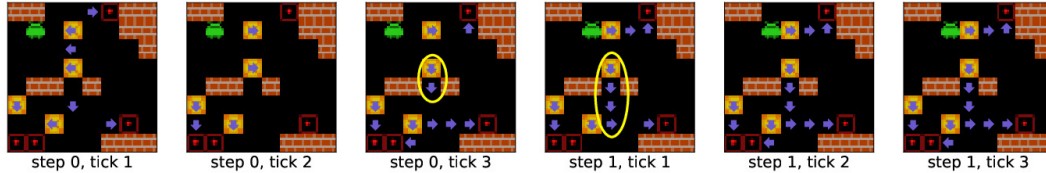

(e) The agent forms a plan for the center-most box by iteratively searching forward from this box to the lower-right target.

Figure 15: Examples of episodes in which the agent formulates its internal plan by iteratively extending planned routes forward from boxes. Blue arrows represent the direction that the agent plans to next push a box off of each square. Yellow circles highlight parts of the agent's plan that it has constructed by iteratively searching forward from boxes to targets. The plans are decoded from the agent's cell state at its first (15a), second (15b and 15c) and third (15d and 15e) layer by a 1x1 probe. The plans are decoded from the agent's cell states at either the final computational tick of the first six steps of episodes (15a, 15b and 15d), or at each computational tick of the first two steps of episodes (15c and 15e).

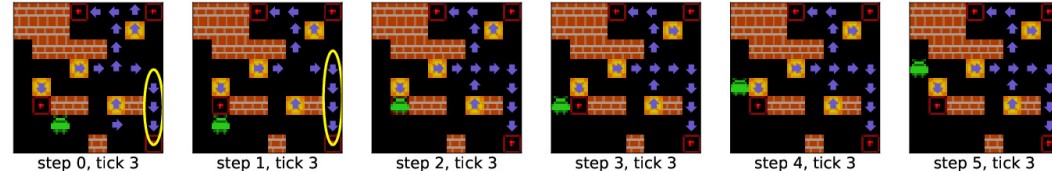

(a) The agent formulates part of its internal plan by iteratively searching backwards from the bottom-right target to the center-most box.

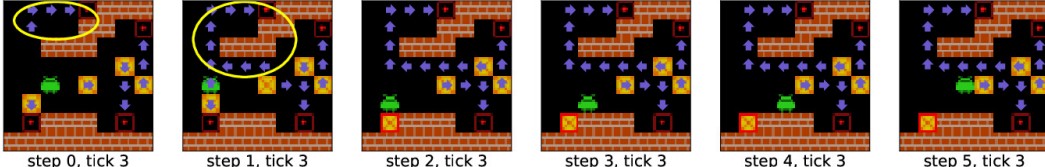

(b) The agent constructs a plan to push a box to the top-most target by iteratively extending a plan backwards from this target. The agent extends this plan backwards until it connects to the top-most box.

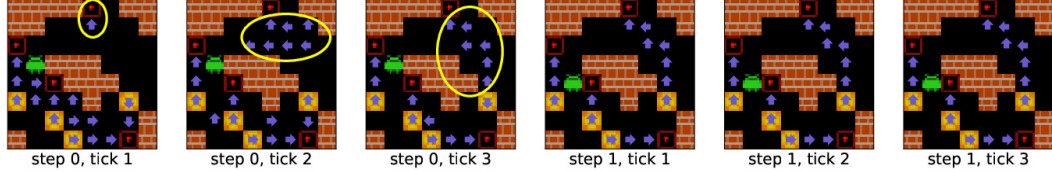

(c) The agent forms an internal plan to push a box to the top-most target by iteratively searching backwards from this target. The agent formulates this part of its plan by searching backwards until this plan connects to the lower-right box.

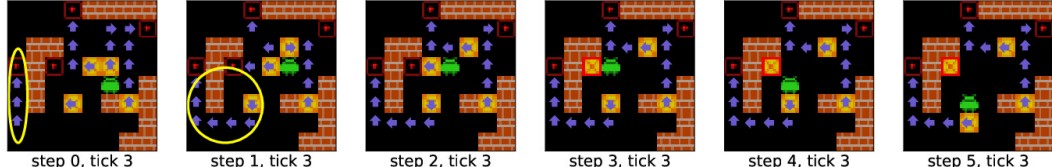

(d) The agent formulates a plan to push the left-most box to the left-most target by searching backwards from the left-most target to the left-most box.

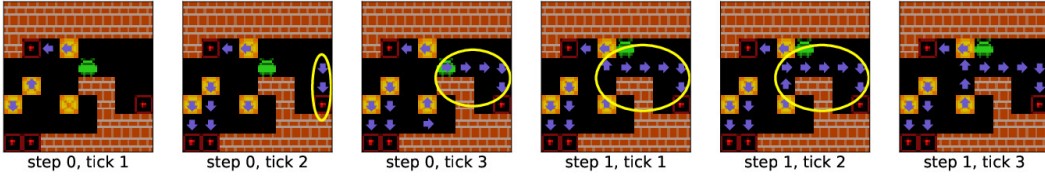

(e) The agent constructs a plan to push a box to the lower-right target by iteratively constructing a plan backwards from this target. The agent extends this plan backwards until it connects to the lower-right box.

Figure 16: Examples of episodes in which the agent formulates its internal plan by iteratively extending planned routes backward from targets. Blue arrows represent the direction that the agent plans to next push a box off of each square. Yellow circles highlight parts of the agent's plan that it has constructed by iteratively searching backwards from targets to boxes. The plans are decoded from the agent's cell state at its first (16a), second (16b and 16c) and third (16d and 16e) layer by a 1x1 probe. The plans are decoded from the agent's cell states at either the final computational tick of the first six steps of episodes (16a, 16b and 16d), or at each computational tick of the first two steps of episodes (16c and 16e).

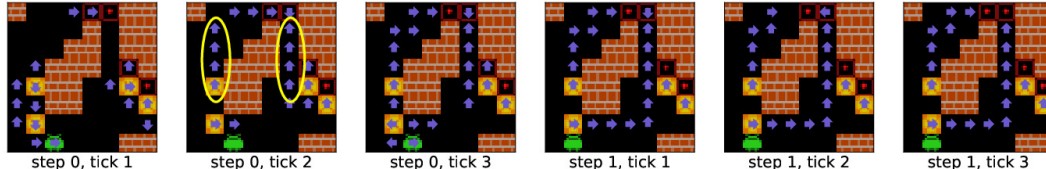

(a) At the fourth computational tick, the agent extends two parts of its plan (e.g. its plans to push boxes to the top-most targets) in parallel.

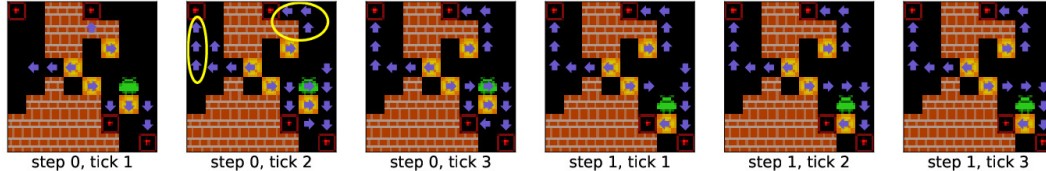

(b) At the second computational tick, the agent iteratively constructs internal plans for the two top-most targets by searching backwards from these two targets in parallel.

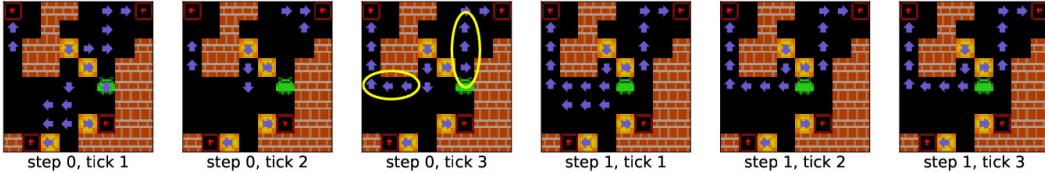

(c) At the third computational tick, the agent iteratively extends its internal plans associated with the two top-most targets by extending these two plans in parallel.

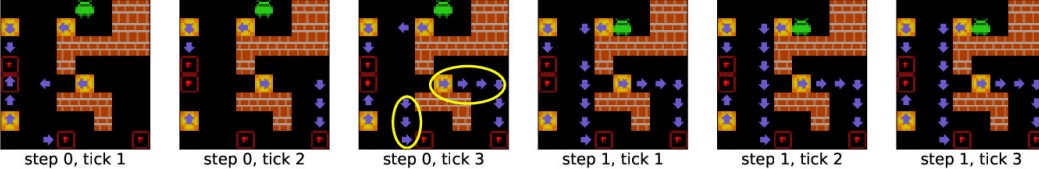

(d) The agent searches backwards from the two bottom-most targets in parallel at the third computational tick.

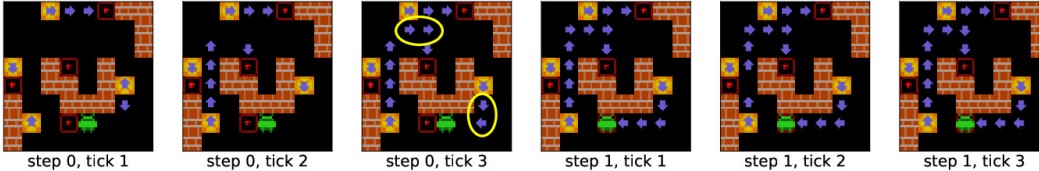

(e) After the third computational tick, the agent extends two parts of its internal plan in parallel.

Figure 17: Examples of episodes in which the agent formulates its internal plan by extending multiple planned routes in parallel over a single computational tick. Blue arrows represent the direction that the agent plans to next push a box off of each square. Yellow circles highlight parts of the agent's plan that it has constructed in parallel over a single tick. The plans are decoded from the agent's cell state at its first (17a), second (17b and 17c) and third (17d and 17e) layer by a 1x1 probe. The plans are decoded from the agent's cell states at either the final computational tick of the first six steps of episodes (17a, 17b and 17d), or at each computational tick of the first two steps of episodes (17c and 17e).

### A.2.6 BLIND PLANNING

In a discrete, deterministic environment such as Sokoban, a natural way for an agent to plan would be for it to search over possible sequences of future actions in search of a sequence of actions that achieved some goal. In Section 5, we noted that the agent's internal plans consistently represent routes connecting boxes and targets. This, alongside all previous visualizations of the agent's plans, suggests that the 'goal' the agent evaluates sequences of actions in terms of when performing search is whether said actions represent a feasible route to push a box along to a target.

If the agent formulates internal plans by searching over potential future actions with the goal of connecting boxes and targets, we would expect the agent's planning algorithm to (at least attempt to) search for plans achieving this goal in *any* Sokoban level so long as that level contained boxes to plan from and targets to plan to. That is, if the agent does indeed formulate internal plans by searching for plans achieving the goal of connecting boxes and targets, we would expect the agent to be able to formulate plans in Sokoban levels drawn from significantly different distributions.

We now provide examples of the agent successfully forming plans in a very different type of level to the levels on which it was trained. Specifically, we provide examples of the agent appearing to search for plans in levels in which it is not itself present. These are Sokoban levels in which the agent observes the level, but is not actually positioned on any square. Note that this represents a significant distribution shift to the levels the agent was trained on. Indeed, the agent can never actually influence these levels. Crucially, however, this distribution shift should not prevent the agent from attempting to form plans if it did indeed plan in the hypothesized manner.

Figure 18 shows the development of the agent's internal plan in levels in which the agent is not itself present. Clearly, the agent (i) still attempts to form plans and (ii) forms internal plans that successfully connect boxes and targets. We take the ability of the agent to continue internally forming plans in the face of this radical distribution shift as evidence that the agent indeed possesses some learned search procedure that searches for plans that achieve the goal of connecting boxes and targets. However, we note that an unexplained curiosity is that, in some such levels, after arriving at a plan the agent will seemingly completely forget it. That is, in some cases of blind planning, the agent will begin to form a plan and then, after many time steps, proceed to forget the plan and represent no plan at all.

### A.2.7 GENERALIZED PLANNING

In the original paper introducing DRC agents, Guez et al. (2019) demonstrated that Sokoban-playing DRC agents trained on the Boxoban dataset of Sokoban levels (i.e. levels containing four boxes and four targets) can successfully generalize to Sokoban levels with additional targets and boxes. Specifically, they showed that such a DRC agent can solve Sokoban levels with additional boxes and targets.

Given the discussion thus far, we hypothesize that the reason for this is that an agent possessing a planning mechanism of the above sort (i.e. an agent that planned by searching for sequences of actions corresponding to routes between boxes and targets) would be able to successfully execute its planning mechanism in such levels. This is because simply introducing a search process that searched for routes connecting boxes and targets could easily generalize to levels in which additional boxes and targets are present.

Figure 19 shows examples of the agent's internal plan at the final tick of the initial six time steps in episodes in which there are either five boxes and targets, or six boxes and targets. As implied by the above discussion, the agent (i) still attempts to form plans and (ii) forms internal plans that successfully connect boxes and targets. We take this as additional evidence of the agent searching for plans that achieve the goal of connecting boxes and targets.

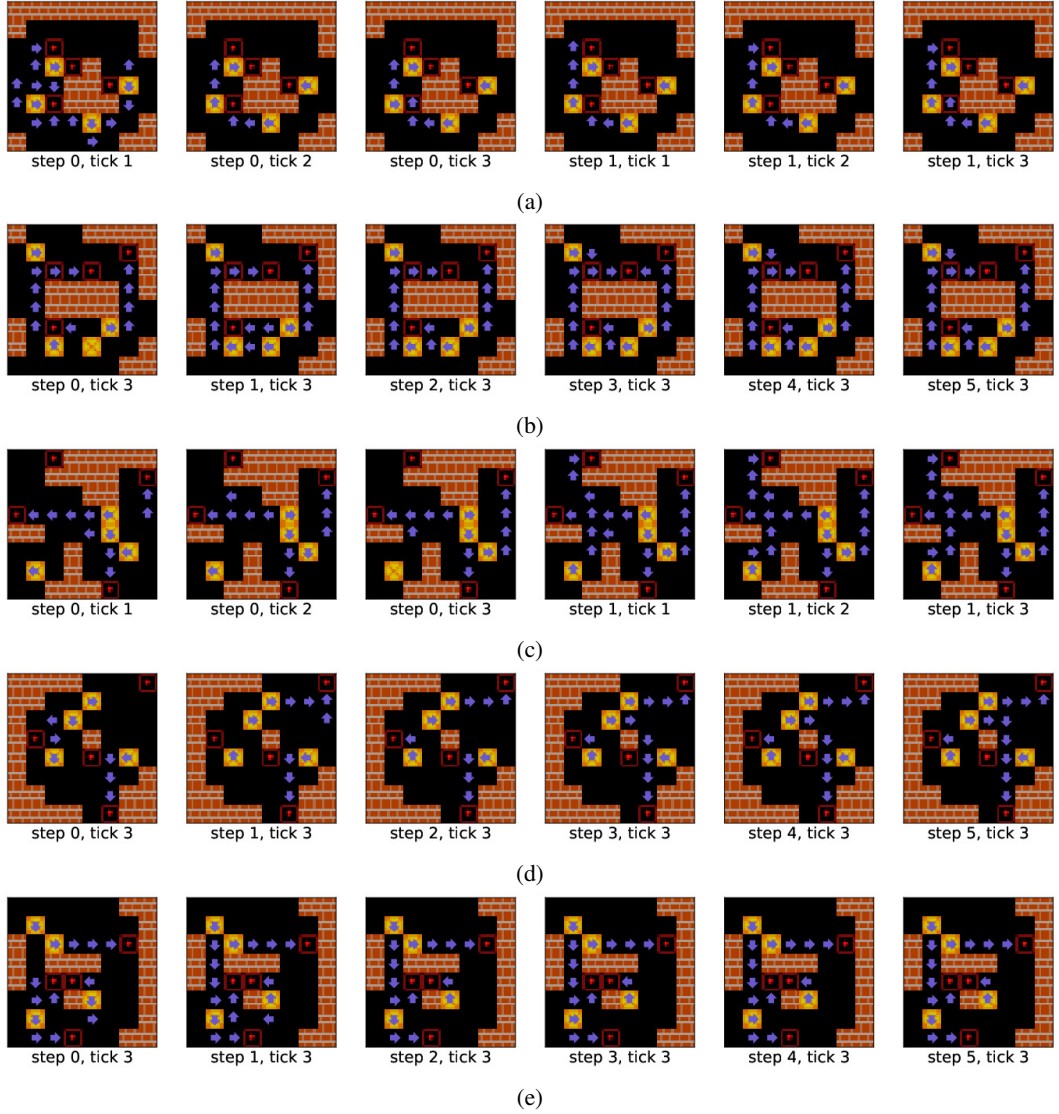

Figure 18: Examples of episodes in which the agent formulates an internal plan despite not being present in the level. Blue arrows represent the direction that the agent plans to next push a box off of each square. The plans are decoded from the agent's cell state at its first (18a), second (18b and 18c) and third (18d and 18e) layer by a 1x1 probe. The plans are decoded from the agent's cell states at the final computational tick of the first six steps of episodes.

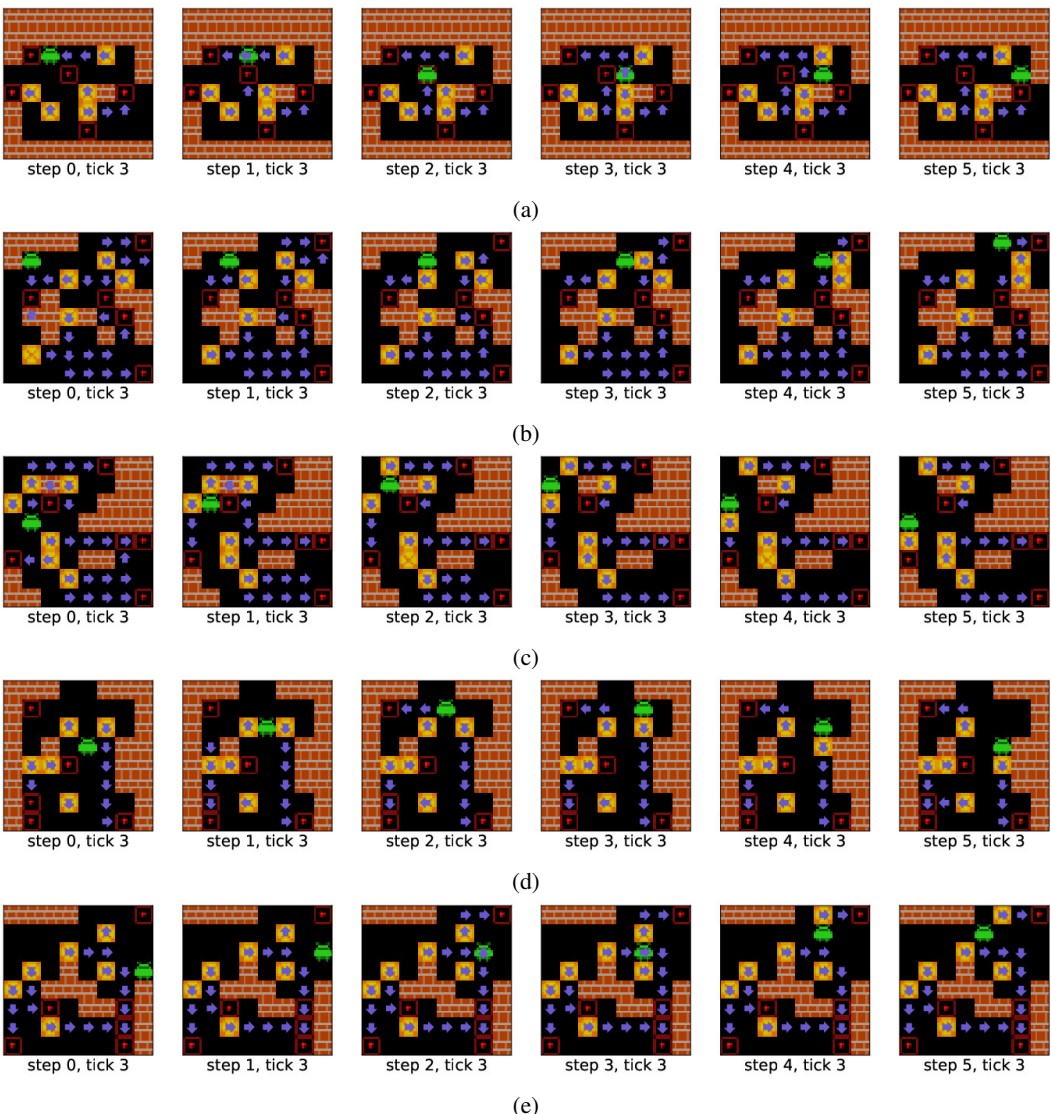

Figure 19: Examples of episodes in which the agent formulates its internal plan despite there being more boxes and more targets than in the levels on which it was trained. Blue arrows represent the direction that the agent plans to next push a box off of each square. The plans are decoded from the agent's cell state at its first (19a), second (19b and 19c) and third (19d and 19e) layer by a 1x1 probe. The plans are decoded from the agent's cell states at the final computational tick of the first six steps of episodes. These episodes take place in levels in which there are five boxes and targets (19a, 19b and 19d), and in levels in which there are six boxes and targets (19c and 19e)

### A.2.8 BLOCKED-ROUTE PLANNING

The previous two sub-sections provided examples of the agent successfully formulating plans in levels that represented significant distribution shifts relative to the training distribution. These previous distribution shifts aimed to induce changes to the agent's environment that would not impede the planning capabilities of an agent that planned via searching for plans that achieved the implicit goal of connecting boxes and targets. In this sub-section and the following sub-section we now consider different forms of distribution shift. Namely, we now consider distribution-shifted Sokoban levels that aim to test the ability of the agent to dynamically evaluate and update its internal plan in response to environmental changes.

We begin by considering Sokoban levels in which, at a time step following the initial time step, an additional wall square is added to the level. Specifically, this wall square is added to a location that blocks an obvious route between a box and a target. The aim of investigating these levels is to determine whether the agent is capable of evaluating that this additional wall square invalidates its current plan, and whether the agent can dynamically form a new plan after doing so. Figure 20 shows the manner in which the agent's internal plan develops in levels in which, at a time step following initialization, we add a wall square to block off an obvious route between a box and a target. Clearly, the agent is capable of (i) recognizing that the added wall invalidates its initial plan and (ii) dynamically adjusting its plan accordingly. We take this as evidence to support the hypothesis that the agent forms plans via an evaluative search process.

### A.2.9 NEW-ROUTE PLANNING

Given the ability of the agent to dynamically update its plans in response to the addition of a wall square to block off an optimal route to push a box, an obvious question to ask is whether the agent can dynamically update its plan in levels representing the reverse type of distribution shift. That is, can the agent dynamically update its plans in levels in which, at some time step following initialization, we remove a wall square to open up an optimal route to push a box to a target that is infeasible prior to the removal of the wall?

Figure 21 shows the development of the agent's internal plan in levels where we, at some time step following initialization, remove a wall square to open up an optimal route to push a box to a target that is infeasible prior to the removal of the wall. In some levels – for example, in Figure 21a – the agent does dynamically respond to this wall-removal by updating its plan to exploit the new, optimal route. However, in other levels – such as in Figure 21b – the agent does not do this. We conjecture that this is potentially due to the agent having a notion of a 'completed route' within its internal plan. That is, we conjecture that the agent represents some plans as being complete and requiring no further search, and this is why the agent modifies its plan following the removal of a wall in some cases but not others.

### A.2.10 DISCUSSION

In discrete, deterministic, fully-observable environments like Sokoban, an agent with access to a perfect environment model can reformulate the problem of 'planning' as the problem of searching for a sequence of future actions – a plan – that achieves a goal (Russell & Norvig, 2010). The agent studied in this paper lacks such a perfect world model.

However, we have demonstrated that the agent we study has learned a spatial correspondence between its cell states and the Sokoban grid, such that it can represent spatially-localized concepts at corresponding positions of its cell state. This can, perhaps, be seen as a learned 'implicit' model of the environment. Importantly, this learned implicit world model is sufficient to (i) represent sequences of future actions and (ii) predict relevant consequences of these actions on the environment. It hence appears to be sufficient to allow the agent to plan via search.

We believe the examples provided in the previous sections support the hypothesis that the agent indeed plans via applying a learned search algorithm to a learned, 'implicit' model of its environment. This is interesting as it implies that the agent's emergent planning capabilities represent a learned analogue to the planning-capable agents introduced in Appendix E.1 that plan via applying an explicit search algorithm to an explicit world model.

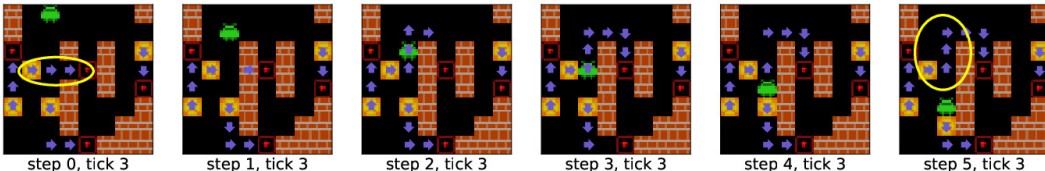

(a) After the first step, a wall is added to block the agent's planned route between center-most box and target. Over the subsequent time steps, the agent realizes this and dynamically forms a new planned route connecting this box and target.

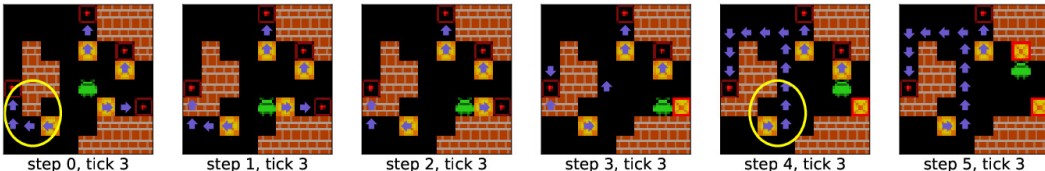

(b) After the first step, a wall is added to block the agent's planned route between left-most box and target. Over the subsequent time steps, the agent realizes this and dynamically forms a new plan that involves pushing this box an alternate, longer route to this target.

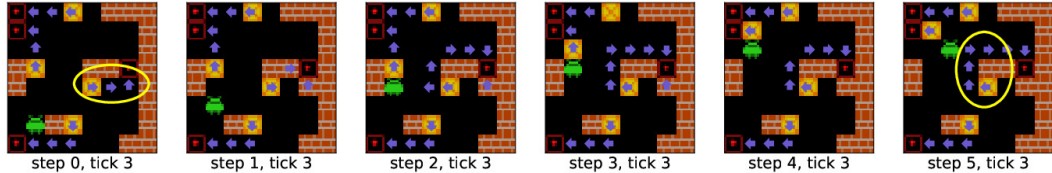

(c) After the first step, a wall is added to block the agent's planned route between right-most box and target. During the steps that follow, the agent realizes that this invalidates its initial plan and dynamically forms a new plan that involves pushing this box an alternative route.

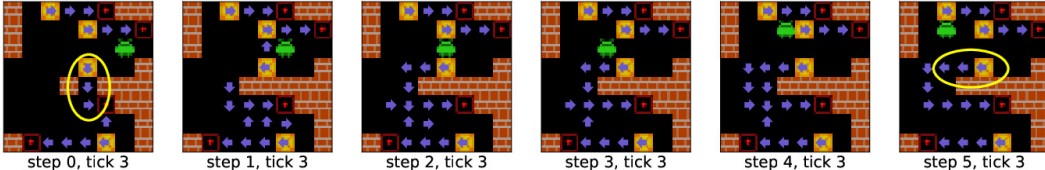

(d) Initially, the agent plans to push the central box down to the central target. After the first step, a wall is added to block this route. During the following steps, the agent realizes that this has occurred and dynamically forms a new plan that involves pushing this box left, down, and then right, to this target.

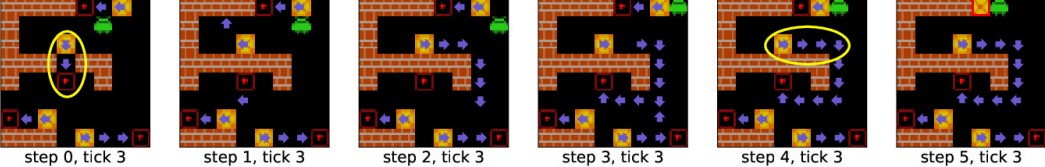

(e) Initially, the agent plans to push the central box down to the central target. However, after the first step, a wall is added to block off this route. During the following steps, the agent realizes that its initial plan is now infeasible and dynamically forms a new plan that instead involves pushing this box right, down, left, and then up, to this target.

Figure 20: Examples of the agent formulating its internal plan in levels in which a wall square is added to the environment to block an obvious route between a box and target at a time step following initialization. Blue arrows represent the direction that the agent plans to next push a box off of each square. Yellow circles highlight relevant parts of the agent's plan before and after the additional wall square is added. The plans are decoded from the agent's cell state at its first (20a), second (20b and 20c) and third (20d and 20e) layer by a 1x1 probe. The plans are decoded from the agent's cell states at the final computational tick of the first six steps of episodes.

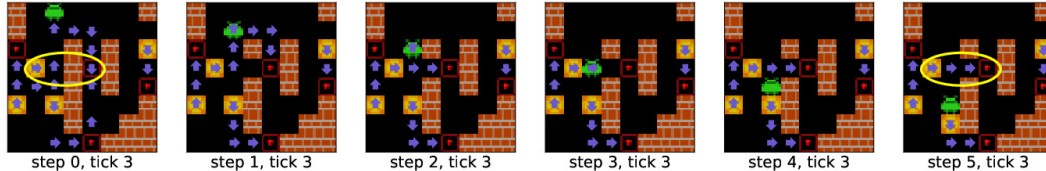

(a) The agent begins planning to push the upper-left box up, right, and down to the center-most target. However, after the first step, a wall is removed such that this box can instead be pushed straight right to the target. The agent realizes this and updates its plan accordingly.

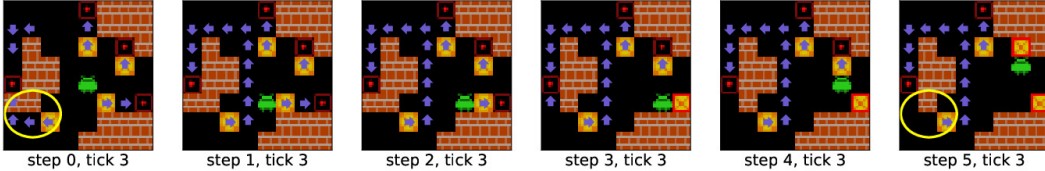

(b) Initially, the agent plans to push the left-left box right, up, left, and down to the left-most target. However, after the third step, a wall is removed such that this box can instead be pushed a shorter route (i.e. left and the up) to the target. The agent does not realize this and does not update its plan in response.

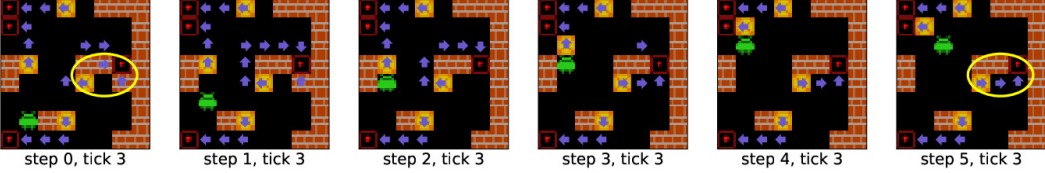

(c) After the second step a wall is removed. The removal of this wall means that the optimal route to push the right-most box to the right-most target is to push it right and then up. Before the removal of the wall the agent plans to push it left, up and then right. However, after the removal, the agent updates its plan to account for the new optimal route.

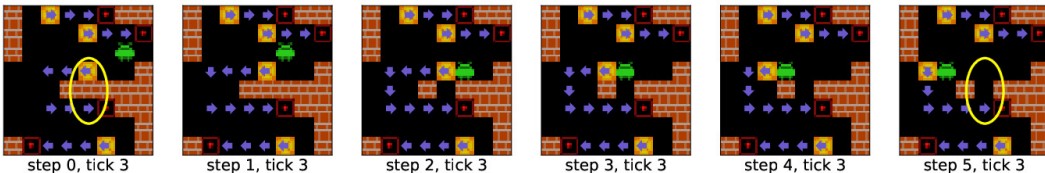

(d) The agent initially plans to push the right-most box up and around the wall that separates it from the right-most target. After the second step, a wall is removed such that this box can now be pushed a shorter route to this target. The agent fails to respond to this.

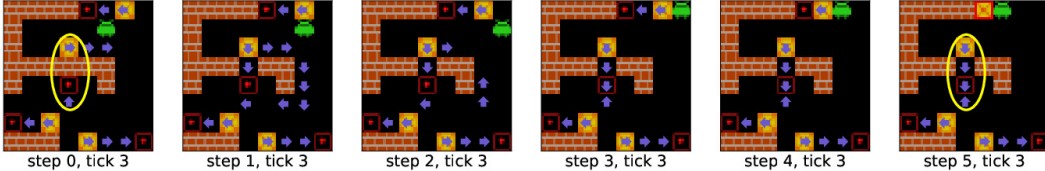

(e) The agent initially plans to push the center-most box around the wall that separates it from the center-most target. After the first step, a wall square is removed such that this box can now be pushed directly down to this target. The agent realizes this and updates its plan to account for the new optimal route to push this box.

Figure 21: Examples the agent formulating internal plans in episodes in which a wall square is removed at a time step after initialization. Removing this wall square opens up a route that it is optimal to push a box to a target through. Blue arrows represent the direction that the agent plans to next push a box off of each square. Yellow circles highlight the relevant parts of the agent's plan (or lack of internal plan) before and after the wall square is removed. The plans are decoded from the agent's cell state at its first (21a), second (21b and 21c) and third (21d and 21e) layer by a 1x1 probe. The plans are decoded from the agent's cell states at the final computational tick of the first six steps of episodes.

This perspective on the agent's concept representations – i.e. that, by enabling the agent form plans and evaluate their consequences, collectively, these representations play a role that can be seen as the role of a learned implicit world model, – aligns with work that has emphasized the importance of world models for generalization capabilities (Richens & Everitt, 2024; Andreas, 2024). It also provides new insights regarding learned world models in RL. Specifically, it complements past work that has investigated explicitly training world models (Ha & Schmidhuber, 2018; Freeman et al., 2019) by showing that world models – or, at least, representations that can play a role traditionally played by world models - -can also emerge spontaneously within the representations of a generic agent.

Additionally, it is interesting that the agent constructs its internal plans by simultaneously searching forward from multiple boxes *and* searching backwards from multiple targets. That is, the agent appears to have learned a form of *parallelized bidirectional planning*. This is very different to the agents introduced in Appendix E.1 that primarily rely on forward search algorithms.

Whilst some past work has had considerable success in applying bidirectional search to RL (Edwards et al., 2018; Lai et al., 2020), RL agents making use of bidirectional planning are still remarkably rare. This is likely due to the difficulty in specifying which states to plan forwards and backwards from in many environments. Indeed, Sokoban is somewhat unique in that it is especially well-suited for bidirectional search. This is because there are obvious candidates to plan forwards from (boxes) and backwards from (targets). As such, the main takeaway from the emergence of bidirectional search within the agent is likely *not* that bidirectional search should be applied more widely within RL.

Instead, we believe the main takeaway from this finding to be that there are benefits to allowing agents to *learn* a planning algorithm (and a implicit-world model to apply it to) rather than forcing an agent to use a handcrafted planning algorithm. This is because the agent can learn to plan in a way that is well-suited for the environment it finds itself in. We suspect this idea – of allowing agents to *learn* search algorithms well-suited for specific domains, rather than forcing them to use a generic, handcrafted search algorithms such as MCTS – may become increasingly prevalent.

This is to say that we hypothesize that the agent we study has learned a form of planning that is especially well-suited to Sokoban relative to other planning algorithms. Bidirectional search is known to be very efficient in certain environments (Kaindl & Kainz, 1997; Russell & Norvig, 2010; Sturtevant et al., 2020). Intuitively, this is because it is more efficient to form plans by searching backwards from goals and forwards from initial states (i.e. because these two plans can 'meet in the middle') than to form plans by either of these means alone. Furthermore, as Sokoban is characterized by actions having negative consequences in the long-run, efficient planning is crucial in Sokoban. This is because forming plans quickly at the start of episodes allows the agent to avoid taking actions early on that would make a level unsolvable. Thus the agent studied in this paper appears to have learned a domain-specific planning algorithm that works well in the environment it finds itself in. Evidence of this can be seen in the fact that one of the highest-performing Sokoban agents that does *not* rely on deep learning also uses a form of forward-backward planning Shoham & Elidan (2021).

### A.3 Further Results Regarding Iterative Plan Refinement

In Section 5, we used Figure 6 to demonstrate that the agent can use additional test-time compute at the start of episodes to refine its internal plan. This would be expected if the agent constructed plans using some form of learned search, since the agent would be able to use additional compute to perform a more thorough search. This additionally helps explain the ability of DRC agents to perform better in Sokoban when given 'thinking time' steps (Guez et al., 2019; Taufeeque et al., 2024) as we have shown that the extra compute afforded by 'thinking time' facilitates plan refinement. In this section, we further investigate the agent's ability to iteratively refine its internal plans when forced to perform 'thinking time' steps – that is, forced stationary steps at the start of episodes – prior to acting.

### A.3.1 Test-Time Plan Refinement Across Layers

In Section 5, Figure 6 only demonstrated that the agent can use 'thinking time' iteratively refine its plan in its final layer. To investigate whether the agent can iteratively refine its plans at all layers when given additional test-time compute prior to acting, we thus again forced the agent to perform

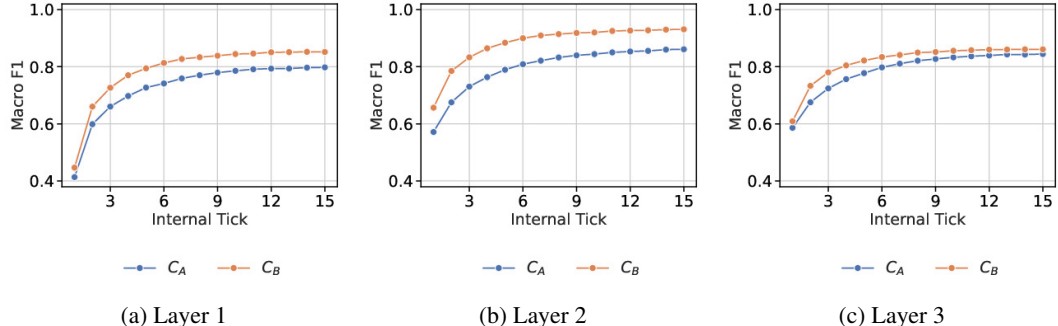

(a) Layer 1        (b) Layer 2        (c) Layer 3

Figure 22: Macro F1 (averaged over 1000 episodes) when using the agent's internal plan at each internal tick during the first 5 steps of an episode to predict (a) the agent's future movements ($C_A$) and (b) future box movements ($C_B$). The 'internal plan' at a layer for a tick corresponds to the agent's representation of $C_A$ and $C_B$ as decoded by a 1x1 probe applied to the agent's cell state at that layer.

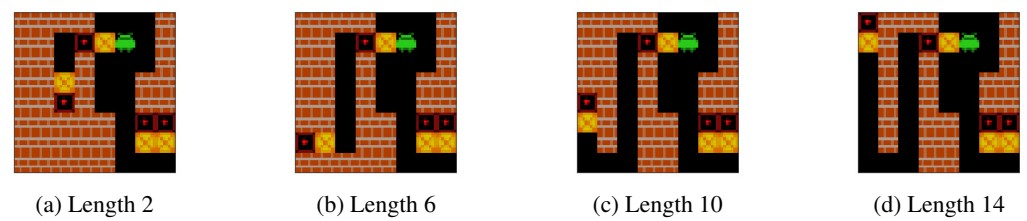

(a) Length 2      (b) Length 6      (c) Length 10      (d) Length 14

Figure 23: Example of one of the levels used to test for behavioral evidence of search when its corridor is of different lengths.

5 'thinking time' steps before beginning to act in 1000 episodes. As with Figure 6, after each of the 15 internal computational ticks performed by the agent during these steps, we applied 1x1 probes to decode the agent's representations of $C_A$ and $C_B$ for each square of the observed Sokoban board at that tick. As argued previously, the agent's internal representations of the concepts over the entire Sokoban board can be seen as its internal plan. We viewed the agent's internal plan at each tick as a prediction of its future behavior ($C_A$) and the effect of this future behavior on the environment ($C_B$), and measured the correctness of these prediction using the macro F1 score. The results can be seen in Figure 22. Clearly, the agent's internal plan gets iteratively refined over the course of 'thinking time' at *all* layers.

### A.3.2 EVIDENCE OF TEST-TIME COMPUTE BEING USED FOR SEARCH

In this paper, we have provided qualitative evidence that supports the hypothesis that the agent plans via learned search procedure, and that the agent reason the agent benefits from additional test-time compute is because it uses this extra compute to search more thoroughly prior to acting. In this section we now complement this with *behavioral* evidence of the agent using extra test-time compute for search.

We do this using a dataset of handcrafted levels. These levels all follow a common schematic. Namely, there is a corridor with a single entrance. At the end of this corridor is a box and a target. The entrance square of the corridor also has a target on it. There is a box adjacent to this target at the entrance to the corridor. In these levels, the agent always starts adjacent to this box. Thus, at the initial time step, a myopic agent will always push this box on to the target. However, doing so blocks off the corridor (e.g. it prevents the agent from ever reaching the box and target at the end). So, we would expect a planning-capable agent to realize this, and instead plan to push the box out of the way so it can enter the corridor. We create 8 handcrafted levels of this sort. For each level, we create a copy where its corridor is of length 2, 6, 10 and 14. Figure 23 shows a version of one of these levels with these 4 corridor lengths. We then reflect and rotate each level. Thus, we have a dataset of 80 such levels with corridors of each length.

Figure 24 shows the percentage of these levels the agent solves when given between zero and 5 thinking steps prior to acting. If the agent planned via search, we would expect it to struggle to solve these levels without additional test-time compute. This is because, plausibly, the agent's search procedure would take multiple steps to extend backward from the end of the corridor to the corridor entrance (i.e. to inform the agent that it should not act myopically). This is clearly the case in Figure 24. Additionally, if the agent planned via search, we would expect it to take more test-time compute to solve levels with longer corridor. This is because, if the agent planned via search, it would presumably take longer for the search process to account for the effect of blocking off the corridor (i.e. by myopically pushing a box onto the target at the entrance) in levels with longer corridors. The expected pattern of the agent requiring more 'thinking steps' to solve levels with longer corridors can be seen in Figure 24. For instance, the number of 'thinking steps' the agent requires to solve at least half of each set of levels increases with the corridor length of these levels. The agent requires 0 thinking steps to solve at least half of the levels with corridors of length 2, 1 thinking step for levels with corridors of length 4, 2 thinking steps with corridors of length 10, and 3 thinking steps to solve levels with corridors of length 14.

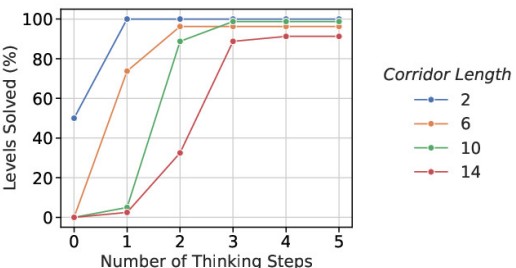

Figure 24: The percentage of the 80 levels introduced in Appendix A.3.2 that the agent solves with different numbers of 'thinking steps' (forced stationary steps prior to acting)

Qualitative evidence of the agent using the additional test-time compute given to it by 'thinking steps' to perform a more thorough search can be seen in Figure 25. In Figure 25, we visualize the agent's internal plan (as formulated in terms of the squares the agent expects to step onto) at the final tick of each 5 additional steps of computation given to the agent when it performs 5 steps of 'thinking time' prior to acting in levels with corridors of length 14. In all of these levels, the agent at the third tick plans to step directly onto the box, myopically pushing it onto the target and making the level unsolvable. Note that this corresponds to the agent's plan without 'thinking steps' and thus explains why the agent fails all of these levels by default. However, over the subsequent steps, the agent iteratively searches backwards from the box at the corridor end. Once this search extends backwards onto the target at the corridor entrance, the agent seems to realize that it should not myopically push a box onto this target as it needs to instead step onto this target to enter the corridor. The agent then alters its plan to instead push the box at the corridor entrance out of the way and enter the corridor.



(a) The agent initially plans to step up into the circled box, pushing the box onto the circled target and blocking off the corridor. However, when given additional 'thinking time', the agent extends its planned route backwards from the end of the corridor and realizes that it needs to step onto this target. It hence changes its plan so that it will step onto, and thus push, the box right so that it can enter the corridor.

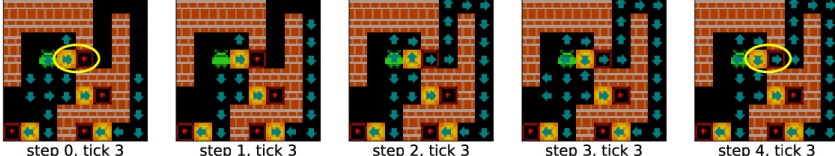

(b) After the third computational tick, the agent plans to step right and push the circled box on to the circled target. Over subsequent steps of 'thinking time', the agent plans backwards from the box at the end of the corridor and realizes that it needs to step into the corridor to reach this box. It thus instead plans to step down onto the box, moving it out of the way.



(c) Initially, the agent plans to step up, pushing the circled box onto the circled target. However, the agent extends its plan backwards from the box at the corridor end and realizes that it needs to step onto this target in order to enter the corridor and reach the box at the corridor end. As such, it alters its plan to first push the circled box left rather than myopically pushing it on to the target.



(d) Without 'thinking steps', the agent would push the circled box right onto the circled target. However, during 'thinking steps', the agent extends a path backwards from the box at the end of the corridor to the circled target. It then plans to instead push the circled box down so that it can follow this path.



(e) Initially, the agent plans pushing the circled box left onto the circled target. However, the agent extends the path it plans to follow to the box at the corridor end backwards, and realizes that it needs to step onto the circled target to reach this box. The agent then alters its plan to first push the circled box down.

Figure 25: Examples of the agent's plan (in terms of $C_A$) over extra steps associated with 5 'thinking steps' in levels with corridors of length 14 as introduced in Appendix A.3.2. During thinking steps, the agent searches backwards from the end of the corridor, and realizes that it must not myopically block this corridor off. The agent fails all of these levels when not forced to perform 'thinking steps', but, when given 5 'thinking steps' the agent successfully solves all levels. Yellow circles highlight the box and target for which the agent changes its plans. Teal arrows represent the direction that the agent plans to next move on to each square. The plans are decoded from the agent's cell state at its first (25a), second (25b and 25c) and third (25d and 25e) layer by a 1x1 probe. The plans are decoded at the final tick of each extra step performed during 5 steps of 'thinking time'.

# B    ADDITIONAL INTERVENTION RESULTS

In Section 6 we provided evidence indicating that the agent's representations of $C_A$ and $C_B$ were responsible for the agent's planning-like behavior. Specifically, in Section 6.1 we outlined the results of experiments in which we used the vector representations of $C_A$ and $C_B$ learned by 1x1 probes to intervene on the agent's cell state to force the agent to formulate and execute sub-optimal plans in Agent-Shortcut and Box-Shortcut levels. The aim of these experiments was to demonstrate that the plans the agent internally formulated using its representations of $C_A$ and $C_B$ causally influenced its behavior in the manner that would be expected of plans.

In this section, we provide further results regarding these experiments.

- Appendix B.1 provides additional examples of the qualitative effect of the interventions from Section 6 on the agent's internal plan.

- Appendix B.2 outlines the results of additional intervention experiments in Agent-Shortcut and Box-Shortcut levels. These additional intervention experiments investigate altering the number of squares intervened upon, and introducing an intervention strength parameter.

- Appendix B.3 details alternate intervention experiments in a new set of handcrafted levels. These levels are designed to test the ability of interventions to force the agent to act optimally when it otherwise would not.

## B.1    ADDITIONAL EXAMPLES OF INTERVENTIONS

In Section 6.1, we provided examples of plans as decoded from the agent's final layer cell state by a 1x1 probe after the first 4 time steps of Box-Shortcut and Agent-Shortcut episodes both when we did, and when we did not, intervene on the agent's final layer cell state. We noted that, when visualizing the agent's plans as decoded by 1x1 probes, we could see that the interventions had the effect of causing the agent to internally formulate and execute the desired type of sub-optimal plan (e.g. a plan that involves either following a longer-than-necessary path, or that involves pushing a box a longer-than-necessary route to a target). We now provide additional examples to further illustrate this. Figure 27 provides additional example Box-Shortcut interventions and Figure 26 provides additional example Agent-Shortcut interventions.

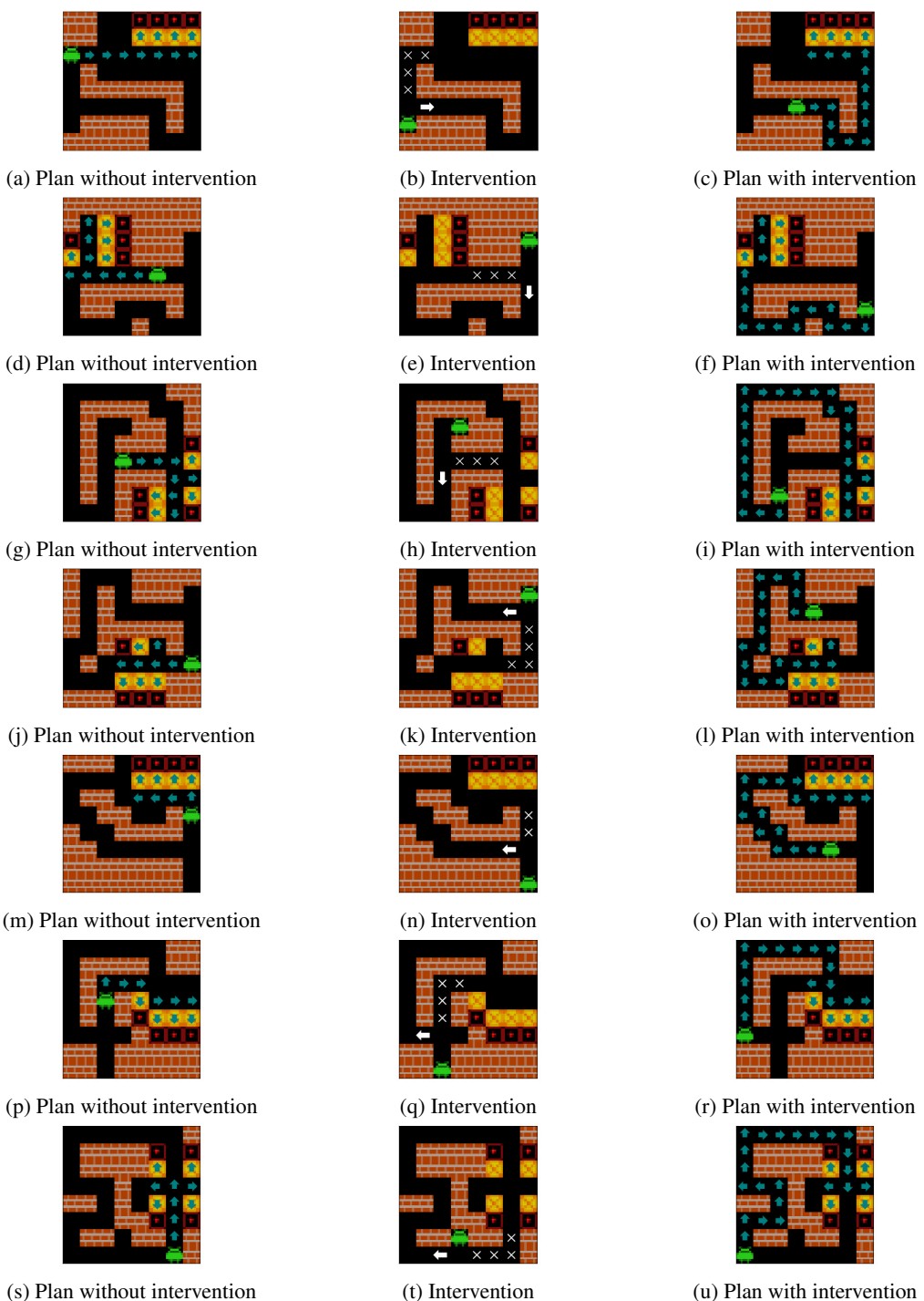

Figure 26: Examples of Agent-Shortcut interventions and their effects on the agent's internal plan. Each row shows (1) the agent's internal plan after 4 steps in a level *without* the intervention, (2) the initial state of that level, and the intervention performed, and (3) the agent's internal plan after 4 steps in that level *with* the intervention. Plans are decoded from the agent's final layer cell state by a 1x1 probe. Teal arrows mean the agent plans to next step onto the associated square in the associated direction. White arrows mark positions which the associated directional representations of $C_A$ are added to. White crosses mark positions of the agent's cell state that representations of NEVER for $C_A$ are added to.

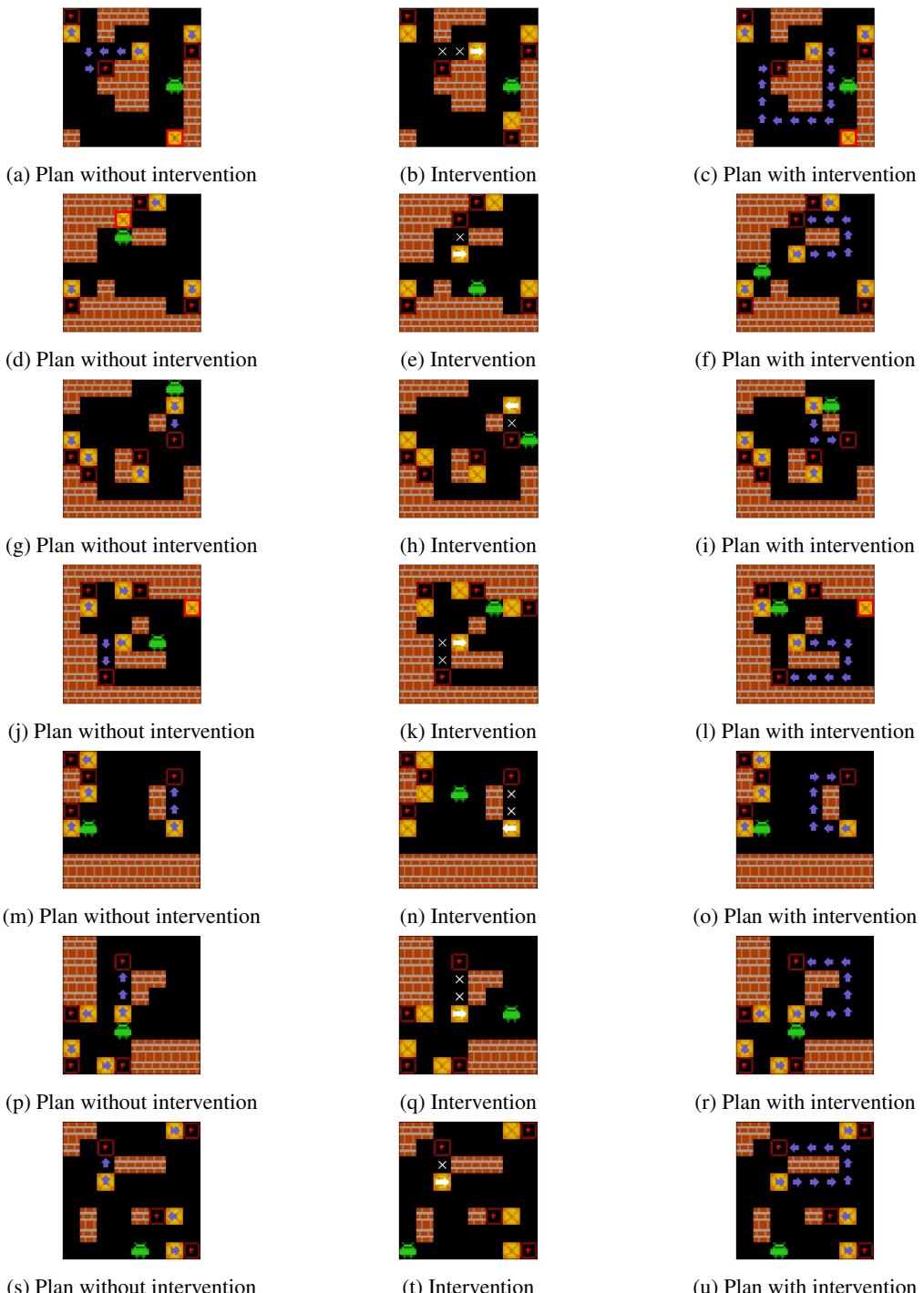

(a) Plan without intervention     (b) Intervention     (c) Plan with intervention

(d) Plan without intervention     (e) Intervention     (f) Plan with intervention

(g) Plan without intervention     (h) Intervention     (i) Plan with intervention

(j) Plan without intervention     (k) Intervention     (l) Plan with intervention

(m) Plan without intervention     (n) Intervention     (o) Plan with intervention

(p) Plan without intervention     (q) Intervention     (r) Plan with intervention

(s) Plan without intervention     (t) Intervention     (u) Plan with intervention

Figure 27: Examples of Box-Shortcut interventions and their effects on the agent's internal plan. Each row shows (1) the agent's internal plan after 4 steps in a level *without* the intervention, (2) the initial state of that level, and the intervention performed, and (3) the agent's internal plan after 4 steps in that level *with* the intervention. Plans are decoded from the agent's final layer cell state by a 1x1 probe. Blue arrows mean the agent plans to push a box of the associated square in the associated direction. White arrows mark positions which the associated directional representations of $C_B$ are added to. White crosses mark positions of the agent's cell state that representations of NEVER for $C_B$ are added to.

---

**Algorithm 1** Agent-Shortcut Intervention

---

1: *ShortRouteSquares* ← All positions $(x, y)$ on the short route
2: $(x_0, y_0)$ ← The first square $(x, y)$ of the long route.
3: *LongRouteSquaresDirs* ← The first $p$ squares $(x, y)$ that the agent would step onto if following
   the longer route, and the direction DIR it would step onto them
4: **for** $t$ in $1, 2, \cdots, EpisodeLength$ **do**
5:     **for** $(x, y)$ in *ShortRouteSquares* **do**                      ▷ Short-route intervention
6:         $c_{(x,y)} \leftarrow c_{(x,y)} + \alpha \times w_{\text{NEVER}}^{C_A}$
7:     **if** Agent has not moved onto $(x_0, y_0)$ this episode **then**
8:         **for** $((x, y), \text{DIR})$ in *LongRouteSquaresDirs* **do**       ▷ Directional intervention
9:             $c_{(x,y)} \leftarrow c_{(x,y)} + \alpha \times w_{\text{DIR}}^{C_A}$

---

---

**Algorithm 2** Box-Shortcut Intervention

---

1: *ShortRouteSquares* ← All positions $(x, y)$ on the short route
2: $(x_0, y_0)$ ← The initial position $(x, y)$ of the box that is not adjacent to any targets.
3: *LongRouteSquaresDirs* ← The first $p$ squares $(x, y)$ that a box would be pushed off of if pushed
   the longer route, and the direction DIR it would be pushed
4: **for** $t$ in $1, 2, \cdots, EpisodeLength$ **do**
5:     **for** $(x, y)$ in *ShortRouteSquares* **do**                      ▷ Short-route intervention
6:         $c_{(x,y)} \leftarrow c_{(x,y)} + \alpha \times w_{\text{NEVER}}^{C_B}$
7:     **if** Agent has not pushed a box off of $(x_0, y_0)$ this episode **then**
8:         **for** $((x, y), \text{DIR})$ in *LongRouteSquaresDirs* **do**       ▷ Directional intervention
9:             $c_{(x,y)} \leftarrow c_{(x,y)} + \alpha \times w_{\text{DIR}}^{C_B}$

---

### B.2 ADDITIONAL INTERVENTION EXPERIMENTS: FURTHER AGENT-SHORTCUT AND BOX-SHORTCUT INTERVENTION EXPERIMENTS

We now consider performing alternate intervention experiments in Box-Shortcut and Agent-Shortcut levels. To reiterate, Agent-Shortcut levels are characterized by the agent having to choose to follow either a longer or a shorter path from its initial position to a region with boxes and targets. Similarly, Box-Shortcut levels are characterized by there being one box that can be pushed either a shorter or a longer route to a target. In both levels, it is optimal for the agent to select the shorter option, and this is indeed what the agent does when not intervened upon. Thus, our interventions aimed to force the agent to formulate and execute a sub-optimal plan involving choosing the longer option.

Our interventions in Box-Shortcut and Agent-Shortcut levels consisted of two sub-interventions:

- **Short-Route Interventions.** These interventions aim to discourage the agent from acting optimally and taking the shorter option. In Agent-Shortcut levels, the short-route intervention consists of adding the representation of NEVER for $C_A$ to cell state positions along the short path the agent could follow. In Box-Shortcut levels, the short-route intervention consists of adding the representation of NEVER for $C_B$ to cell state positions along the short route the box could be pushed along. This intervention is repeated at every time step.

- **Directional Interventions.** These interventions aim to encourage the agent to act sub-optimally and take the longer option. In Agent-Shortcut levels, the directional intervention consists of adding the appropriate directional representation of $C_A$ to the first square the agent would step onto if it followed the longer path. In Box-Shortcut levels, the directional intervention consists of adding the appropriate directional representation of $C_A$ to the box's initial position to encourage it to be pushed the long route. This intervention is repeated at every time step until the agent either pushes the box off the initial squares (Box-Shortcut interventions) or steps onto the first square of the long-route (Agent-Shortcut Interventions).

The experiments in Section 6.1 that performed these two interventions did not investigate three possible axes of variation that could influence the success rate of interventions. First, the previous experiments simply added the un-scaled vector representations learned by 1x1 probes to the agent's

cell state and did not consider the effect of introducing an intervention strength parameter $\alpha$ to scale representations by before using them for interventions. Second, the previous experiments did not consider varying the directional intervention - specifically, they did not consider whether interventions become more successful if we intervene upon more squares along the longer path. Finally, they did not consider whether the interventions could be successful without the short-route intervention. We now consider the effect of these three factors on the success rate of interventions.

Algorithms 1 and 2 respectively provide pseudoscope for general Agent- and Box-Shortcut interventions in which we (1) intervene on the first $p$ squares of the long route as part of the 'directional' intervention, and (2) introduce an intervention strength $\alpha$. Note that we can also choose not to perform the 'short-route' intervention. The interventions in Section 6.1 correspond to algorithms 1 and 2 with $p$ and $\alpha$ set to 1.

As with the experiments in Section 6.1, all experiments in this section are repeated with 5 independently trained and initialized probes, and interventions are considered a success if they cause the agent to solve the level in the desired, sub-optimal way.

### B.2.1 VARYING THE NUMBER OF DIRECTIONAL INTERVENTIONS

First, we also consider intervening in Agent-Shortcut and Box-Shortcut experiments while varying the number of squares intervened upon as part of the directional intervention along the 'long' route. Specifically, we vary the number of squares intervened upon in 'directional' interventions between 0 squares and 3 squares. When intervening upon an additional square on the 'long route' we intervene on the square following the already-intervened-upon squares. That is, we vary the value of $p$ between 0 and 3 in algorithms 1 and 2. For instance, when we intervene upon two squares in Agent-Shortcut levels, we intervene upon the first two squares the agent would step onto if it followed the longer path. Then, when intervening upon three squares we would additionally intervene on the third square the agent would step onto if it followed the longer path. We also consider varying $\alpha$

Figures 28 and 29 show the success rate when intervening on the agent in Agent-Shortcut and Box-Shortcut levels when varying the intervention strength $\alpha$ and the number of squares intervened upon in the longer route. A few observations can be made from these figures.

First using too high or too low of an $\alpha$ harms the intervention success rate. This would be expected: when $\alpha$ is too low, interventions will not meaningfully change the agent's internal concept representations, while when $\alpha$ is too high, the intervention will cause the agent's internal activations to go off-distribution.

Likewise, performing additional interventions on the long path improves intervention success rate for low $\alpha$, but harms the success rate for high $\alpha$. We posit this is because, to alter the agent's concept representations for a low $\alpha$, additional interventions are useful. However, when $\alpha$ is high, these additional interventions cause the agent's activations to go further off-distribution, impeding the ability of the intervention to steer the agent.

### B.2.2 REMOVING THE SHORT-ROUTE INTERVENTION

We also considered intervening on Agent-Shortcut and Box-Shortcut levels when not performing any short-route interventions and solely performing directional interventions. These directional-only interventions aimed to assess the extent to which the agent's planning mechanism is driven by avoiding planning into squares it represents as having the class NEVER as opposed to extending plans it constructs with the directional concept classes.

Figures 30 and 31 show the success rate when intervening on the agent in Agent-Shortcut and Box-Shortcut levels when varying the intervention strength $\alpha$ and the number of squares intervened upon in on the longer route. Clearly, intervening without the short-route intervention is less successful. However, the reduction in success rate relative to Figures 28 and 29 is significantly more pronounced for Agent-Shortcut interventions than for Box-Shortcut interventions. We thus hypothesize that the agent utilizes a different planning mechanism when planning paths for it to follow as opposed to planning routes to push boxes. Specifically, the agent's path-planning mechanism seems to be more driven by avoiding certain squares (e.g., those it represents using the class NEVER of $C_A$) than the agent's box-route-planning mechanism.

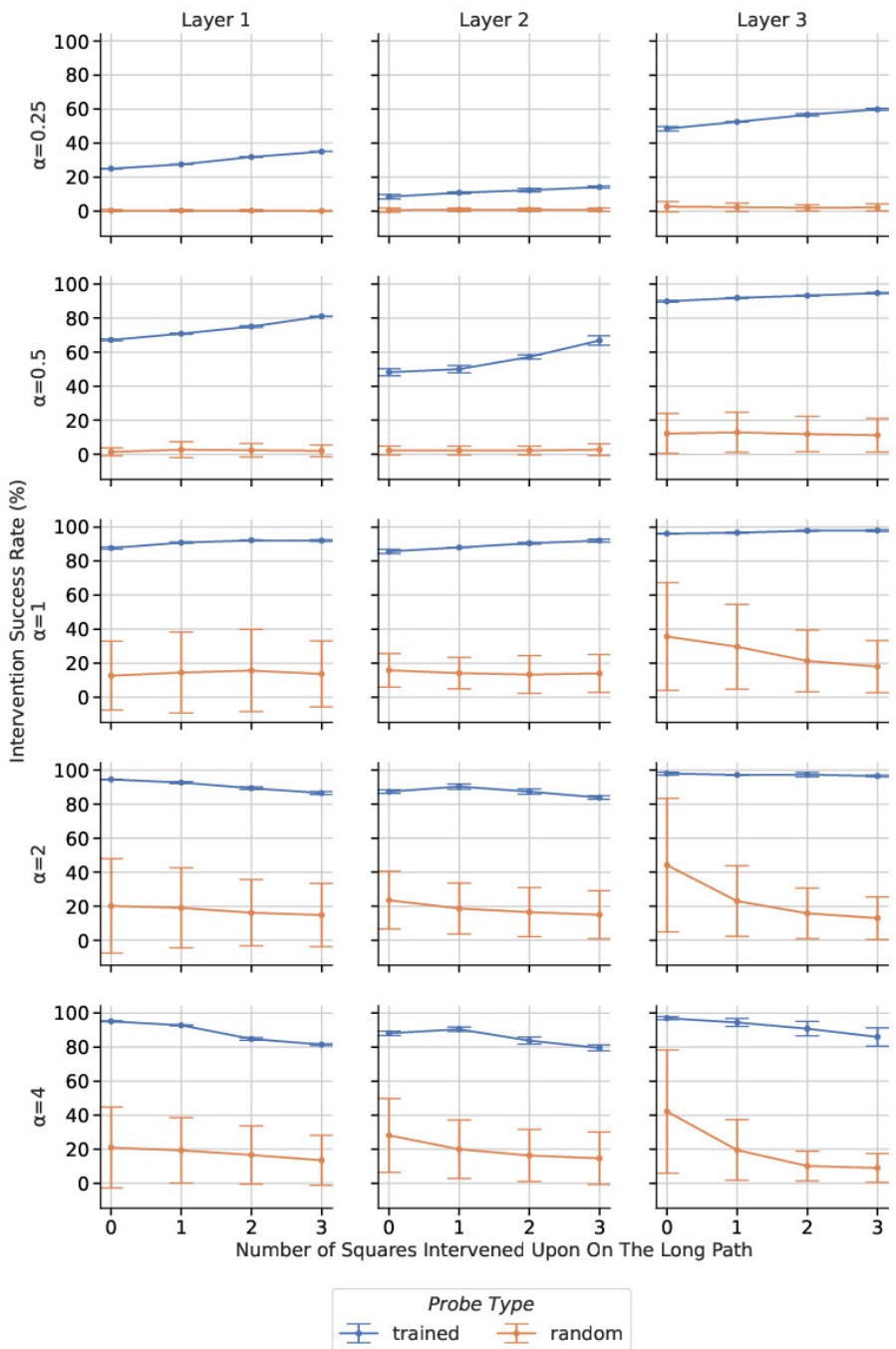

Figure 28: Success rate when intervening in **Agent-Shortcut levels** when varying the number of cell state positions intervened on along the 'long path' during the 'directional' part of the intervention. Interventions are performed using the vector representations of $C_A$ learned by 1x1 probes. Interventions are performed using different intervention strengths $\alpha$ on the agent's cell state at each of its ConvLSTM layers. For each layer, intervention strength, and number of squares intervened upon, we repeat the intervention 5 times using 5 independently trained probes and report the average success rate. We compare interventions performed with trained probes to interventions performed with randomly-initialized probes. Error bars report $\pm 1$ standard deviations.

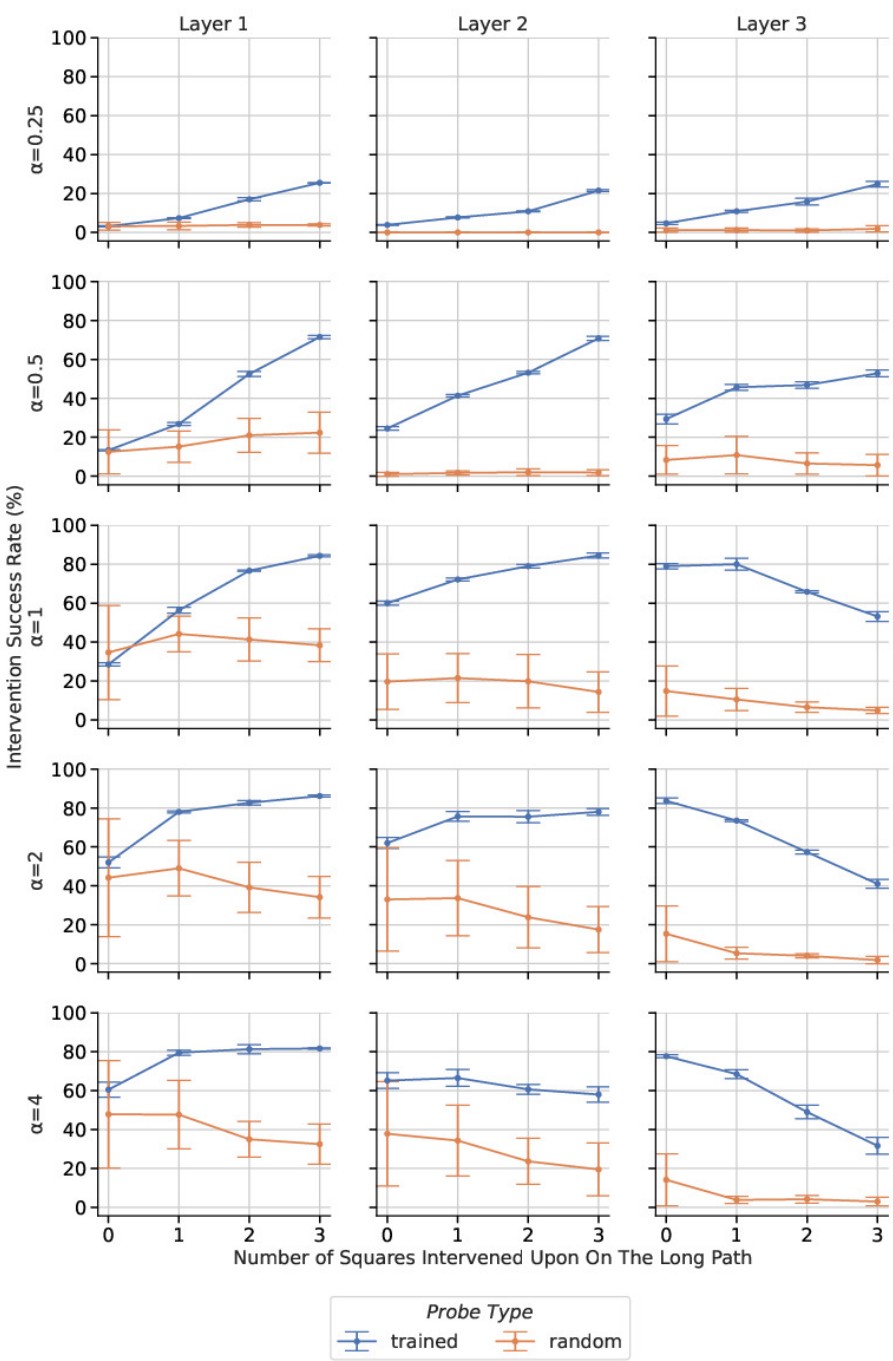

Figure 29: Success rate when intervening in **Box-Shortcut levels** when varying the number of cell state positions intervened on along the 'long path' during the 'directional' part of the intervention. Interventions are performed using the vector representations of $C_B$ learned by 1x1 probes. Interventions are performed using different intervention strengths $\alpha$ on the agent's cell state at each of its ConvLSTM layers. For each layer, intervention strength, and number of squares intervened upon, we repeat the intervention 5 times using 5 independently trained probes and report the average success rate. We compare interventions performed with trained probes to interventions performed with randomly-initialized probes. Error bars report $\pm 1$ standard deviations.

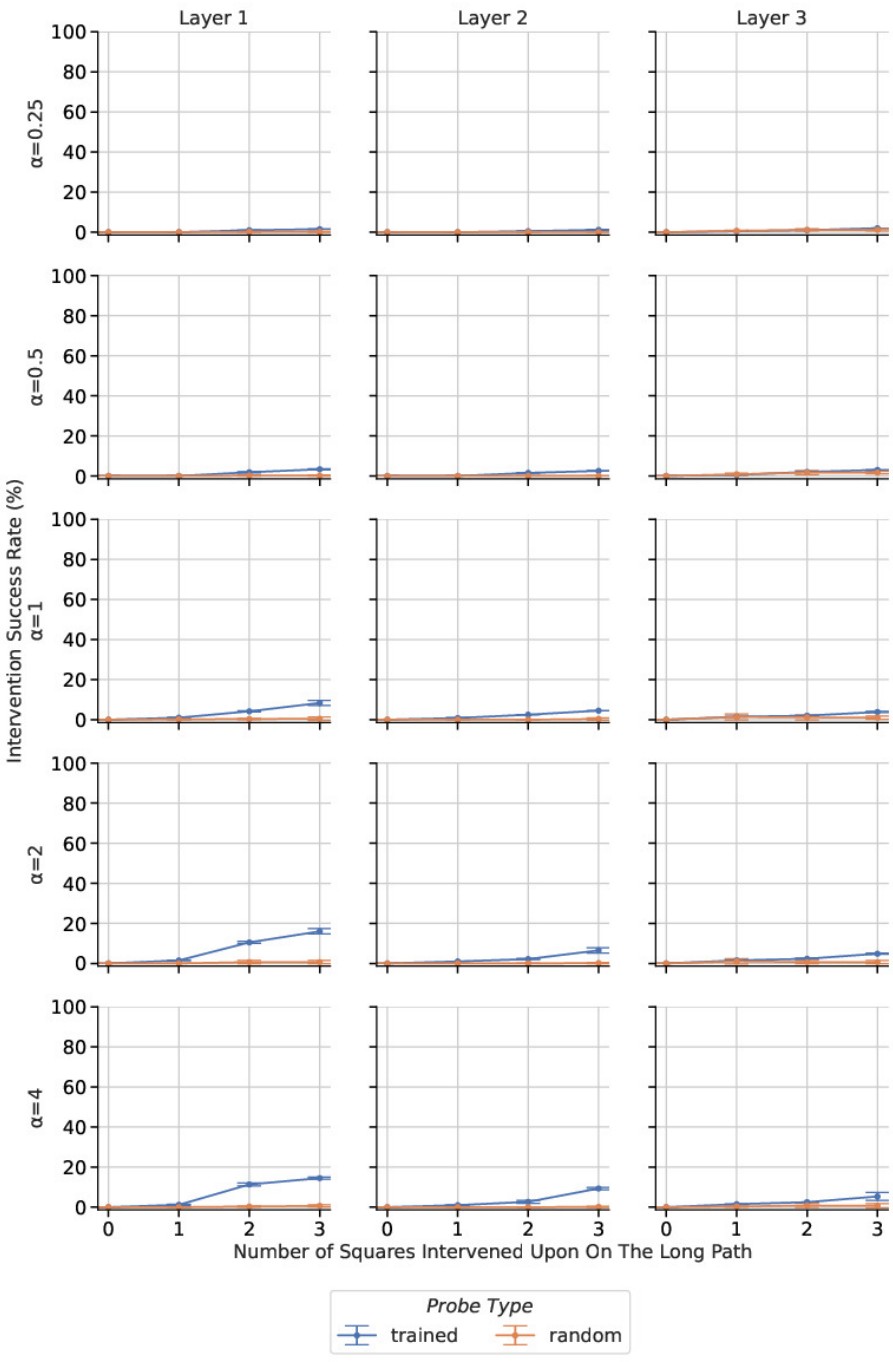

Figure 30: Success rate when intervening in **Agent-Shortcut levels but not performing the 'short-route' part of the intervention**. Identical to Figure 28 otherwise.

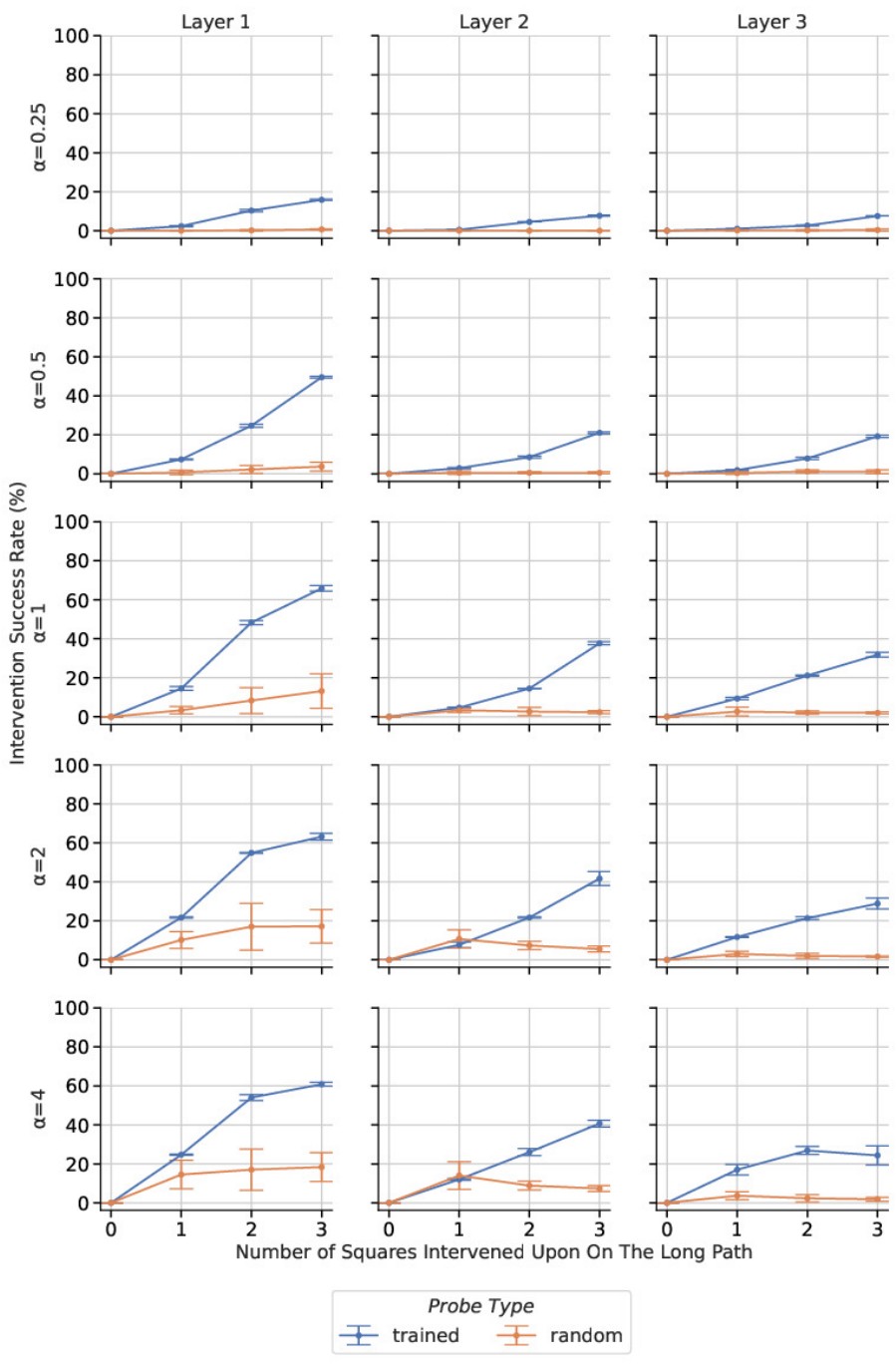

Figure 31: Success rate when intervening in **Box-Shortcut levels, but not performing the 'short-route' part of the intervention**. Identical to Figure 29 otherwise.

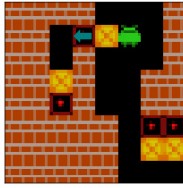

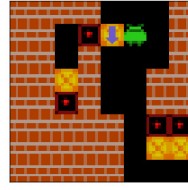

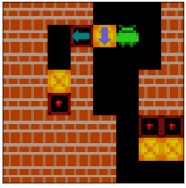

| (a) 'Agent-Only' Intervention | (b) 'Box-Only' Intervention | (c) 'Agent-and-Box' Intervention |

Figure 32: Examples of (a) 'Agent-Only', and (b) 'Box-Only', and (c) 'Agent-and-Box' interventions in a Cutoff level. We add the associated directional representation of $C_A$ to the position with the teal arrow (e.g. in this example the representation of UP for $C_A$). We add the associated directional representation of $C_B$ to the position with the blue arrow (e.g. in this example the representation of RIGHT for $C_B$).

### B.3 ADDITIONAL INTERVENTION EXPERIMENTS: INTERVENING IN A NEW SET OF LEVELS TO ENCOURAGE OPTIMAL BEHAVIOR

We also considered performing interventions in a different set of levels. We call these levels 'Cutoff' levels. Cutoff levels follow a schematic very similar to the levels introduced in Appendix A.3.2 and are designed in such a way that solving them requires the agent to forsee the long-run consequences of its actions. Specifically, these are levels in which a box is adjacent to a target at the entrance of a corridor. The agent always begins levels one square removed from this box. At the end of this (variable-length) corridor is another box adjacent to another target. To solve these levels, the agent must not act myopically – i.e. it must not immediately push the box at the corridor entrance onto the adjacent target as doing so would block the corridor and make the level unsolvable – and must instead push the box out of the way so that it can enter the corridor and reach the box at the corridor end.

We use a dataset of 200 Cutoff levels. These were created by designing 25 Cutoff levels (with corridors of varying lengths), and then making 8 copies of each by applying vertical reflection and 90°, 180°, and 270° rotations. Figure 33 shows the percent of these levels the agent solves when given varying number of 'thinking steps'. Recall that 'thinking steps' refer to steps at the start of episodes where the agent is forced to remain stationary prior to acting. By default – that is, without any thinking time – the agent solves none of the 200 Cutoff levels. However, the agent can solve the vast majority of Cutoff levels when given additional test-time compute to refine its plans. Thus, we consider intervening in Cutoff levels with the aim of artificially replicating the effect of additional test-time compute. That is, we perform interventions that aim to aid the agent in solving the

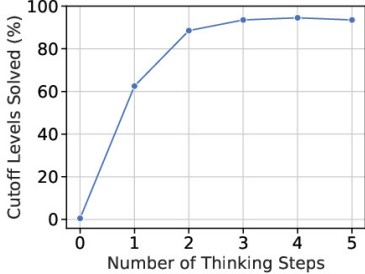

Figure 33: The percentage of 200 Cutoff levels the fully-trained agent solves when performing different numbers of 'thinking steps' prior to acting.

levels without additional decision-time compute. We investigate these interventions in order to determine whether we can intervene on the agent's internal plan to aid it in forming and executing *optimal* plans. This is in contrast to Section 6.1 in which we focused on intervening on the agent to encourage it to form and execute *sub-optimal* plans.

We perform three types of interventions in Cutoff levels: 'Agent-and-Box' interventions, 'Agent-Only' interventions, and 'Box-Only' interventions. 'Agent-Only' interventions consist of adding an appropriate directional representation of $C_A$ to the target at the entrance of the corridor that corresponds to the agent stepping into the corridor. An example is shown in Figure 32a. The motivation for this intervention is that it should correspond to the agent having extended its plan back fully from the end of the corridor such that the agent knows it should not block the corridor off. If this is the case, the agent should then plan to not myopically push the box onto the target at the corridor entrance. 'Box-Only' interventions consist of adding an appropriate directional representation of

$C_B$ to the initial position of the box that could be myopically pushed into the corridor entrance to encourage the agent to instead push this box out of the way of the corridor entrance. An example is shown in Figure 32b. The motivation for this intervention is that it should remove the need for the agent to complete planning backwards from the end of the corridor. Finally, 'Agent-and-Box' interventions consist of the conjunction of 'Agent-Only' and 'Box-Only' interventions. An example is shown in Figure 32c.

In all cases, the interventions are performed by adding the corresponding representations learned by 1x1 probes to the agent's cell state at one of its ConvLSTM layers. The interventions are repeated at each computational tick until the agent moves the box at the corridor entrance at which point the interventions cease. As previously, we repeat all interventions with 5 trained and 5 randomly initialized probes. Since the agent cannot solve any Cutoff levels without steps of 'thinking time', we consider an intervention to be a success if it leads to the agent solving the level.

Figures 34a, 34b and 34c respectively show success rates when performing 'Agent-Only', 'Box-Only' and 'Agent-and-Box' interventions on the agent's cell state at each layer using different intervention strengths $\alpha$. A few key takeaways can be drawn from these results. First, and most importantly, these interventions can replicate the effect of additional test-time compute and allow the agent to solve levels it would otherwise fail. That the concept representations of $C_A$ and $C_B$ decoded by our 1x1 probes can be used to intervene on the agent in such a way that induces a behavioral effect comparable to that of additional test-time compute is further evidence that the plans decoded by the probes causally influence the agent's behavior in the way that would be expected.

Another interesting takeaway from these results is that the pattern of success rates across layers is notably different for 'Agent-Only' (Figure 34a) and 'Box-Only' (Figure 34b) interventions. This suggests potential insights into the role of each ConvLSTM in the planning process. For instance, note that interventions on layer 2 using representations from trained probes are highly successful when performing 'Agent-Only' interventions but are no better than baseline interventions with random probes when performing 'Box-Only' interventions. This suggests the agent performs some type of 'plan conflict detection' computation either at layer 2, or between layers 2 and 3, such that intervening on layer 2 to encourage the agent to plan to step onto the target causes the agent to detect that acting myopically would conflict with its plans. Likewise, 'Box-Only' interventions are much more successful than 'Agent-Only' interventions at layer 3. This indicates that agent does not perform 'plan conflict detection' computations at layer 3 – since, if it did so we would expect 'Agent-Only' interventions to be successful – but does update the routes it plans to push boxes at this layer.

## C  ADDITIONAL TRAINING-TIME INTERPRETABILITY RESULTS

In Section 6.2 we briefly investigated the emergence of planning-relevant representations during training. Precisely, we demonstrated the co-occurrence during training of (i) the agent's internal representations of $C_A$ and $C_B$ in its final layer cell state, and (ii) the ability of the agent to perform better when given additional test-time compute. In this section, we now detail experiments in which we further interpret the agent during the early stages of training. Specifically, we show that:

- The agent's internal representations of $C_A$ and $C_B$ largely emerge at the start of training (Appendix C.1).

- The agent's ability to refine its internal plans when given additional test-time compute emerges early on in training (Appendix C.2).

- As shown for the final layer in Section 6.2, the emergence of the agent's representations of $C_A$ and $C_B$ in its cell state at *all* layers coincides with the emergence of its ability to perform better when given additional test-time compute (Appendix C.3).

- The emergence during training of the agent's ability to improve its internal plans when given extra test-time compute coincides with the emergence of its ability to perform better when given this extra compute (Appendix C.4).

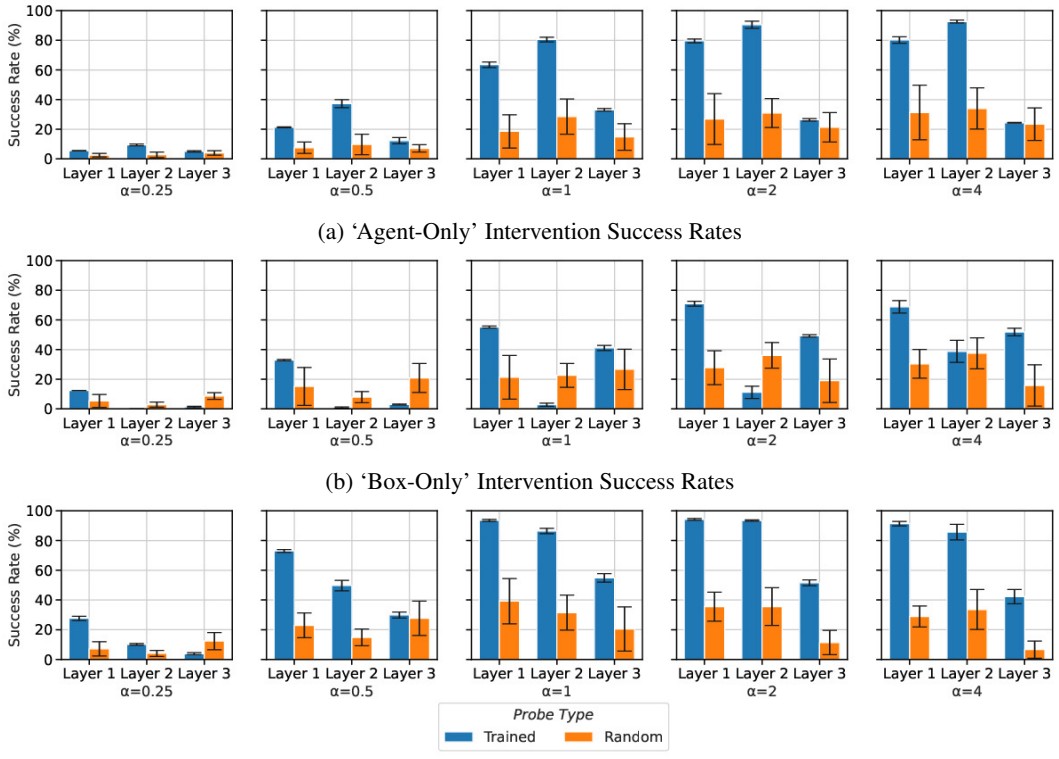

(a) 'Agent-Only' Intervention Success Rates

(b) 'Box-Only' Intervention Success Rates

(c) 'Agent-and-Box' Intervention Success Rates

Figure 34: Success rate when intervening on the agent in Cutoff levels. We consider 'agent-only', 'box-only' and 'agent-and-box interventions'. These interventions are respectively performed using the vector representations of $C_A$, $C_B$, and both $C_A$ and $C_B$, as learned by 1x1 probes. Interventions are performed using different intervention strengths $\alpha$ on the agent's cell state at each of its Con-vLSTM layers. For each layer and intervention strength, we repeat the intervention 5 times using 5 probes and report the average success rate. We compare interventions performed with trained probes to interventions performed with randomly-initialized probes. Error bars report $\pm 1$ standard deviation.

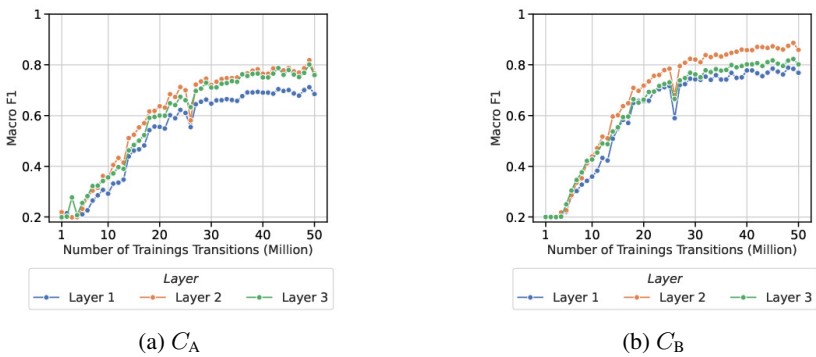

(a) $C_A$          (b) $C_B$

Figure 35: Macro F1 scores achieved when training 1x1 probes to predict (a) $C_A$ and (b) $C_B$ using the agent's cell state activations at each layer at checkpoints every 1 million transitions over the first 50 million transitions of training.

### C.1    Investigating the Emergence of Planning-Relevant Concept Representations During Training

First, we investigate the emergence of the planning-relevant concepts we study – $C_A$ and $C_B$ – over the course of training. We do this by taking checkpoints of the agent every 1 million transitions over the first 50 million (of 250 million total) transitions of training. For each checkpoint, we train 1x1 probes to predict $C_A$ and $C_B$ using the agent's cell state activations *at that checkpoint*. That is, we train and test probes on datasets generated by collecting the agent's cell state activations when running the agent *at that checkpoint*. Note that re-training probes at each checkpoint is important since $C_A$ and $C_B$ are behavior-dependent concepts in that the classes they assign to squares depends on how the agent behaves (which itself changes during training as the agent's parameters update).

Figure 35 shows the macro F1 scores achieved when training probes to predict $C_A$ and $C_B$ based on the agent's cell state activations over the course of the first 50 million transitions of training. Clearly, the agent's internal representations of these concepts largely emerge early on in training. However, we note that, even after 50 million transitions, the macro F1s achieved by our probes are still somewhat lower than the macro F1s achieved when probing the fully-trained agent as shown in Figure 4. This suggests that the agent does slightly improve its representations of these concepts over the entire course of training.

### C.2    Investigating The Emergence of Test-Time Plan Refinement During Training

In Section 5, we used Figure 6 to demonstrate that the agent can use additional test-time compute at the start of episodes to refine its internal plan. Thus, a natural question to ask is *when* during training does this ability emerge - is this an ability that emerges early on in training and that is gradually improved, or is it an ability that the agent only develops towards the end of training?

We can investigate the emergence of the ability to refine plans when given additional test-time compute by repeating the setup from Figure 6 – i.e. predicting the agent's future actions using its internal plans decoded from its cell state by a 1x1 probe during steps of 'thinking time' whilst the agent is forced to remain stationary prior to moving – but now whilst doing so for checkpoints of the agent taken throughout training.

Specifically, we now use the 1x1 probes from Appendix C.1 to decode the agent's plan when given additional test-time compute. As before, we use checkpoints of the agent taken every 1 million transitions during the first 50 million transitions of training. We force the agent at each checkpoint to perform 5 'thinking steps' prior to acting at the start of 1000 episodes and use 1x1 probes to decode the agents internal representations of $C_A$ and $C_B$ at each of the 15 corresponding internal computational ticks. We view the predictions made by our probes at each tick as being the agent's internal plan at each tick. We then measure the average correctness of the agent's plan at each of the 15 ticks by averaging the macro F1 of the probe's predictions at each tick. Finally, we measure the extent to which the agent can utilize the extra test-time compute afforded by 'thinking time' by measuring the increase in average macro F1 when using the probe's predictions at the 15th tick relative to at the 1st tick.

Figure 36 shows the increase in macro F1 for different checkpoints of the agent when using probe predictions from the 15th tick of thinking time relative to the 1st tick of thinking time to predict $C_A$ for each checkpoint. Figure 37 shows the analogous results when predicting $C_B$ for each checkpoint. Clearly, the agent acquires the ability to use additional test-time compute to refine its 'internal plan' (i.e. its internal representations of $C_A$ and $C_B$) early on in training.

### C.3    Investigating the Co-Emergence of Planning-Relevant Concept Representations and Planning-Like Behavior During Training

In Section 6.2 we illustrated that (i) the agent's internal representations of $C_A$ and $C_B$ in its final layer cell state and (ii) the ability of the agent to perform better when given additional test-time compute emerge concurrently during training. This naturally leads to the question of whether the emergence of this type of planning-like behavior coincides with the emergence of the agent's representations of $C_A$ and $C_B$ at all layers. We now provide evidence that this is indeed the case.

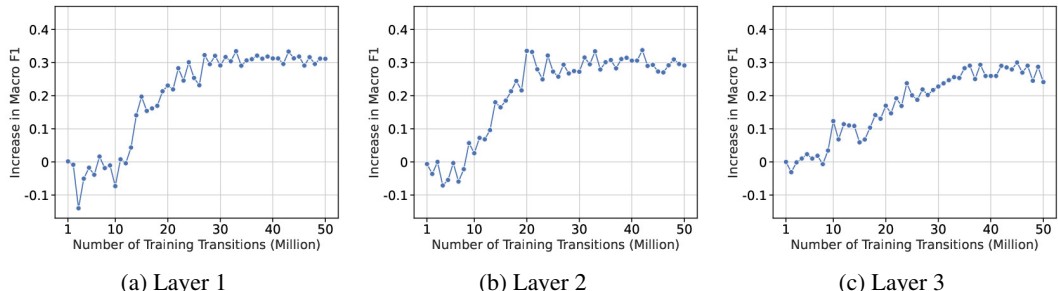

|     |     |     |
| --- | --- | --- |
| (a) Layer 1 | (b) Layer 2 | (c) Layer 3 |

Figure 36: Increase in macro F1 when predicting the **agent's future movements** ($C_A$) using the agent's internal plan at each layer as decoded at the 1st and 15th tick. Each data point corresponds to the increase in macro F1 between the first and last tick performed during 5 thinking steps for a checkpoint of the agent taken every million transitions during the first 50 million transitions of training. Here, the 'internal plan' at a layer for a tick corresponds to the agent's representation of $C_A$ as decoded by a 1x1 probe applied to the agent's cell state at that layer at that tick.

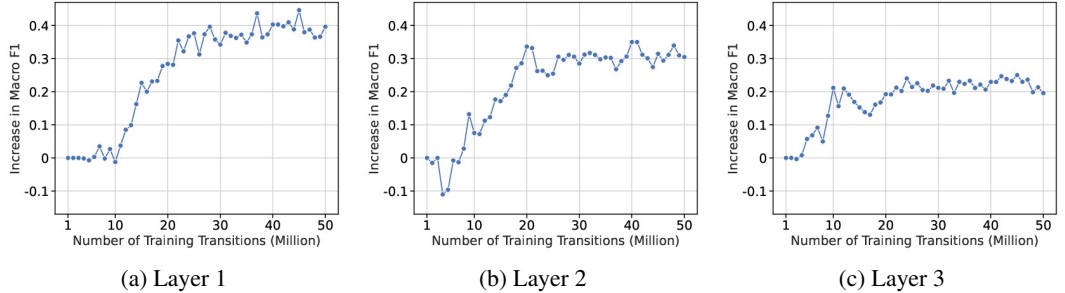

|     |     |     |
| --- | --- | --- |
| (a) Layer 1 | (b) Layer 2 | (c) Layer 3 |

Figure 37: Increase in macro F1 when **predicting future box movements** ($C_B$) using the agent's internal plan at each layer as decoded at the 1st and 15th tick. Each data point corresponds to the increase in macro F1 between the first and last tick performed during 5 thinking steps for a checkpoint of the agent taken every million transitions during the first 50 million transitions of training. Here, the 'internal plan' at a layer for a tick corresponds to the agent's representation of $C_B$ as decoded by a 1x1 probe applied to the agent's cell state at that layer at that tick.

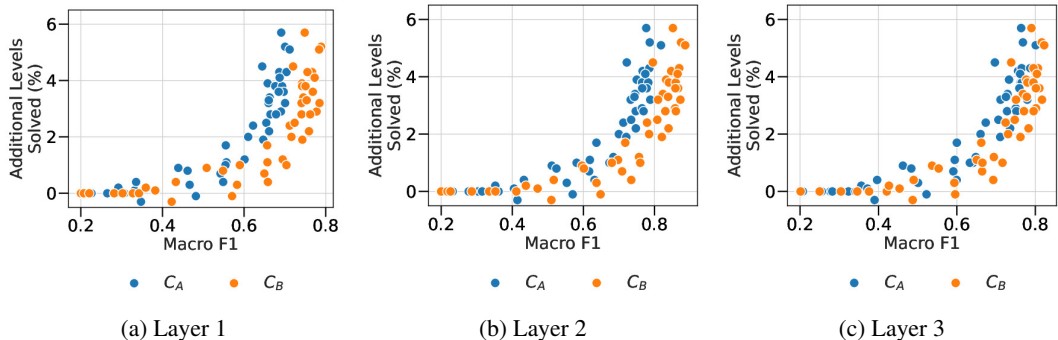

(a) Layer 1            (b) Layer 2            (c) Layer 3

Figure 38: The relationship between (i) the percentage of extra medium levels solved when an agent is given 5 steps to 'think', and (ii) macro F1 score of probes when predicting $C_A$ (blue) and $C_B$ (orange) from the agent's cell state at each layer at different checkpoints taken during training. Each point corresponds to these quantities calculated for a single checkpoint.

As in Section 6.2 we do this by inspecting both (i) the macro F1 of probes trained to predict $C_A$ and $C_B$ using the agent's cell state activations at all layers, and (ii) benefit from additional test-time compute. We measure these quantities at checkpoints of the agent taken ever 1 million transitions over the first 50 million transitions of training. As in Section 6.2, we measure the ability of the agent to benefit from additional test-time compute by counting the number of 1000 medium-difficult levels from the Boxoban dataset (Guez et al., 2018a) the agent cannot solve by default, but can solve when given 5 'thinking steps'.

The effect illustrated in Figure 9 – namely, that the period of training in which the agent begins to solve additional levels when given extra compute is the same as the period in which its representations of $C_A$ and $C_B$ become increasingly well-developed – can be seen across all layers. This can be seen in Figure 38 in which we plot the relationship between (i) the percentage of additional medium levels solved and (ii) the macro F1 when training probes to predict $C_A$ and $C_B$ using the agent's cell state activation *at each layer* for that checkpoint. As would be expected if the agent's representations of $C_A$ and $C_B$ were used as part of an internal planning process, the emergence of these representations at all layers is clearly related to the emergence of the agent's ability to benefit from extra test-time compute.

### C.4 INVESTIGATING THE CO-EMERGENCE OF TEST-TIME PLAN REFINEMENT AND PLANNING-LIKE BEHAVIOR DURING TRAINING

Yet, Figure 38 only shows that the emergence of the agent's representations of $C_A$ and $C_B$ is related to the emergence of the agent's planning-like behavior. It hence does not show that the planning process the agent uses these representations as a part of is related to the behavioral evidence of planning exhibited by the agent.

However, recall that in Appendix C.2 we showed that we could inspect the emergence of the agent's internal planning process by inspecting the manner in which the plans the agent internally represents in terms of $C_A$ and $C_B$ become more accurate when given 'thinking steps'. Specifically, we can use probes to decode the agent's internal plan at the 1st and final tick of additional compute given by 5 steps of 'thinking time' and measure the correctness of these plans using the macro F1 achieved when viewing these plans as predictions of the agent's future behavior ($C_A$) and its effect on the environment ($C_B$). We can then measure the extent to which the agent is internally using these representations as part of an internal planning process by measuring the *increase* in macro F1 achieved when using the agent's plans after the first tick and after the final tick of 'thinking time'. If the agent uses its representations of $C_A$ and $C_B$ as part of an internal planning process, the agent ought to be able to use the extra compute associated with 'thinking steps' to form a better plan with these representations. That is, if these representations are used for planning, we expect the macro F1 of the agent's internal plan to increase during 'thinking steps'.

Figure 39 shows, for checkpoints of the agent taken over the first 50 million transitions of training, the relationship between (i) the percentage of addition levels the agent solves with 5 'thinking steps',

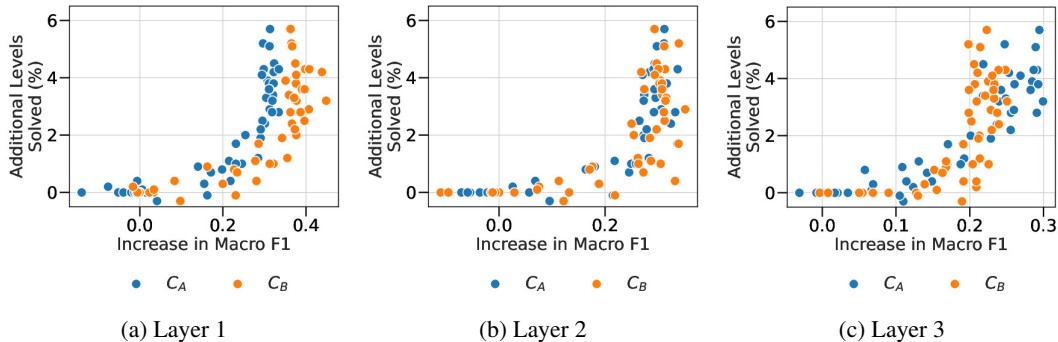

(a) Layer 1          (b) Layer 2          (c) Layer 3

Figure 39: The relationship between (i) the percentage of extra medium levels solved when an agent is given 5 steps to 'think', and (ii) the increase in macro F1 when predicting $C_A$ (blue) and $C_B$ (orange) from the agent's cell state at each layer between the 1st and 15th computational tick of 5 'thinking steps' at different checkpoints taken during training. Each point corresponds to these quantities calculated for a single checkpoint.

and (ii) the increase in macro F1 when using the agent's internal plan at the 1st and 15th tick of 5 steps of 'thinking time' to predict future agent movements ($C_A$) and box movements ($C_B$). As would be expected if the agent used its representations of $C_A$ and $C_B$ for planning, checkpoints at which the agent can use extra test-time compute to improve its internal plans are also checkpoints at which the agent solves more levels when given extra compute.

## D    ADDITIONAL PROBING RESULTS

Our methodology is underpinned by our use of linear probes. In this section, we now provide further details regarding the probes we use and further relevant experimental results. This section is organized as follows:

- Appendix D.1 provides additional details regarding how we train probes.
- Appendix D.2 provides additional metrics – specifically, class-specific recalls, precisions and F1s – for the 1x1 and 3x3 probes we discuss from Section 4.2.
- Appendix D.3 details the macro F1 scores achieved when using larger probes.
- Appendix D.4 outlines the performance of 1x1 and 3x3 probes trained to predict alternate square-level concepts to $C_A$ and $C_B$.
- Appendix D.5 provides the results of applying 'global' probes that receive the agent's entire cell state as input to predict the agent's future actions directly.

### D.1    PROBE TRAINING DETAILS

In this section, we now provide a brief overview of the manner in which the probes considered in this paper are trained. All probes are trained for 10 epochs using the AdamW optimizer (Loshchilov & Hutter, 2019) and implemented as convolutions with a batch size of 16, learning rate of 0.001 and a weight decay of 0.001. We train probes to predict the concepts assigned to Sokoban squares using the agent's cell state activations both after training, and at checkpoints taken during training. As the concepts we study depend on the agent's behavior and thus the agent's parameters, we train and test probes on different datasets when investigating the agent at different points of training.

All probes trained to predict concept classes using the fully-trained agent's cell states are trained and tested on datasets consisting of 106.6k and 25.7k transitions. The training dataset is generated by running the fully-trained agent for 3000 episodes on levels from the Boxoban unfiltered training dataset (Guez et al., 2018a). The test dataset is generated by running the fully-trained agent for 1000 episodes on levels drawn from Boxoban unfiltered validation dataset.

All probes trained to decode concepts from the agent's cell state at checkpoints taken during training are trained and tested on checkpoint-specific datasets. For any checkpoint, the training and test

| Class | Metric | Layer 1 | Layer 2 | Layer 3 | Baseline |
|---|---|---|---|---|---|
| NEVER | F1 | $0.9787 \pm 0.0000$ | $0.9821 \pm 0.0001$ | $0.9845 \pm 0.0000$ | $0.9049 \pm 0.0000$ |
| | Precision | $0.9795 \pm 0.0002$ | $0.9838 \pm 0.0004$ | $0.9826 \pm 0.0001$ | $0.8279 \pm 0.0000$ |
| | Recall | $0.9779 \pm 0.0001$ | $0.9804 \pm 0.0003$ | $0.9864 \pm 0.0001$ | $0.9978 \pm 0.0000$ |
| UP | F1 | $0.7563 \pm 0.0004$ | $0.8187 \pm 0.0004$ | $0.8396 \pm 0.0003$ | $0.0000 \pm 0.0000$ |
| | Precision | $0.7405 \pm 0.0026$ | $0.8238 \pm 0.0011$ | $0.8237 \pm 0.0019$ | $1.0000 \pm 0.0000$ |
| | Recall | $0.7728 \pm 0.0025$ | $0.8137 \pm 0.0008$ | $0.8561 \pm 0.0019$ | $0.0000 \pm 0.0000$ |
| DOWN | F1 | $0.7548 \pm 0.0003$ | $0.8278 \pm 0.0007$ | $0.8255 \pm 0.0004$ | $0.0000 \pm 0.0000$ |
| | Precision | $0.7516 \pm 0.0061$ | $0.8216 \pm 0.0039$ | $0.8440 \pm 0.0021$ | $1.0000 \pm 0.0000$ |
| | Recall | $0.7581 \pm 0.0066$ | $0.8342 \pm 0.0026$ | $0.8078 \pm 0.0018$ | $0.0000 \pm 0.0000$ |
| LEFT | F1 | $0.7985 \pm 0.0004$ | $0.8282 \pm 0.0002$ | $0.7955 \pm 0.0005$ | $0.0000 \pm 0.0000$ |
| | Precision | $0.7958 \pm 0.0019$ | $0.8126 \pm 0.0020$ | $0.8199 \pm 0.0019$ | $1.0000 \pm 0.0000$ |
| | Recall | $0.8013 \pm 0.0021$ | $0.8445 \pm 0.0025$ | $0.7726 \pm 0.0010$ | $0.0000 \pm 0.0000$ |
| RIGHT | F1 | $0.7238 \pm 0.0007$ | $0.8228 \pm 0.0003$ | $0.8129 \pm 0.0002$ | $0.2262 \pm 0.0000$ |
| | Precision | $0.7362 \pm 0.0029$ | $0.8181 \pm 0.0029$ | $0.8131 \pm 0.0010$ | $0.2707 \pm 0.0000$ |
| | Recall | $0.7119 \pm 0.0035$ | $0.8277 \pm 0.0033$ | $0.8128 \pm 0.0011$ | $0.1943 \pm 0.0000$ |

Table 2: Average and standard deviation of class-specific performance metrics when probing for $C_A$ using 1x1 probes.

datasets are created by collecting transitions when running the agent at that checkpoint for 1000 and 500 episodes in the unfiltered training and unfiltered validation Boxoban datasets respectively. These datasets are of different sizes for each checkpoint as the agent's behavior changes over the course of training.

## D.2 ADDITIONAL PROBING METRICS

In Section 4, we investigated training 1x1 and 3x3 probes to predict $C_A$ and $C_B$. Figure 4 illustrated the macro F1s achieved by these probes. We now provide additional metrics for these probes. Specifically, we provide, for each class, the precision, recall, and F1 achieved by our probes when viewing that class as the positive class and all other classes as belonging to a single negative class. Since we train 5 probes with different initialization seeds, we report both the mean and standard deviation for all metrics. Tables 2 and 3 respectively show these metrics for 1x1 and 3x3 probes trained to predict $C_A$. Tables 4 and 5 respectively show these metrics for 1x1 and 3x3 probes trained to predict $C_B$.

| Class | Metric | Layer 1 | Layer 2 | Layer 3 | Baseline |
|---|---|---|---|---|---|
| NEVER | F1 | $0.9862 \pm 0.0001$ | $0.9874 \pm 0.0002$ | $0.9877 \pm 0.0003$ | $0.9294 \pm 0.0005$ |
| | Precision | $0.9865 \pm 0.0003$ | $0.9882 \pm 0.0004$ | $0.9877 \pm 0.0003$ | $0.8887 \pm 0.0015$ |
| | Recall | $0.9860 \pm 0.0001$ | $0.9866 \pm 0.0002$ | $0.9878 \pm 0.0005$ | $0.9739 \pm 0.0008$ |
| UP | F1 | $0.8614 \pm 0.0006$ | $0.8614 \pm 0.0004$ | $0.8735 \pm 0.0013$ | $0.3300 \pm 0.0110$ |
| | Precision | $0.8619 \pm 0.0046$ | $0.8568 \pm 0.0024$ | $0.8669 \pm 0.0040$ | $0.4682 \pm 0.0092$ |
| | Recall | $0.8610 \pm 0.0037$ | $0.8660 \pm 0.0020$ | $0.8802 \pm 0.0020$ | $0.2553 \pm 0.0151$ |
| DOWN | F1 | $0.8616 \pm 0.0006$ | $0.8677 \pm 0.0006$ | $0.8734 \pm 0.0005$ | $0.3464 \pm 0.0082$ |
| | Precision | $0.8656 \pm 0.0030$ | $0.8662 \pm 0.0035$ | $0.8791 \pm 0.0033$ | $0.4297 \pm 0.0085$ |
| | Recall | $0.8576 \pm 0.0019$ | $0.8691 \pm 0.0026$ | $0.8678 \pm 0.0026$ | $0.2907 \pm 0.0151$ |
| LEFT | F1 | $0.8606 \pm 0.0005$ | $0.8654 \pm 0.0002$ | $0.8726 \pm 0.0003$ | $0.3528 \pm 0.0048$ |
| | Precision | $0.8570 \pm 0.0034$ | $0.8579 \pm 0.0026$ | $0.8741 \pm 0.0017$ | $0.4541 \pm 0.0026$ |
| | Recall | $0.8641 \pm 0.0029$ | $0.8731 \pm 0.0028$ | $0.8710 \pm 0.0016$ | $0.2885 \pm 0.0070$ |
| RIGHT | F1 | $0.8536 \pm 0.0005$ | $0.8626 \pm 0.0013$ | $0.8713 \pm 0.0015$ | $0.3524 \pm 0.0085$ |
| | Precision | $0.8491 \pm 0.0031$ | $0.8653 \pm 0.0046$ | $0.8722 \pm 0.0052$ | $0.4631 \pm 0.0091$ |
| | Recall | $0.8581 \pm 0.0026$ | $0.8600 \pm 0.0024$ | $0.8704 \pm 0.0039$ | $0.2848 \pm 0.0136$ |

Table 3: Average and standard deviation of class-specific performance metrics when probing for $C_A$ using 3x3 probes.

| Class | Metric | Layer 1 | Layer 2 | Layer 3 | Baseline |
|---|---|---|---|---|---|
| NEVER | F1 | $0.9907 \pm 0.0000$ | $0.9943 \pm 0.0000$ | $0.9913 \pm 0.0000$ | $0.9634 \pm 0.0000$ |
| | Precision | $0.9906 \pm 0.0002$ | $0.9943 \pm 0.0001$ | $0.9913 \pm 0.0001$ | $0.9311 \pm 0.0000$ |
| | Recall | $0.9909 \pm 0.0002$ | $0.9942 \pm 0.0001$ | $0.9912 \pm 0.0001$ | $0.9980 \pm 0.0000$ |
| UP | F1 | $0.8027 \pm 0.0008$ | $0.9126 \pm 0.0002$ | $0.8805 \pm 0.0002$ | $0.0000 \pm 0.0000$ |
| | Precision | $0.7968 \pm 0.0039$ | $0.9054 \pm 0.0016$ | $0.8562 \pm 0.0008$ | $1.0000 \pm 0.0000$ |
| | Recall | $0.8089 \pm 0.0041$ | $0.9200 \pm 0.0017$ | $0.9061 \pm 0.0013$ | $0.0000 \pm 0.0000$ |
| DOWN | F1 | $0.8386 \pm 0.0003$ | $0.9257 \pm 0.0003$ | $0.8590 \pm 0.0004$ | $0.0000 \pm 0.0000$ |
| | Precision | $0.8407 \pm 0.0031$ | $0.9327 \pm 0.0010$ | $0.8741 \pm 0.0024$ | $1.0000 \pm 0.0000$ |
| | Recall | $0.8366 \pm 0.0032$ | $0.9188 \pm 0.0014$ | $0.8444 \pm 0.0015$ | $0.0000 \pm 0.0000$ |
| LEFT | F1 | $0.8954 \pm 0.0001$ | $0.9247 \pm 0.0003$ | $0.8058 \pm 0.0007$ | $0.0000 \pm 0.0000$ |
| | Precision | $0.8968 \pm 0.0019$ | $0.9194 \pm 0.0010$ | $0.8144 \pm 0.0008$ | $1.0000 \pm 0.0000$ |
| | Recall | $0.8940 \pm 0.0018$ | $0.9301 \pm 0.0016$ | $0.7975 \pm 0.0012$ | $0.0000 \pm 0.0000$ |
| RIGHT | F1 | $0.8111 \pm 0.0008$ | $0.9162 \pm 0.0005$ | $0.8315 \pm 0.0005$ | $0.3079 \pm 0.0000$ |
| | Precision | $0.8182 \pm 0.0020$ | $0.9194 \pm 0.0008$ | $0.8321 \pm 0.0016$ | $0.2707 \pm 0.0000$ |
| | Recall | $0.8041 \pm 0.0034$ | $0.9131 \pm 0.0013$ | $0.8310 \pm 0.0016$ | $0.3570 \pm 0.0000$ |

Table 4: Average and standard deviation of class-specific performance metrics when probing for $C_B$ using 1x1 probes.

| Class | Metric | Layer 1 | Layer 2 | Layer 3 | Baseline |
|---|---|---|---|---|---|
| NEVER | F1 | $0.9950 \pm 0.0000$ | $0.9956 \pm 0.0001$ | $0.9949 \pm 0.0000$ | $0.9664 \pm 0.0001$ |
| | Precision | $0.9952 \pm 0.0002$ | $0.9962 \pm 0.0001$ | $0.9949 \pm 0.0002$ | $0.9436 \pm 0.0003$ |
| | Recall | $0.9948 \pm 0.0002$ | $0.9951 \pm 0.0001$ | $0.9949 \pm 0.0002$ | $0.9904 \pm 0.0004$ |
| UP | F1 | $0.9201 \pm 0.0005$ | $0.9313 \pm 0.0008$ | $0.9230 \pm 0.0008$ | $0.3976 \pm 0.0108$ |
| | Precision | $0.9127 \pm 0.0016$ | $0.9219 \pm 0.0007$ | $0.9160 \pm 0.0021$ | $0.5518 \pm 0.0121$ |
| | Recall | $0.9277 \pm 0.0012$ | $0.9409 \pm 0.0012$ | $0.9302 \pm 0.0014$ | $0.3112 \pm 0.0166$ |
| DOWN | F1 | $0.9312 \pm 0.0003$ | $0.9347 \pm 0.0015$ | $0.9292 \pm 0.0003$ | $0.4246 \pm 0.0074$ |
| | Precision | $0.9376 \pm 0.0027$ | $0.9360 \pm 0.0038$ | $0.9313 \pm 0.0017$ | $0.5733 \pm 0.0153$ |
| | Recall | $0.9248 \pm 0.0030$ | $0.9334 \pm 0.0022$ | $0.9272 \pm 0.0014$ | $0.3376 \pm 0.0144$ |
| LEFT | F1 | $0.9208 \pm 0.0006$ | $0.9346 \pm 0.0006$ | $0.9160 \pm 0.0007$ | $0.4068 \pm 0.0039$ |
| | Precision | $0.9182 \pm 0.0020$ | $0.9246 \pm 0.0021$ | $0.9196 \pm 0.0026$ | $0.5884 \pm 0.0054$ |
| | Recall | $0.9233 \pm 0.0019$ | $0.9448 \pm 0.0019$ | $0.9125 \pm 0.0038$ | $0.3109 \pm 0.0059$ |
| RIGHT | F1 | $0.9179 \pm 0.0005$ | $0.9384 \pm 0.0007$ | $0.9317 \pm 0.0004$ | $0.4190 \pm 0.0053$ |
| | Precision | $0.9153 \pm 0.0052$ | $0.9378 \pm 0.0020$ | $0.9338 \pm 0.0030$ | $0.5819 \pm 0.0125$ |
| | Recall | $0.9206 \pm 0.0044$ | $0.9391 \pm 0.0011$ | $0.9297 \pm 0.0027$ | $0.3276 \pm 0.0098$ |

Table 5: Average and standard deviation of class-specific performance metrics when probing for $C_{\text{B}}$ using 3x3 probes.

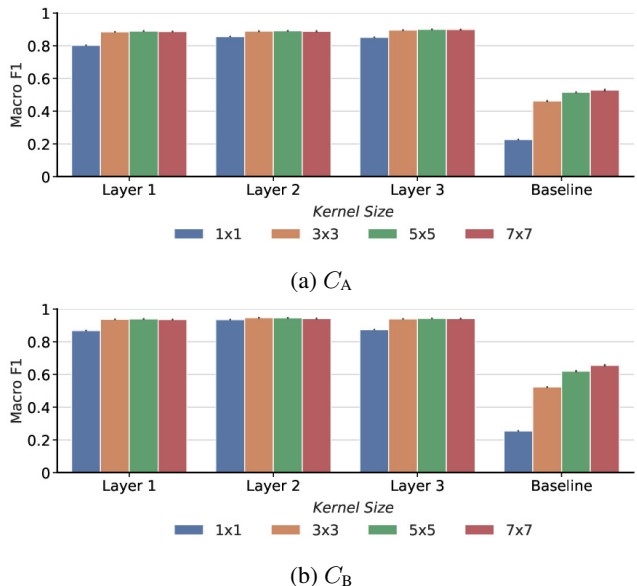

(a) $C_\text{A}$

(b) $C_\text{B}$

Figure 40: Comparisons of macro F1 achieved when probing the cell state at each layer of the agent for 'Agent Approach Direction' ($C_\text{A}$) and 'Box Push Direction' ($C_\text{B}$) using 1x1, 3x3, 5x5 and 7x7 probes. Reported F1 scores are averaged over five independent training runs. Error bars report standard deviations.

### D.3    PROBING USING LARGER PROBES

In addition to probing the agent's ConvLSTM cell states using 1x1 and 3x3 square-level concepts, we now consider using '5x5' and '7x7' probes. These are probes that are identical to the 1x1 and 3x3 probes considered previously, but that predict the concept class of a square $(x, y)$ using the activations in, respectively, 5x5 and 7x7 grids about the $(x, y)$ position of the agent's cell state. We investigate these probes in order to investigate whether the agent represents square-level concepts in a spatially-localized way at individual positions of its cell states, or whether it represents square-level concepts in a distributed way across multiple cell state positions. As previously we also consider baseline versions of these probes trained on the raw observation $x_t$.

Figure 40b shows the macro F1 when using 1x1, 3x3, 5x5 and 7x7 probes. This figure illustrates that increasing the probe size leads to minimal gains in performance when probing the agent's cell state relative to the increase in the performance of the baseline probes. We take this as evidence that the agent indeed localizes its representations of square-level concepts at individual cell state positions.

### D.4    PROBING FOR ALTERNATIVE SQUARE-LEVEL CONCEPTS

In this section, we now investigate the extent to which alternative concepts to the 'main' concepts considered in the paper – 'agent approach direction' ($C_\text{A}$) and 'box push direction' ($C_\text{B}$) – can be successfully decoded from the agent's cell state by 1x1 and 3x3 probes. Specifically, we investigate binary simplifications of the 'main' concepts, and versions of the 'main' concepts in which the on-off asymmetry is reversed.

#### D.4.1    PROBING FOR SIMPLIFIED BINARY CONCEPTS

The first set of additional concepts we probe for are binary simplifications of the concepts studied in the main paper. These binary simplifications are concepts that remove the directional components from $C_\text{A}$ and $C_\text{B}$, and just reflect whether an agent will move onto, or push a box off of, a square. We call these concepts 'Agent Approach' and 'Box Push'.

These are binary multi-class concepts that map Sokoban squares to the class {NEVER, AGAIN}. For instance, 'Agent Approach' maps a square to NEVER if the agent will never step onto that square

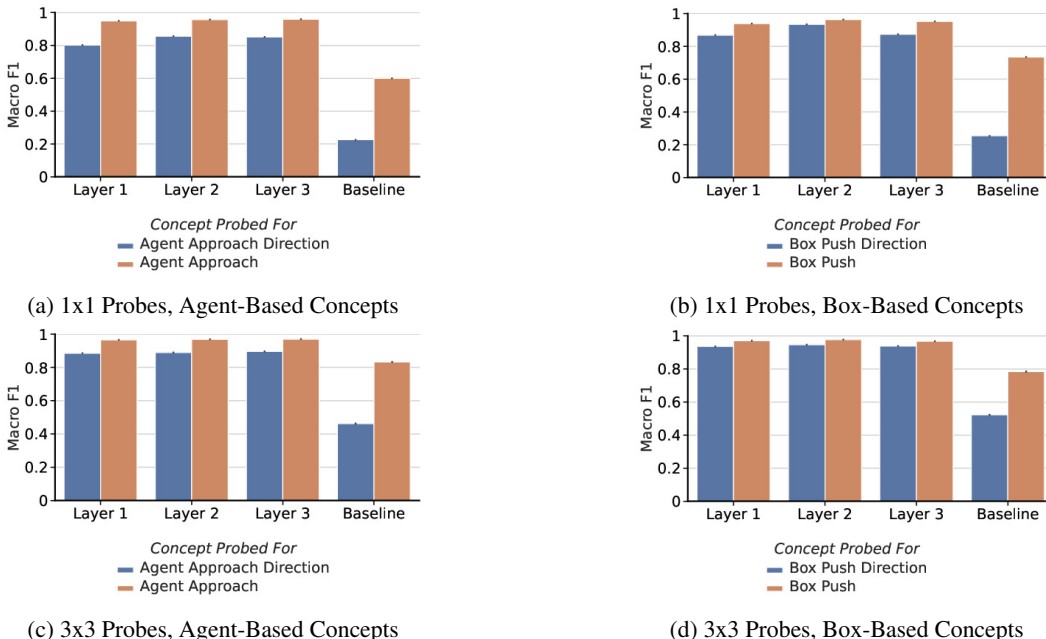

(a) 1x1 Probes, Agent-Based Concepts

(b) 1x1 Probes, Box-Based Concepts

(c) 3x3 Probes, Agent-Based Concepts

(d) 3x3 Probes, Box-Based Concepts

Figure 41: Comparisons of the macro F1 achieved when probing the cell state at each layer of the agent for the 'main' concepts ('Agent Approach Direction' and 'Box Push Direction') and the 'binary simplification' concepts ('Agent Approach' and 'Box Push'). Reported F1 scores are averaged over five independent training runs. Error bars report standard deviations.

again in the remainder of the current episode, and maps that square to AGAIN otherwise. Likewise, 'Box Push' maps a square to NEVER if the agent will never push a box off of that square again in the remainder of the current episode, and maps that square to AGAIN otherwise. Probing for these simplified concepts allows us to determine the extent to which the agent learns the directions it will move and push boxes when it visits future squares as opposed to learning simpler concepts merely reflecting which squares it will visit and which squares it will push boxes off of.

Figures 41a and 41c show the F1 scores achieved when using 1x1 probes to probe for 'Agent Approach' and 'Box Push' respectively. Figures 41b and 41d show the analogous results when using 3x3 probes. Importantly, probing performance increases only mildly relative to 'Agent Approach Direction' and 'Box Push Direction', suggesting the agent has learned the more complex directional concepts rather than these simpler alternatives. A potential explanation for the minor performance gain when probing for these simpler concepts is that the agent may sometimes know it will visit certain squares but be uncertain what action it will perform regarding them.

### D.4.2 PROBING FOR REVERSED ASYMMETRICAL CONCEPTS

The next set of additional concepts we probe for are versions of the concepts studied in the main paper where the on-off asymmetry (i.e. the asymmetry in which $C_A$ captures the direction an agent moves *on* to a square while $C_B$ captures the direction in which a box is pushed *off* of a square) is reversed. We call these concepts 'Agent Exit Direction' and 'Box Approach Direction'. Probing for these reversed asymmetrical concepts allows us to determine whether we were correct to probe for the asymmetrical concepts focused on in the main paper.

Both 'Agent Exit Direction' and 'Box Approach Direction' are multi-class concepts that map Sokoban squares to the classes {LEFT, RIGHT, UP, DOWN, NEVER}. 'Agent Exit Direction' maps squares to the direction which the agent moves the next time it moves off of them (if it ever does in the remainder of the episode). For instance, if the next time the agent moves off of a square the agent moves left, 'Agent Exit Direction' maps that square to the class LEFT. 'Box Approach Direction' maps squares to the direction the next box is pushed onto them (if a box is ever pushed onto them

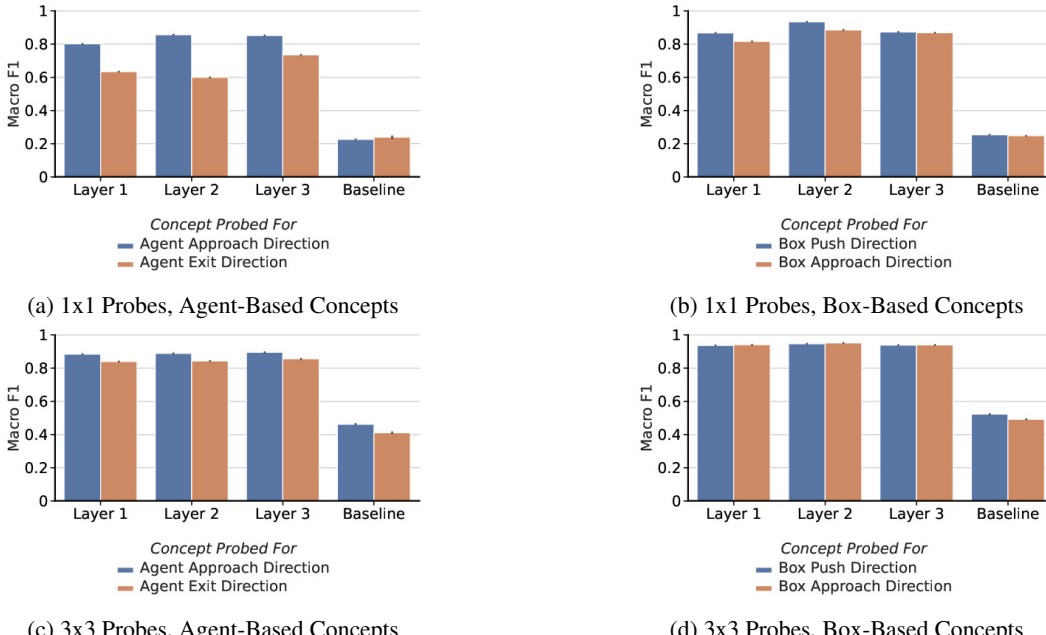

Figure 42: Comparisons of macro F1 achieved when probing the cell state at each layer of the agent for 'main' concepts ('Agent Approach Direction' and 'Box Push Direction') and the 'reversed asymmetrical' concepts ('Agent Exit Direction' and 'Box Approach Direction'). Reported F1 scores are averaged over five independent training runs. Error bars report standard deviations.

again in the remainder of the episode. For example, if the next box pushed onto a square is pushed down onto this square, 'Box Approach Direction' maps this square to DOWN.

Figures 42a and 42c compare macro F1 scores when using 1x1 probes to predict, respectively, (a) $C_A$ as opposed to 'Agent Exit Direction', and (b) 'Box Approach Direction' as opposed to $C_B$. Figures 42a and 42d show the analogous results when using 3x3 probes. The key takeaway from these figures is that there are moderate gains in probing performance when probing for 'Agent Approach Direction' rather than 'Agent Exit Direction' when using 1x1 probes, and small gains when probing for $C_B$ as opposed to 'Box Approach Direction' when using 1x1 probes. That the two concepts with the highest performance are 'Agent Approach Direction' and 'Box Push Direction' is expected given that these concepts better reflect the transition dynamics of Sokoban in which the agent pushes boxes off of squares by moving on to squares.

## D.5 PROBING FOR FUTURE ACTIONS

In the main paper, we apply the methodology introduced in Section 3.1 to show that the DRC agent we study plans in the way that we hypothesized it would in Section 3.2: by planning in terms of the square-level concepts $C_A$ and $C_B$. In this section, we now demonstrate how this methodology can be used to falsify a hypothesis regarding how we might expect an agent to plan.

Specifically, we apply the methodology to falsify the hypothesis that the agent plans by determining which actions it expects to take in a specific number of time steps. Under this hypothesis, the agent would internally represent concepts such as 'Action To Take in 1 Time Step', 'Action To Take in 2 Time Steps' and so on. These concepts would, for example, assign the class LEFT if the agent moved left in the relevant number of time steps. An agent that represented these concepts would then be able to formulate plans by forming planned action sequences of the form (LEFT, LEFT, UP, RIGHT) that the agent would be able to sequentially execute.

To apply our methodology to determine whether the agent plans in this way, we must first use linear probes to determine whether the agent linearly represents these concepts. Unlike the square-level concepts studied in the main paper, these concepts are global in the sense that they depend on the

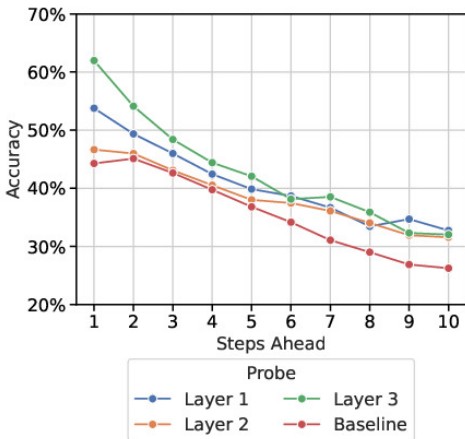

Figure 43: Accuracy achieved when probing the agent's cell state or, for the baseline, observation to predict the agent's action in a specific number of time steps.

entire observed Sokoban board. As such, we can use standard 'global' linear probes, i.e. linear probes that receive the agent's entire cell state at a layer as input.

We therefore train global probes to predict the agent's action 1,2, ..., 10 time steps into the future using the agent's cell state activations at each layer. These global probes have 10,240 parameters . These probes are trained using the same set-up as spatially-local probes as described in Appendix D.1. We also train baseline global probes that receive the agent's entire observation as input. Note that we use accuracy (not macro F1) as a measure of probe performance here as class imbalance is a lesser issue. However, for these results, accuracy is usually higher than macro F1.

Figure 43 shows the accuracies achieved by global probes trained to predict the 'Action To Take in $n$ Time Steps' for $n \in \{1, \cdots, 10\}$ into the future. While probes trained on the agent's cell state activations do outperform the baseline, they do not do so by a large margin. The performance of these probes seems especially poor relative to the performance of 1x1 probes (which have 64x fewer parameters) trained to predict $C_A$ and $C_B$. The poor performance of these probes implies that these concepts are not linearly represented. We can explain the ability of the global probes to slightly outperform the baseline by noting that global probes can infer some information regarding the agent's future actions based on the agent's internal representations of $C_A$ and $C_B$. Note, however, that global probes cannot simply read future actions off of the internal plan as it will usually be unclear which planned path the agent will follow.

Since our probes imply that the agent does not linearly represent the concepts 'Action To Take in $n$ Time Steps', we have, according to our methodology, falsified the hypothesis that the agent plans in the way suggested at the start of this section. That is, we have falsified the hypothesis that the agent plans by directly forming planned sequences of actions it expects to sequentially execute.

## E  ADDITIONAL BACKGROUND MATERIAL

In this section, we now provide additional background to the paper that may be of interest to the reader. Specifically, this section is organized as follows:

- Appendix E.1 compares our pragmatic characterization of decision-time planning to past definitions of planning.
- Appendix E.2 provides further details on the Sokoban environment.
- Appendix E.3 outlines the DRC agent architecture in greater detail.
- Appendix E.4 details the manner in which the agent we investigate was trained.
- Appendix E.5 provides preliminary behavioral evidence of planning exhibited by the agent we study.

- Appendix E.6 outlines how we operationalize the kinds of concepts we study in this paper.

- Appendix E.7 briefly details how our methodology could be applied to alternate agents in alternate environments.

## E.1 DECISION-TIME PLANNING

In Section 2.1, we provided a pragmatic characterization of *decision-time planning*. In this section, we now briefly overview definitions of decision-time planning in other fields. We do so to demonstrate that our characterization is very similar to these past definitions. Specifically, we consider approaches to decision-time planning in: (1) classical AI, (2) neuroscience, and (3) reinforcement learning.

In classic AI, or, symbolic AI, planning is viewed as the process of an agent formulating a sequence of actions to perform in order to achieve its goal (Russell & Norvig, 2010; Hendler et al., 1990). From this perspective, planning is a search problem: it involves an agent searching for a sequence of actions that can be performed to reach a goal state from the current state (Korf, 1987). This perspective of planning is very similar to our characterization in that it understands planning as requiring formulating sequences of actions and evaluating their consequences (i.e. predicting whether performing the sequence of actions will allow the agent to reach the goal state).

Decision-time planning is also a topic of interest in neuroscience (Miller et al., 2017; Jensen et al., 2024). Within neuroscience, Mattar & Lengyel (2022) have defined planning as 'the process of selecting an action or sequence of actions in terms of the desirability of their outcomes'. This is again very similar to our characterization in that it defines planning as involving formulating sequences of actions and evaluating their consequences.

Finally, in reinforcement learning, decision-time planning is usually taken to refer to the process of an agent interacting with a world model in order to determine which actions, when performed in the current state, will yield positive consequences (Hamrick et al., 2020). As stated in Section 2.1, this definition is very similar to our characterization of planning. The only difference is that this 'model-based' definition requires an agent to interact with an explicit world model to predict and evaluate the consequences of actions. Our definition loosens this requirement, only requiring that an agent *somehow* predict and evaluate the consequences of its actions. Table 6 shows some common approaches to designing RL agents capable of performing decision-time planning. Note that, other than DRC agents as studied in this paper, these approaches all maintain at least some dependence on handcrafted artefacts in order to plan. Finally, in RL, decision-time planning is often contrasted with 'background planning' which refers to the process of an agent interacting with a world model during training to learn a better policy and/or value function Sutton & Barto (2018). Importantly, our characterization of planning does *not* aim to capture background planning.

| | Explicit World Model (Known) | Explicit World Model (Learned) | No Explicit World Model |
|---|---|---|---|
| **Explicit Search Algorithm (Handcrafted)** | Example: AlphaZero (Silver et al., 2018) | Example: MuZero (Schrittwieser et al., 2020) | - |
| **Explicit Search Algorithm (Partially Learned)** | Example: MCTSNet (Guez et al., 2018b) | Example: I2A (Racanière et al., 2017) | - |
| **Explicit Search Algorithm (Fully Learned)** | - | Example: Thinker (Chung et al., 2024a) | - |
| **No Explicit Search Algorithm** | - | - | *Potential* Example: DRC (Guez et al., 2019) |

Table 6: Summary of common approaches to creating RL agents capable of decision-time planning. Agents are categorized based on the extent to which they rely on handcrafted, explicit world models and search algorithms. A search algorithm is explicit if it relies on handcrafted rather than learned elements. A world model is explicit if it depends more on handcrafted elements and less on learnable components. DRC agents are listed as *potential* examples as, prior to this work, it was unclear whether they performed decision-time planning.

### E.2 SOKOBAN

This paper investigates planning in the context of Sokoban. As explained in Section 2.2, Sokoban is a deterministic, episodic environment in which an agent operating in a 8x8 gridworld seeks to navigate around walls to push four boxes onto four targets. In this section, we provide a detailed explanation of the transition and reward dynamics of Sokoban, as well as of the symbolic representations of Sokoban boards our agent observes.

Sokoban's transition dynamics are as follows. At each time step, a Sokoban agent must choose to either move up, down, left, right or not to move. When an agent moves left, right, up or down onto a square currently containing a box, that box is respectively pushed on to the square to the left, the right, below, or above. If the move an agent attempts to perform would involve pushing a box into a non-empty square - that is, a square containing either a wall or another box - neither the box nor the agent moves. The agent cannot push two adjacent boxes simultaneously. The agent can not pull boxes. An episode ends either when (a) the agent successfully pushes all boxes onto targets or (b) when an episode length exceeds a random number between 115 and 120.

Sokoban's reward structure is as follows.

- The agent receives a reward of -0.01 at each environment step.
- The agent receives a reward of +1 when it pushes a box on to a target square.
- The agent receives a reward of -1 when it pushes a box off of a square
- The agent receives a reward of +10 after pushing a box onto all four targets.

In this paper, we study a version of Sokoban that uses *symbolic* environment representations. Each square of a Sokoban board is always in one of seven states: it is either a wall, an empty square, a box on an otherwise empty square, the agent on an otherwise empty square, a box on a target, the agent on a target, or a target with nothing on it. Thus, in our symbolic representation, we represent each square of an observed Sokoban board as a seven-dimensional one-hot vector and then combine these vectors into an array to produce the agent's observation $x_t \in \mathbb{R}^{8 \times 8 \times 7}$ of the environment state at time $t$. Importantly, however, in all figures in which we show an example of a Sokoban board, we use an RGB pixel representation of Sokoban. We do this to allow the reader to more easily understand visualized levels. Figure 2 compares the pixel and symbolic representation of a Sokoban board, where each color in the symbolic representation denotes which dimension of the one-hot vector is active for each square. It should also be noted that our version of Sokoban forgoes the layer of wall squares that is sometimes appended to the edge of Sokoban boards in previous work.

### E.3 DEEP REPEATED CONVLSTM (DRC) AGENT ARCHITECTURE

The agent studied in this paper is a Deep Repeated ConvLSTM agent as introduced by Guez et al. (2019). DRC agents are a recurrent actor-critic architecture. At each time step $t$, a convolutional encoder $e$ processes the agent's observation of the current environment state, $x_t$, into an encoding $i_t \in \mathbb{R}^{H_0 \times W_0 \times G_0}$. This is then processed by the recurrent backbone of the DRC architecture, which is a stack of ConvLSTMs (Shi et al., 2015). A ConvLSTM is a modified LSTM (Hochreiter & Schmidhuber, 1997) that include a 3D hidden state and uses convolutional connections. The DRC architecture utilises a stack of $D$ ConvLSTM units with untied parameters $\theta = (\theta_1, \cdots, \theta_d)$. These ConvLSTM units performs recurrent computations and, at time $t$, have an internal state $s_t^d = (h_t^d, g_t^d)$ where $h_t^d, g_t^d \in \mathbb{R}^{H_d \times W_d \times G_d}$ are, respectively, the output and cell state of the $d$-th ConvLSTM unit. For the agent we study, the encoder and all ConvLSTM units have a hidden dimensionality of $G_d = 32, \ \ 0 \leq d \leq D$ and utilise kernels of size 3 with a single layer of zero padding appended to convolution inputs. This means that all ConvLSTM states maintain the spatial dimensions of the environment state, so that $H_d = W_d = 8$ for all $0 \leq d \leq D$.

The DRC ConvLSTM stack includes a number of enhancements relative to standard ConvLSTM architectures. These are generic modifications that aim to improve the broad capacity of the architecture as a function approximator. The most notable of these is that, rather than performing a single step of recurrent computation for each time step, the DRC architecture performs $N$ steps of recurrent computation per time step. If the current state of the stack of $D$ ConvLSTM units is $s_{t-1}$, and we denote the operation of this stack on input encoding $i_t$ as $f_\theta(i_t, s_{t-1})$, the computation performed

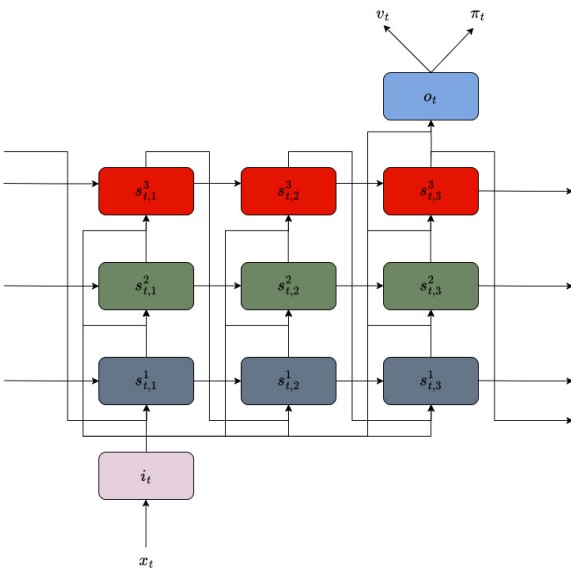

Figure 44: Illustration of DRC(3,3) architecture. For each time step, the architecture encodes the input $x_t$ as a convolutional encoding $i_t$, passes it to a stack of 3 ConvLSTMs which perform three ticks of recurrent computation and then outputs policy logits $\pi_t$ and a value estimate $v_t$.

by the ConvLSTM stack at each time step $t$ can be described by the equations below.

$$s_{t,0} = s_{t-1}, \tag{2}$$

$$s_{t,n} = f_\theta(i_t, s_{t,n-1}), \ \ 0 < n \le N, \tag{3}$$

$$s_t = s_{t,N}. \tag{4}$$

The stack of $D$ ConvLSTM units hence performs $N$ ticks of internal recurrent computation for each single time step $t$ in the environment. DRC agents are referred to as a $DRC(D, N)$ agents to make the choice of hyperparameters $D$ and $N$ explicit. The DRC architecture also includes the following additional modifications relative to a baseline ConvLSTM architecture:

- **Bottom-Up Skip Connections**: To allow information to flow up the ConvLSTM stack, the input encoding $i_t$ is provided as an input to all ConvLSTM units in the stack.

- **Top-Down Skip Connections**: To allow information to additionally flow down the ConvLSTM stack, the output of the final ConvLSTM unit on the current tick is provided as an additional input to the bottom ConvLSTM unit on the next tick.

- **Pool-and-Inject**: to allow spatial information to spread rapidly, each ConvLSTM cell additionally receives a version of its output $h_{t,n-1}^d$ on the prior tick that is spatially pooled. Specifically, this pooled output $p_{t,n-1}^d$ is produced by separately mean- and max-pooling $h_{t,n-1}^d$ spatially, passing the concatenated pooled vectors through an affine transformation, and then reshaping the result to match the dimensions of the $h_{t,n-1}^d$. This is shown below.

$$m_{t,n-1}^d = [\text{MeanPool}_{H_d,W_d}(h_{t,n-1}^d), \text{MaxPool}_{H_d,W_d}(h_{t,n-1}^d)]^T \in \mathbb{R}^{2G_d}, \tag{5}$$

$$\hat{p}_{t,n-1}^d = W_{p_d} m_{t,n-1}^d + b_{p_d} \in \mathbb{R}^{H_d W_d G_d}, \ \ W_{p_d} \in \mathbb{R}^{H_d W_d C_d \times 2G_d}, \ \ b_{p_d} \in \mathbb{R}^{W_d H_d G_d}, \tag{6}$$

$$p_{t,n-1}^d = \text{Reshape}_{H_d \times W_d \times G_d}(\hat{p}_{t,n-1}^d) \in \mathbb{R}^{H_d \times W_d \times G_d}. \tag{7}$$

Finally, the output $h_{t,N}^D$ of the final ConvLSTM cell at the final tick $N$ is concatenated with the input encoding $i_t$ and undergoes an affine transformation followed by a ReLU non-linearity to generate a vector of activations $o_t$ which is then fed to a policy head and a value head. The policy head performs an affine transformation on $o_t$ to generate a vector of action logits which parameterises a categorical

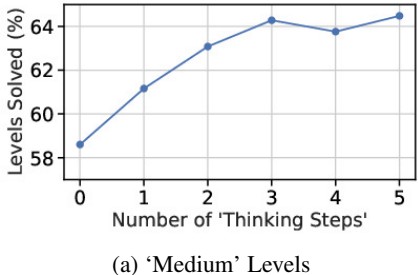
(a) 'Medium' Levels

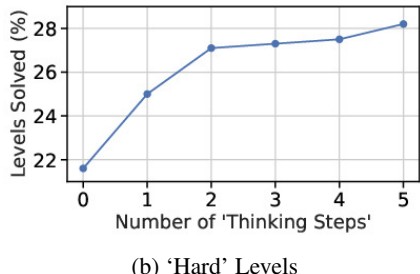
(b) 'Hard' Levels

Figure 45: The percentage of 1000 'Medium' and 'Hard' levels that the agent solves when the agent performs 'thinking steps'. Zero thinking steps corresponds to the agent's standard behavior.

distribution from which the next action can be sampled $a_t \sim \pi_t$. The value head estimates the state-value $v_t$ of the current policy for the current environment state as a linear combination of $o_t$. The policy and value heads are used as an actor and critic in order to train the agent in an actor-critic fashion. Figure 44 illustrates the computation performed a DRC(3,3) agent on a single time step.

### E.4 DRC AGENT TRAINING DETAILS

This paper focuses on analyzing a Deep Repeated ConvLSTM (DRC) agent trained to play Sokoban. The DRC agent we investigate is trained on 900k levels from the unfiltered training set of the Boxoban dataset (Guez et al., 2018a). The agent is trained in an actor-critic setting using IMPALA (Espeholt et al., 2018) for 250 million transitions.

We train the agent using a discount rate of $\gamma = 0.97$ and V-trace target of $\lambda = 0.97$. The agent is trained by additionally imposing a $\mathcal{L}^2$ penalty of size 1e-3 on the action logits, $\mathcal{L}^2$ regularisation of strength 1e-5 on the policy and value heads, and adding an entropy penalty of strength 1e-2 on the policy. Optimisation is performed using propagation through time with an unroll length of 20. We use the Adam optimiser (Kingma & Ba, 2015) with a batch size of 16 and a learning rate that decays linearly from 4e-4 to 0.

During training, the agent selects actions by sampling from a categorical distribution parameterized by its policy head logits. Once trained, the agent acts greedily by always performing the action with the greatest logit. After training, the agent solves 97.3% of unseen levels from the unfiltered test set of the Boxoban dataset (Guez et al., 2018a).

### E.5 BEHAVIORAL EVIDENCE OF PLANNING EXHIBITED BY THE DRC AGENT

As explained in Section 2.3, this paper is motivated by the phenomenon of DRC agents behaving in a way that suggests that they perform planning. For instance, DRC agents have been found in past work to solve additional Sokoban levels when forced to perform 'thinking steps', which are steps at the start of episodes where the agent is forced to remain stationary (Guez et al., 2019; Taufeeque et al., 2024). We now confirm that the agent we analyze in this paper also exhibits this planning-like behavior.

We do this by investigating amount of levels the fully-trained agent solves when given differing numbers of thinking steps. We investigate this in two datasets of Sokoban levels taken from the Boxoban dataset (Guez et al., 2018a). These are the 'Medium' and 'Hard' subsets of levels. As suggested by their names, these levels are more difficult than the 'unfiltered' subset of levels the agent was trained on. We use subsets consisting of 1000 levels taken from each dataset. Figure 45 shows the percentage of these 1000 Medium and Hard levels the agent solves when performing between zero and five thinking steps. Zero thinking steps corresponds to the agent's standard behavior. Clearly, the agent performs better when forced to perform 'thinking steps'. This represents planning-like behavior. This is because an agent capable of planning would be able to make use of the additional test-time compute afforded by 'thinking steps' to refine its plan.

### E.6 OPERATIONALIZING CONCEPTS

The concepts we investigate in this paper differ somewhat from the standard operationalization of concepts implicit in the concept-based interpretability literature. In this section, we thus explain the ways in which the concepts we study are abnormal, and provide a formal operationalization of this type of concept.

A discrete concept $C$ that takes one of $W$ values in the set $\Lambda_C = \{c_1, \cdots, c_W\}$ for every possible model input $x \in S$ is typically operationalized as a mapping $C : S \to \Lambda_C$ that maps every input $x$ to the value taken by the concept on that input, $C(x) \in \Lambda_c$ (McGrath et al., 2022). The concepts we investigate differ in the following ways:

- **Behavior Dependence**: We believe that defining concepts to be mappings from environment states to concept classes is overly restrictive in the context of reinforcement learning. This is because RL agents are situated in a continuing interaction with their environment whereby current actions influence future environment states. Hence, RL agents could plausibly learn concepts depending upon their own behavior. An example of such a concept depending on both the environment state and agent behavior in Sokoban might be 'a Sokoban board that will be solved within five moves'. Importantly for present purposes, we believe that behavior dependent concepts would be natural concepts for a planning-capable agent to learn. This is because existing model-based planners rely on explicit world models for predicting future behavior when evaluating immediate actions to perform planning. Such predictions are implicit within behavior-dependent concepts. Thus, the concepts we study depend on the agent's current parameters (since these directly determine agent behavior) and on the past observations encountered by an agent in an episode (since the DRC agent's recurrent architecture allows these to influence future behavior).

- **Spatial Localization**: We also believe that, in environments with spatially-localized dynamics like Sokoban, agents could learn similarly spatially-localized concepts. All concepts we investigate are hence features of individual squares in Sokoban boards.

We thus propose the following pragmatic operationalization of a discrete concept $C$ that takes one of $K$ values in the set $\Lambda_C = \{c_1, \cdots, c_K\}$ for a square in a Sokoban board. This definition is proposed primarily to characterise the specific concepts we study though could serve as inspiration for future definitions of concepts in RL. Let $\mathcal{S}$ denote the state space of Sokoban boards, and let $\mathcal{G} = \{(i,j)\}_{i,j=1}^8$ be a set of square indexes describing an 8x8 Sokoban board. The index $(i,j)$ refers to the square in the $i$-th row of the $j$-th column. Further, let $\Theta$ be the set of all possible parameters for a given DRC agent and let $\mathcal{H}$ be the set of all possible sequences of observed past Sokoban boards. A concept $C$ is then defined as a mapping $C : \mathcal{S} \times \mathcal{G} \times \mathcal{H} \times \Theta \to \Lambda_C$ from a particular square $(i,j) \in \mathcal{G}$ of a presently-observed Sokoban board $x_t \in \mathcal{S}$, and from an agent's current parameters $\theta \in \Theta$ and past episode observations $(x_o, x_1, \cdots, x_{t-1}) \in \mathcal{H}$ to the value $c_{x_t}^{(i,j)}$ that the concept takes on that square given agent behavior. More compactly, and suppressing dependence on past observations for notational simplicity, the concept value taken on square $(i,j)$ of Sokoban board $x_t$ when an agent has parameters $\theta$ can be written as $C_{x_t}^{(i,j)} = C(x_t, (i,j), \theta)$.

### E.7 APPLICATION OF METHODOLOGY TO OTHER MODEL-FREE ARCHITECTURES

We believe that, at a high level, the methodology introduced in Section 3.1 is general and could be applied to any model-free agent in any environment. Applying the method to a general model-free agent in a general environment would involve three steps. We illustrate these steps using the example of a model-free agent trained on Breakout:

1. **Probe For Concept Representations** In the first step, we hypothesize concepts the agent could plan with, and then probe for these concepts. For instance, the Breakout agent might plan using concepts corresponding to which bricks it plans to remove over the next 10 hits of the ball.

2. **Investigate Plan Formation** In the second step, we would inspect the manner in which the agent's concept representations develop at test-time. For instance, we might investigate whether the Breakout agent's representations of the above concepts developed in a way that

corresponded to iteratively constructing a planned hole to drill through the wall from the bottom to the top of the wall.

3. **Confirm Behavioral Dependence** In the final step, we would investigate whether we could use the vectors from the linear probes to intervene to steer the agent in the expected way. For instance, we could intervene on the Breakout agent to force it to drill a hole at a specific location of the wall.

Note, however, that the practical application of each of these steps will depend on assumptions made by the researcher in the specific experimental setting. In this paper, we made the following assumptions that informed the application of the methodology:

- **Spatial Localization** We assumed that the DRC agent we investigated represented concepts in a spatially-localized manner. This informed our choice of spatially-local probes. However, while the assumption of spatially-localized concept representations may hold in some cases (e.g. CNN-based Atari agents), it is unlikely to hold for all agents (e.g. MLP-based Mujoco agents). In cases where it doesn't hold, we would have to probe all of the agent's activations at a specific layer rather than using spatially-localized probes.

- **Linear Concept Representations** We likewise assumed that any concepts the DRC agent represented, it represented *linearly* (Mikolov et al., 2013). This informed our decision to use *linear* probes. However, this assumption may be argued to be overly-restrictive. If this were the case, we would instead have to use *non-linear probes* (i.e. probes containing non-linearities).

## F  INVESTIGATING PLANNING IN DRC AGENTS OF DIFFERENT SIZES

In the main paper, we investigate emergent planning in a DRC agent with $D = 3$ layers that performs $N = 3$ internal ticks of computation per time step. In this section, we now perform a preliminary investigation of DRC agents of different sizes and provide evidence that they too engage in planning. Specifically, we investigate whether two DRC agents of different sizes internally represent $C_A$ and $C_B$. Using the terminology from Appendix E.3 – in which we referred to a DRC agent with $D$ layers that performed $N$ ticks of computation per step as a DRC(D,N) agent – the agents we investigate are:

- A DRC(1,9) agent (Appendix F.1)
- A DRC(9,1) agent (Appendix F.2)

Both agents investigated in this section are trained for 100 million transitions using the training scheme described in Appendix E.4. Likewise, both agents exhibit behavioral evidence of planning. For instance, the DRC(1,9) and DRC(9,1) agents respectively solve additional medium difficulty levels when given five 'thinking steps' (i.e. forced stationary steps) prior to acting at the start of episodes.

We use 1x1 and 3x3 probes to investigate whether these agents internally represent $C_A$ and $C_B$. As in Section 4, we train probes with 5 different initialization seeds. All probes in this section are trained using a training scheme identical to that described in Appendix D.1 except for the fact that the training and test datasets consist of all transitions generated when running the corresponding agent for 500 and 250 episodes respectively.

### F.1  INVESTIGATING PLANNING IN A DRC(1,9) AGENT

We first investigate whether the DRC(1,9) agent exhibits evidence of internally planning. To reiterate, this agent only has a single ConvLSTM layer, but performs 9 internal ticks of computation for each environment time step. Due to only having a single layer, this agent represents an interesting case-study of a low-capacity model-free agent. After 100m transitions of training, this agent solves 84.9% of unseen levels, and solves an additional 0.8% of medium-difficulty levels when given five 'thinking steps'. While clearly less capable than the DRC(3,3) agent we focus on, the ability of the agent to solve unseen Sokoban levels, and to benefit from additional test-time compute, represents behavioral evidence of planning.

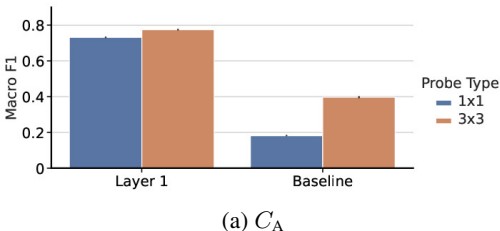
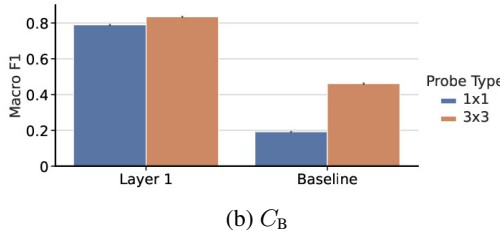

(a) $C_A$                                          (b) $C_B$

Figure 46: Macro F1s achieved by probes when predicting (a) $C_A$ and (b) $C_B$ using the **DRC(1,9)** agent's cell state, or, for the baseline probes, using the observation. Error bars show $\pm$ 1 standard deviation.

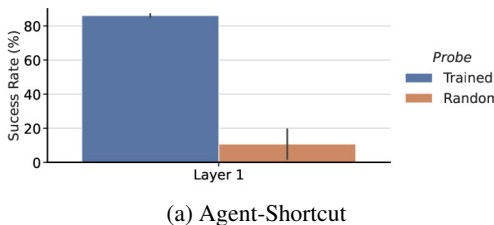
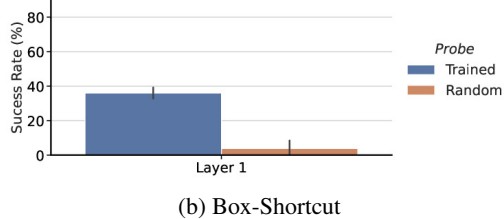

(a) Agent-Shortcut                            (b) Box-Shortcut

Figure 47: Success rates when intervening on the cell state of the **DRC(1,9)** agent in (a) Agent-Shortcut and (b) Box-Shortcut levels using trained and randomly-initialized probes. Error bars show $\pm$ 1 standard deviation.

**Probing For Planning-Relevant Concepts** Figures 46a and 46b respectively show the macro F1 scores achieved when probing this agent for $C_A$ and $C_B$. As with the DRC(3,3) agent investigated in the paper, the agent appears to represent these planning-relevant concepts. This can be seen in the strong performance of the 1x1 and 3x3 probes relative to the respective baseline. Similarly, as with the DRC(3,3) agent, the agent appears to represent these concepts in a spatially-localized manner. This can be seen in the minimal increase in performance when moving from the 1x1 probes to 3x3 probes relative to the baseline. Given that these concepts correspond to predictions of future behavior, and of the impacts of future behavior on the environment, these results suggest that the DRC(1,9) agent is planning.

**Investigating Plan Formation** Further evidence of the DRC(1,9) using these concepts to plan can be seen in Figure 48 in which we force the agent to perform five 'thinking steps' prior to acting and measure the average macro F1 when predicting $C_A$ and $C_B$ using the agent's cell state at each internal tick the agent performs during these thinking steps. As with the DRC(3,3) agent, the DRC(1,9) agent seems to iteratively refine its internal plan as formulated in terms of $C_A$ and $C_B$. Qualitative evidence of the DRC(1,9) agent internally planning can be seen in Figures 52a, 53a and 54a in which we visualize the agent's internal representations of $C_B$ over the initial 12 steps of episodes.

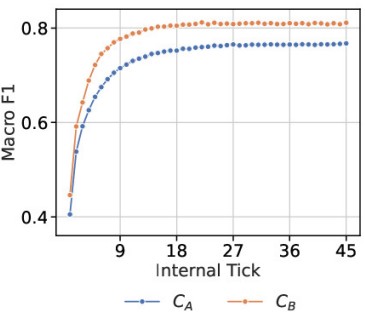

Figure 48: Macro F1 when using 1x1 probes to decode $C_A$ and $C_B$ from the **DRC (1,9)** eighth-layer agent's cell state at each of the additional 45 internal ticks performed by the DRC (1,9) agent when the agent is given 5 'thinking steps', averaged over 1000 episodes.

**Confirming Behavioral Dependence** Finally, Figures 47a and 47b show the success rates when intervening with the vectors learned by 1x1 probes to steer the behavior of the DRC(1,9) agent in Agent- and Box-Shortcut levels in the manner described in Section 6.1. Agent-Shortcut interventions are very successful. While Box-Shortcut are somewhat less

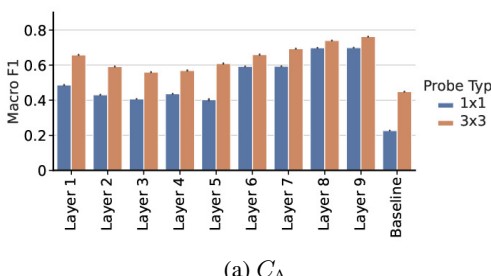
(a) $C_A$

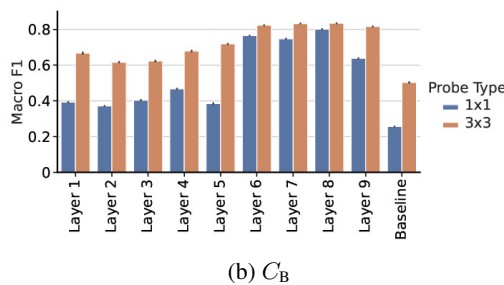
(b) $C_B$

Figure 49: Macro F1s achieved by probes when predicting (a) $C_A$ and (b) $C_B$ using the **DRC(9,1)** agent's cell state at each layer, or, for the baseline probes, using the observation. Error bars show $\pm$ 1 standard deviation.

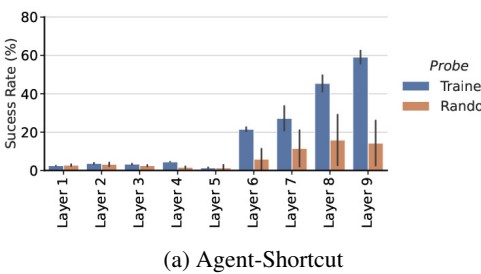
(a) Agent-Shortcut

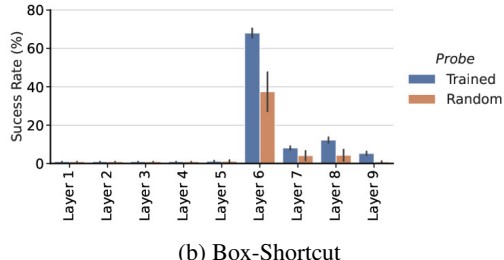
(b) Box-Shortcut

Figure 50: Success rates when intervening on the cell state of the **DRC(9,1)** agent at each layer in (a) Agent-Shortcut and (b) Box-Shortcut levels using trained and randomly-initialized probes. Error bars show $\pm$ 1 standard deviation.

successful than Agent-Shortcut interventions, they still are significantly more successful than interventions with random probe vectors.

### F.2 INVESTIGATING PLANNING IN A DRC(9,1) AGENT

We now turn attention to investigating whether the DRC(9,1) exhibits evidence of internally planning. To re-iterate, this agent has 9 ConvLSTM layers but only performs a single recurrent tick of computation per time step in the environment. The lack of additional internal ticks means this agent is an instance of a generic recurrent, model-free agent. As such, it is an interesting case-study for investigating whether generic recurrent agents can learn to internally plan. After 100m transitions of training, this agent exhibits behavioral evidence of planning as it solves 94.2% of unseen levels, and solves an additional 5.3% of medium-difficulty levels when given five 'thinking steps'.

**Probing For Planning-Relevant Concepts** Figures 49a and 49b respectively show the macro F1 scores achieved when probing this agent for $C_A$ and $C_B$. As with the DRC(3,3) agent investigated in the paper, the agent appears to represent these planning-relevant concepts, and appears to do so in a spatially-localized manner. However, the agent only does so at a few layers. Namely, the agent only appears to robustly represent $C_A$ at layers 8 and 9, and to robustly represent $C_B$ at layers 6, 7 and 8. Evidence of this can be seen in the fact that these are the layers at which 1x1 probes strongly outperform the

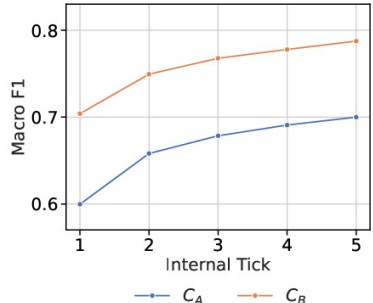

Figure 51: Macro F1 when using 1x1 probes to decode $C_A$ and $C_B$ from the **DRC (9,1)** agent's eighth-layer cell state at each of the additional 5 internal ticks performed by the DRC (9,1) agent when the agent is given 5 'thinking steps', averaged over 1000 episodes.

baseline, and at which there are only minimal gains in macro F1 when moving from 1x1 probes to 3x3 probes. Given that these concepts correspond to predictions of future actions and their impact on the environment, the fact that the agent represents these concepts provides evidence that it is planning.

**Investigating Plan Formation** Further evidence of the DRC(9,1) agent planning can be seen in Figure 51. In Figure 51, we force the agent to perform five 'thinking steps' prior to acting and measure the average macro F1 when predicting $C_A$ and $C_B$ using the eighth-layer agent's cell state after each of these thinking steps. We use the agent's eighth layer as this is the layer at which probes achieve the highest macro F1. Clearly, the agent's internal plan, as formulated in terms of $C_A$ and $C_B$, becomes iteratively more accurate when the agent is provided with additional test-time compute. This would be expected if the agent was indeed engaging in iterative planning. Qualitative evidence of the DRC(9,1) agent internally planning can be seen in Figures 52b, 53b and 54b in which we visualize the agent's internal representations of $C_B$ over the initial 12 steps of episodes. Note that, as would be expected, the DRC(9,1) agent takes more environment steps to arrive at plans than the DRC(1,9) and DRC(3,3) agents.

**Confirming Behavioral Dependence** Finally, Figures 50a and 50b show the success rates when intervening with the vectors learned by 1x1 probes to steer the behavior of the DRC(1,9) agent in Agent- and Box-Shortcut levels in the manner described in Section 6.1. Note that, in the interventions detailed in Figures 50a and 50b, it was found to be necessary to scale probe vectors by a scaling factor of 4. These results indicate that the DRC(9,1) agent does use its representations of $C_A$ and $C_B$ for planning. This is because, in general, the layers at which interventions are most successful (relative to the baseline) are the layers at which probes achieve the highest macro F1 scores. Note, however, that the success of interventions cannot be fully explained by the success of the probing the respective layer. A notable example of this is the much greater success rate of Box-Shortcut interventions when intervening at layer 6 rather than layer 8, even though probes are somewhat more accurate at layer 8. We hypothesize that this is a consequence of the DRC(9,1) agent accurately representing its plans to push boxes at many layers (layers 6-8), but only causally altering these plans at a single layer (layer 6). This aligns with the observation that Agent-Shortcut interventions become more successful at later layers, as it may be that the role of these later layers is to determine which actions the agent needs to perform to execute its plans to push boxes.

# G INVESTIGATING PLANNING IN A DIFFERENT ARCHITECTURE: RESNET

As mentioned previously, two questions left open by the main paper are (1) whether an agent with a more generic architecture like a ResNet can learn to internally plan, and (2) whether we can use the methodology introduced in Section 3.1 to determine whether such an agent is planning. In this section, we now apply our methodology to a ResNet agent trained to play Sokoban, and provide preliminary evidence suggesting an affirmative answer to both of the aforementioned questions.

The results here regard a very simple ResNet agent. This agent is parameterized by a network consisting of 24 simplified residual blocks. At each residual block, the input is passed through a convolution, followed by a layer norm, a ReLU, another convolution and another layer norm. The original input is then added to the result of these operations, and is passed through a final ReLU. These residual blocks perform no down- or up-sampling, and make use of no pooling operations. For consistency with the agents studied in this paper, all residual blocks have 32 channels. After the final residual block, the activations are flattened, passed through an MLP of dimensionality 256, and then passed to policy and value heads. This agent is trained for 250 million transitions using IMPALA with the same training scheme as described in Appendix E.4.

**Probing For Planning-Relevant Concepts** Figures 55a and 55b respectively show the macro F1 scores achieved when probing the hidden state of this agent for $C_A$ and $C_B$ after the final ReLU at each layer using both 1x1 and 3x3 probes. Note that, as with the DRC agents, the ResNet agent appears to possess spatially-localized concepts. Further, note that the 1x1 probes become iteratively more accurate over layers until a point at which the reverse trend begins. Specifically, the 1x1 probes for $C_B$ improve until about layer 10, whilst the probes for $C_A$ improve for longer until about layer 16. We hypothesize that this means that the ResNet agent is internally planning using spatially-localized concepts relating to box and agent movements, and that it is doing so by first determining how to move boxes, and then, afterward, reasoning about what that means for its own movements.

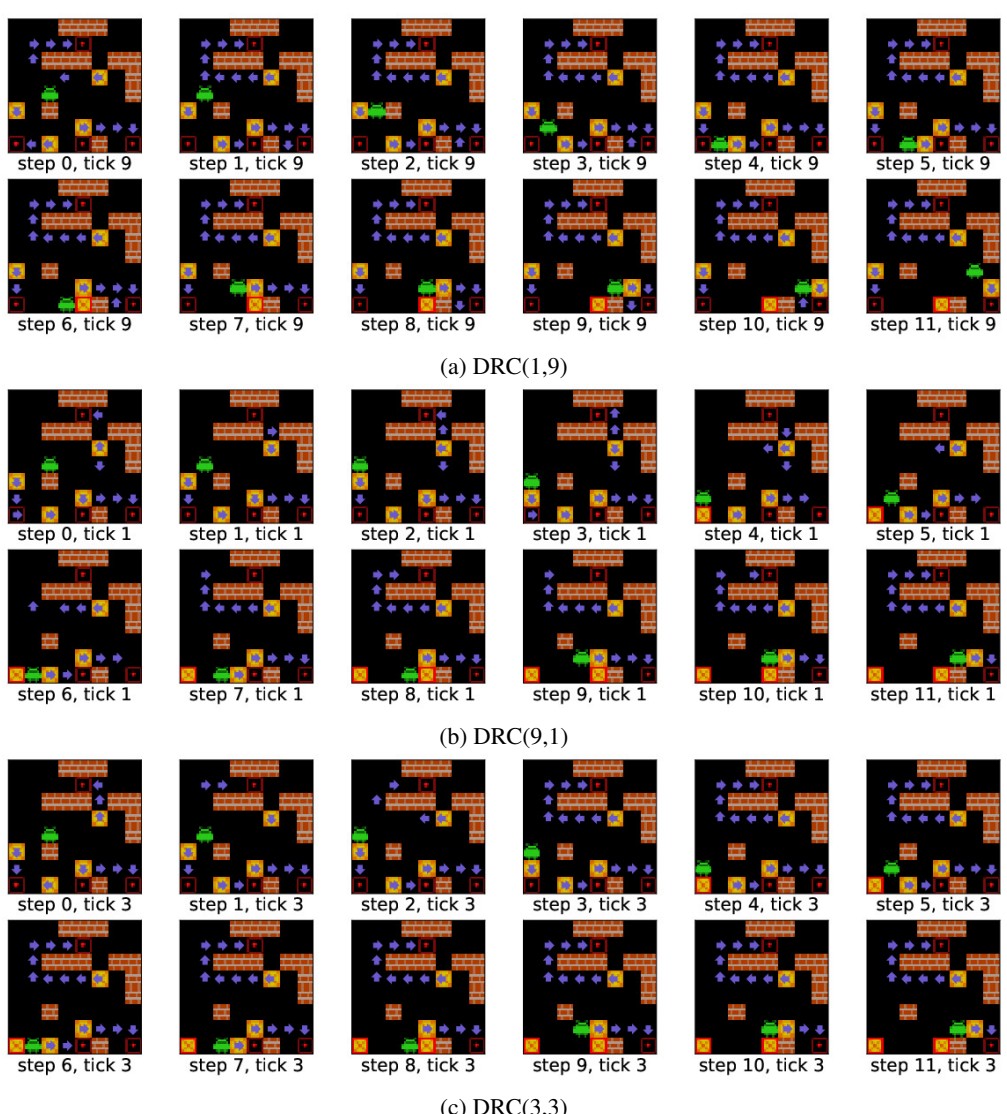

Figure 52: The internal plan of a (a) DRC(1,9), (b) DRC(9,1) and (c) DRC(3,3) agent after the final internal tick over 12 steps of the same level. Plans are decoded from the agents' (a) first, (b) eighth and (c) third layer using a 1x1 probe.

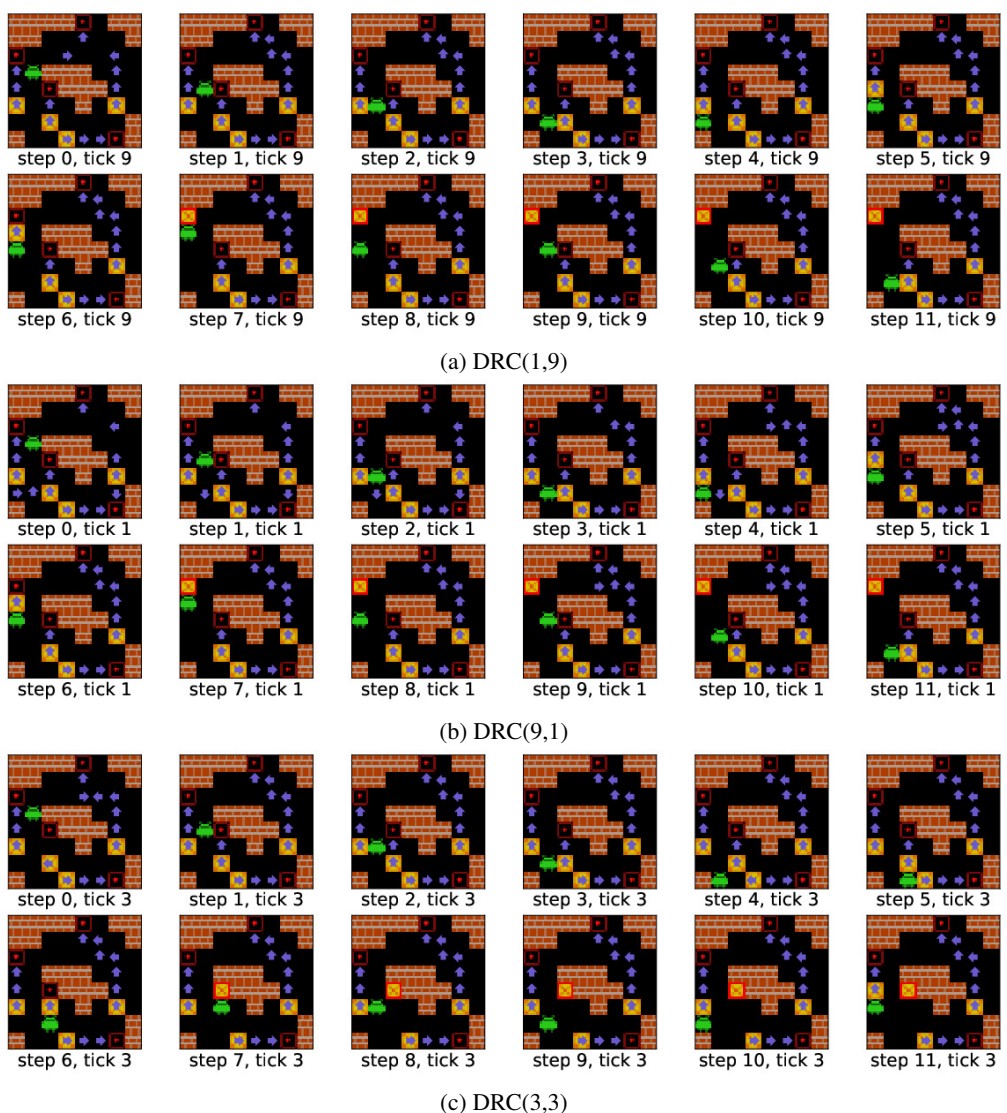

Figure 53: The internal plan of a (a) DRC(1,9), (b) DRC(9,1) and (c) DRC(3,3) agent after the final internal tick over 12 steps of the same level. Plans are decoded from the agents' (a) first, (b) eighth and (c) third layer using a 1x1 probe.

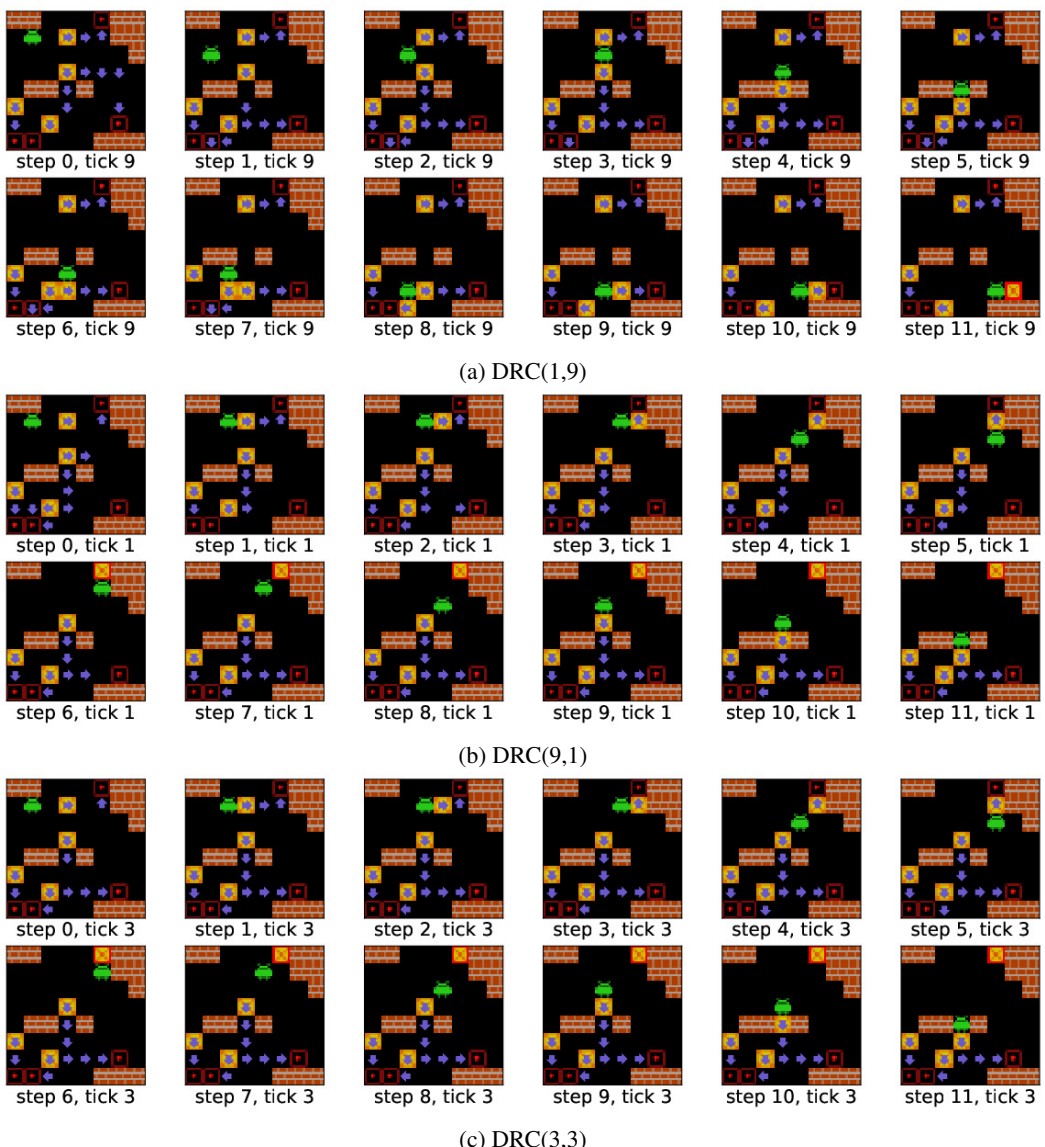

Figure 54: The internal plan of a (a) DRC(1,9), (b) DRC(9,1) and (c) DRC(3,3) agent after the final internal tick over 12 steps of the same level. Plans are decoded from the agents' (a) first, (b) eighth and (c) third layer using a 1x1 probe.

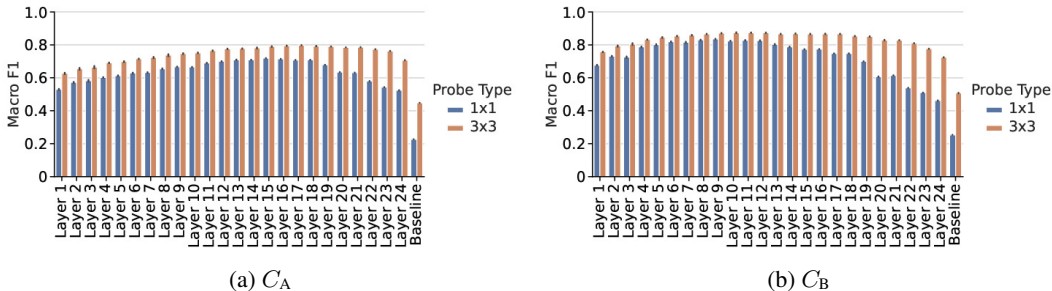

(a) $C_A$        (b) $C_B$

Figure 55: Macro F1s achieved by probes when predicting the concepts (a) $C_A$ and (b) $C_B$ using the ResNet agent's hidden state after the final ReLU of the residual block at each layer, or, for the baseline probes, using the raw observation. For each layer and probe type, we train five probes with five unique initialization seeds in the manner described in Appendix D.1. The reported error bars show $\pm$ 1 standard deviation.

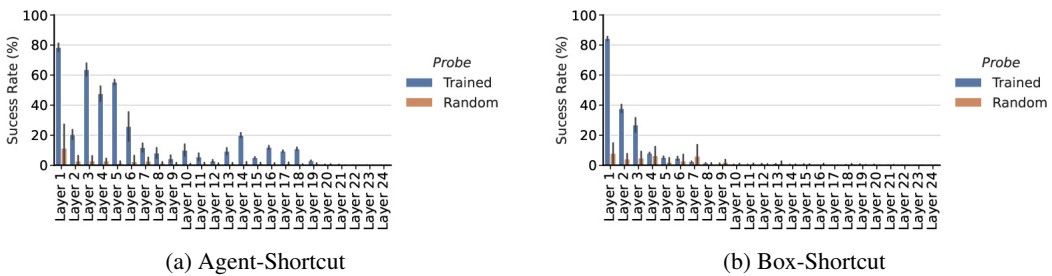

(a) Agent-Shortcut        (b) Box-Shortcut

Figure 56: Success rates when intervening on the hidden state of the ResNet agent after the residual block at each layer in (a) Agent-Shortcut and (b) Box-Shortcut levels using trained and randomly-initialized probes. Error bars show $\pm$ 1 standard deviation.

Furthermore, the fact that the probes are becoming iteratively more accurate over layers suggests that the agent has learned to perform something akin to iterative planning despite its lack of recurrent connections.

**Investigating Plan Formation** . Qualitative evidence of the agent iteratively planning across its layers can be seen in Figure 57 in which we visualize the agent's internal plan (in terms of $C_B$) over the first ten of its layers at the first time step of four episodes. Of these four episodes, the agent constructs a successful plan by its tenth layer in two episodes (Figures 57a and Figures 57b), but does not do so in the other two (Figures 57c and Figures 57d). Beyond suggesting that the ResNet agent is iteratively planning across layers, Figure 57 provides preliminary evidence that the ResNet agent is iteratively planning in a manner similar to the DRC(3,3) agent focused on in the paper.

Specifically, like the DRC agent, the ResNet agent appears to possess an internal planning mechanism with similarities to evaluative bi-directional search. For instance, evidence of plan evaluation can be seen in Figure 57b in which the agent initially (e.g. at layer 1) plans to push both left-hand boxes to the left-most target, before realizing that the upper-left box must be pushed to this target (e.g. it is the only target that the upper-left box can feasibly be pushed to) and forming a plan to instead push the lower-left box to a further target. Further evidence of the agent adapting a plan in response to apparent evaluation can be seen in Figure 57c in which the agent appears to realize that it cannot push the upper-right box up and left to the nearest target. Similarly, evidence of the agent planning forward from boxes and backwards from targets can be seen in Figure 57b in which the agent seems to plan forward from the lower-left boxes and backwards from the upper-right targets.

**Confirming Behavioral Dependence** Finally, Figures 56a and 56a show the success rates when intervening with the vectors learned by 1x1 probes to steer the behavior of the ResNet agent in Agent- and Box-Shortcut levels using the procedure described in Section 6.1. Note that, in the interventions detailed in Figures 56a and 56b, it was found to be necessary to scale probe vectors by a scaling factor of 4.

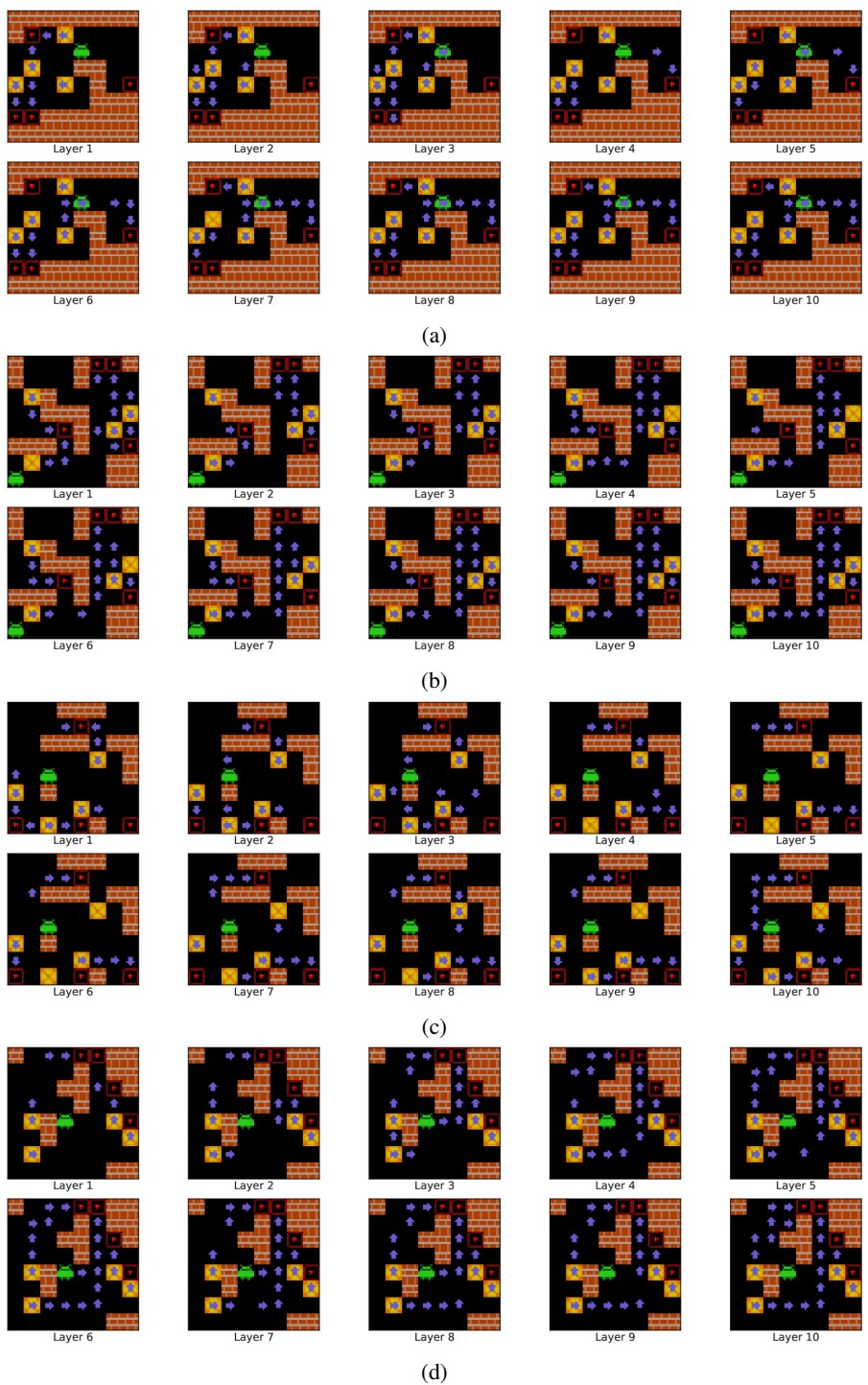

Figure 57: Examples of the ResNet agent's internal plan (in terms of its square-level representations of $C_B$ as decoded by a 1x1 probe) over its first ten layer in four levels. In two of these instances (57a and 57b), the agent arrives at a successful plan by layer ten. In the other two (57c and 57d) it does not. The visualized plans are taken from the first step of each episode.

A few things can be noted from these figures. First, interventions are very successful in both levels at the first layer. This is consistent with the hypothesis that the agent does use these concepts for planning. Interestingly, however, this is despite it being the case that the probes are more accurate when predicting these concepts at later layers. We hypothesize that this is perhaps because the agent forms its initial plans at its lowest layer and then refines these over later layers, such that the agent's plan at the lowest layer is especially amenable to being intervened upon. It is also interesting that Agent-Shortcut interventions are more successful at later layers than Box-Shortcut interventions. This is consistent with the notion that the agent primarily focuses on planning box movements at earlier layers, and then subsequently refines plans for its own actions at later layers such that it is more amenable to changing its planned movements at later layers than to changing planned box movements.

## H INVESTIGATING PLANNING IN A DIFFERENT ENVIRONMENT: MINI PACMAN

In the main paper, we use the methodology introduced in Section 3.1 to provide evidence indicating that a DRC agent trained to play Sokoban internally performs planning. However, it is natural to ask the extent to which the finding that model-free agents can learn to internally plan generalizes. This is because the 3D structure of the DRC agent's ConvLSTM cell states means the agent is particularly well-suited to learning to plan in an environment such as with a grid-based structure and localized transition dynamics. In this section, we now provide preliminary results when investigating whether a DRC agent can learn to internally plan in a different environment: Mini PacMan.

### H.1 MINI PACMAN

Mini PacMan is, like Sokoban, a grid-based environment. In Mini PacMan, an agent must navigate around walls in a grid-world and eat food. Initially, each non-wall square has food on, and levels end when the agent eats all food. However, the agent must also avoid ghosts which chase the agent. Ghosts chase the agent using A* search. In each level, there are also 'power pills'. When the agent steps onto a square with a power pill, ghosts flee, and the agent eats any ghosts it steps onto for the next 20 turns. The agent gets a reward of +1 for eating food, +2 for eating a pill, and +5 for eating a ghost. When the agent eats all food, the level is re-populated with food and new ghosts spawn in. An episode of Mini PacMan ends when the agent is either eventually eaten by a ghost, or when the agent fails to progress to the next level of an episode within 500 time steps of that level starting. Figure 58 shows an example of a Mini PacMan maze near the start of a level.

We study a version of Mini PacMan that is similar to the version studied in Hamrick et al. (2020). The version of Mini PacMan we train our agent on consists of mazes that are randomly generated each episode by (1) generating mazes using Primm's algorithm, and then (2) randomly removing each wall square with two empty adjacent non-adjacent tiles with a probability of 0.3. Each maze contains 4 pills. The number of ghosts in the initial level of each episode is equal to 1 plus an integer drawn from a Poisson(1) distribution. The number of ghosts at each subsequent level then increases by the floor of a 0.25 plus the level number times a number drawn from Unif[0, 2]. Unlike Hamrick et al. (2020) who use mazes of size 15x19, we use smaller square mazes of size 13x13. Across all levels in a single episode, the same maze is used. We use a version of Mini PacMan where the agent observes a symbolic representation $x_t \in \mathbb{R}^{13 \times 13 \times 14}$ of the environment. However, as with Sokoban, we present all visualisations using pixel representations of the Mini PacMan board.

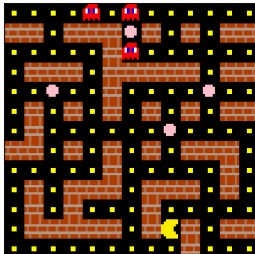

Figure 58: An example of a Mini PacMan board. The agent (the yellow pacman) must eat the food (the yellow dots) and avoid the ghosts (red) that are chasing it. When the agent eats a pill (the pink circles), the agent is able to eat ghosts over the next few steps, and ghosts change colour (blue) and flee the agent. Levels end when all food is eaten.

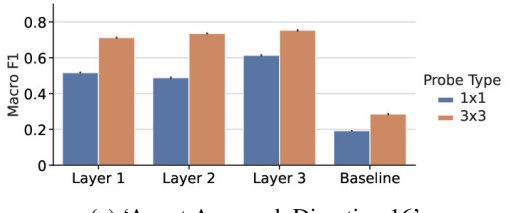 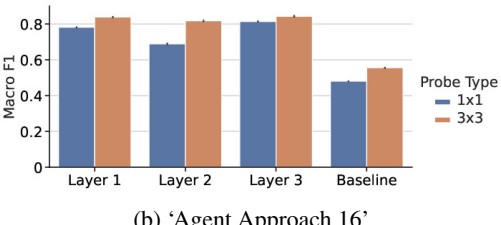

(a) 'Agent Approach Direction 16'      (b) 'Agent Approach 16'

Figure 59: Macro F1s achieved by probes when predicting (a) 'Agent Approach Direction 16' and (b) 'Agent Approach 16' using the agent's cell state at each layer, or, for the baseline probes, using the observation.

The preliminary results provided in this section regard a DRC(3,3) agent trained for 250 million transitions on this version of Mini PacMan. This agent is trained using the same training scheme as the Sokoban agents as described in Appendix E.4. As with all agents studied in this paper, this DRC agent shares the spatial dimensions of the environment it is operating in.

## H.2   PRELIMINARY PROBING RESULTS

We now present some very preliminary results regarding the aforementioned DRC agent. We initially tried probing for the concept 'Agent Approach Direction' ($C_A$) as in Sokoban but found little evidence of the agent representing it. After experimentation, however, we found probes to be able to decode the following concept from the agent's cell state:

- **Agent Approach Direction 16**: This concept tracks which squares the agent will step off of, and which direction it will do so in, over the next 16 time steps. That is, this is a variant of 'Agent Approach Direction' that only accounts for the agent's actions over the next 16 steps.

- **Agent Approach 16**: This concept tracks which squares the agent will step off of over the next 16 time steps. That is, this is a variant of 'Agent Approach Direction' that only accounts for the agent's actions over the next 16 steps, and ignores the directions that the agent enters squares from.

Figures 59a and 59a show the macro F1s achieved by 1x1 and 3x3 probes when predicting 'Agent Approach Direction 16' and 'Agent Approach 16' respectively. These probes are trained and tested on datasets consisting of 23k and 6k transitions respectively. We are unsure whether to interpret these results as indicating that either (1) the agent possesses spatially-localized representations of the concept 'Agent Approach 16', or (2) the agent posseses a representation of the concept 'Agent Approach Direction 16' distributed across adjacent positions of its cell state. This is because, for 'Agent Approach 16', 1x1 probes can accurately predict this concept, and we see minimal improvement in performance when moving from a 1x1 to 3x3 probe. In contrast, we see large improvements in performance when moving from 1x1 to 3x3 probes when predicting 'Agent Approach Direction 16'. This is consistent both with the agent representing 'Agent Approach 16' at individual positions of its cell state, and with the agent representing 'Agent Approach Direction 16' across adjacent positions of its cell state.

Figures 60 and 61 shows examples of the predictions made by a 1x1 probe trained to predict 'Agent Approach Direction 16' when applied to the agent's final-layer cell state at its final internal over the first 24 transitions at different points of 2 example episodes. Similarly, Figures 62 and 63 show the predictions made by a 1x1 probe trained to predict 'Agent Approach Direction 16' when applied to the agent's final-layer cell state at its final internal over the first 24 transitions at different points of 2 (different) example episodes. Note that the ghosts turn blue when edible, and purple on their final two turns of being edible.

These examples indicate that, as in Sokoban, the agent uses its concept representations (of whichever concept it does represent) to form an internal plan. Here, the agent's internal plan consists of the squares it plans to visit in the near-future (and, potentially, the directions it will step on to those squares from). A few observations can be made of the agent's internal plan in these examples. First,

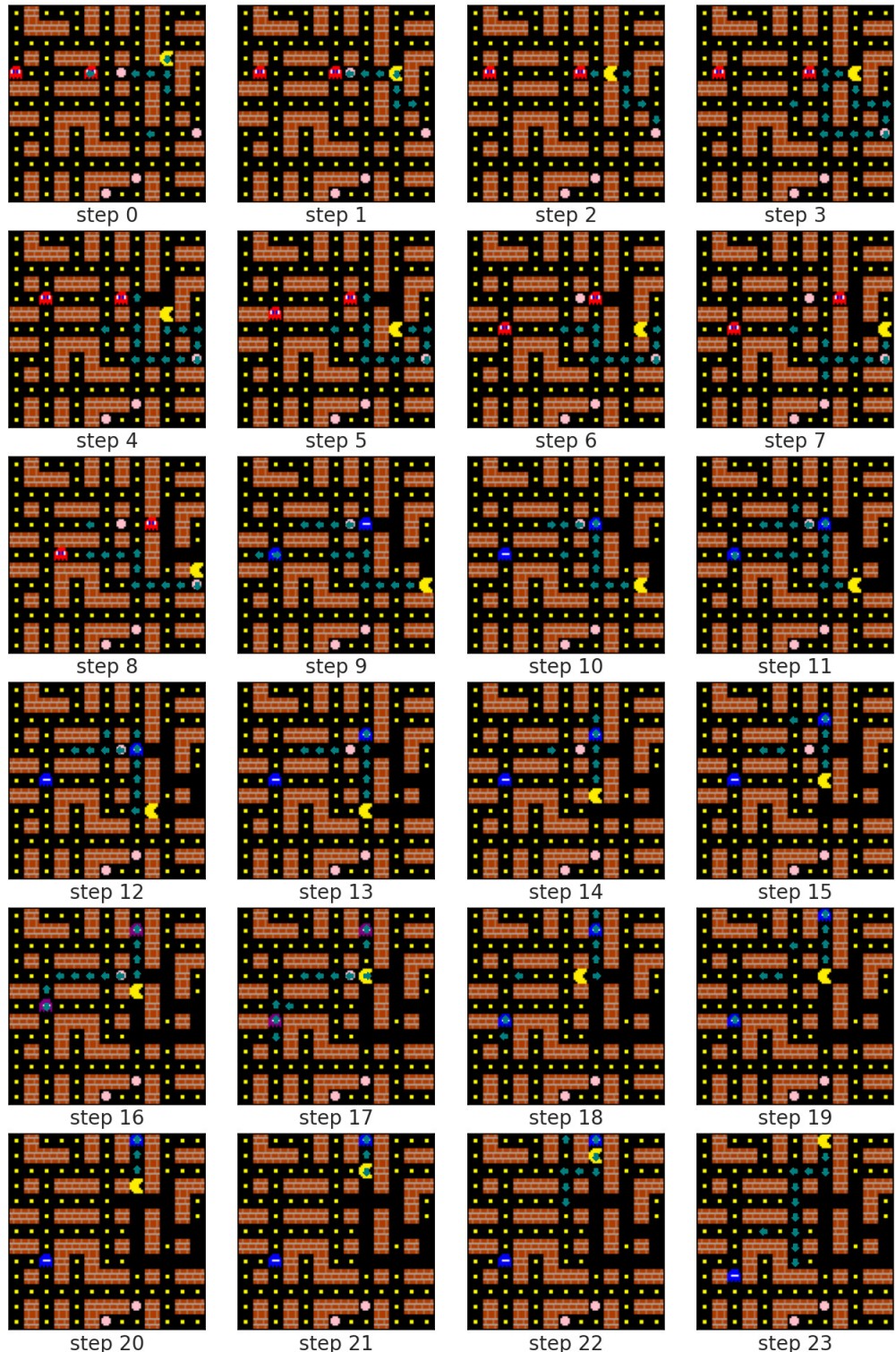

Figure 60: The DRC agent's internal plan (in terms of its square-level representations of 'Agent Approach Direction 16' as decoded from its final-layer cell state by a 3x3 probe) after its final internal tick over the first 24 steps of an episode. A teal arrow corresponds to a probe predicting that the agent expects to step onto a square from the corresponding direction over the next 16 time steps.

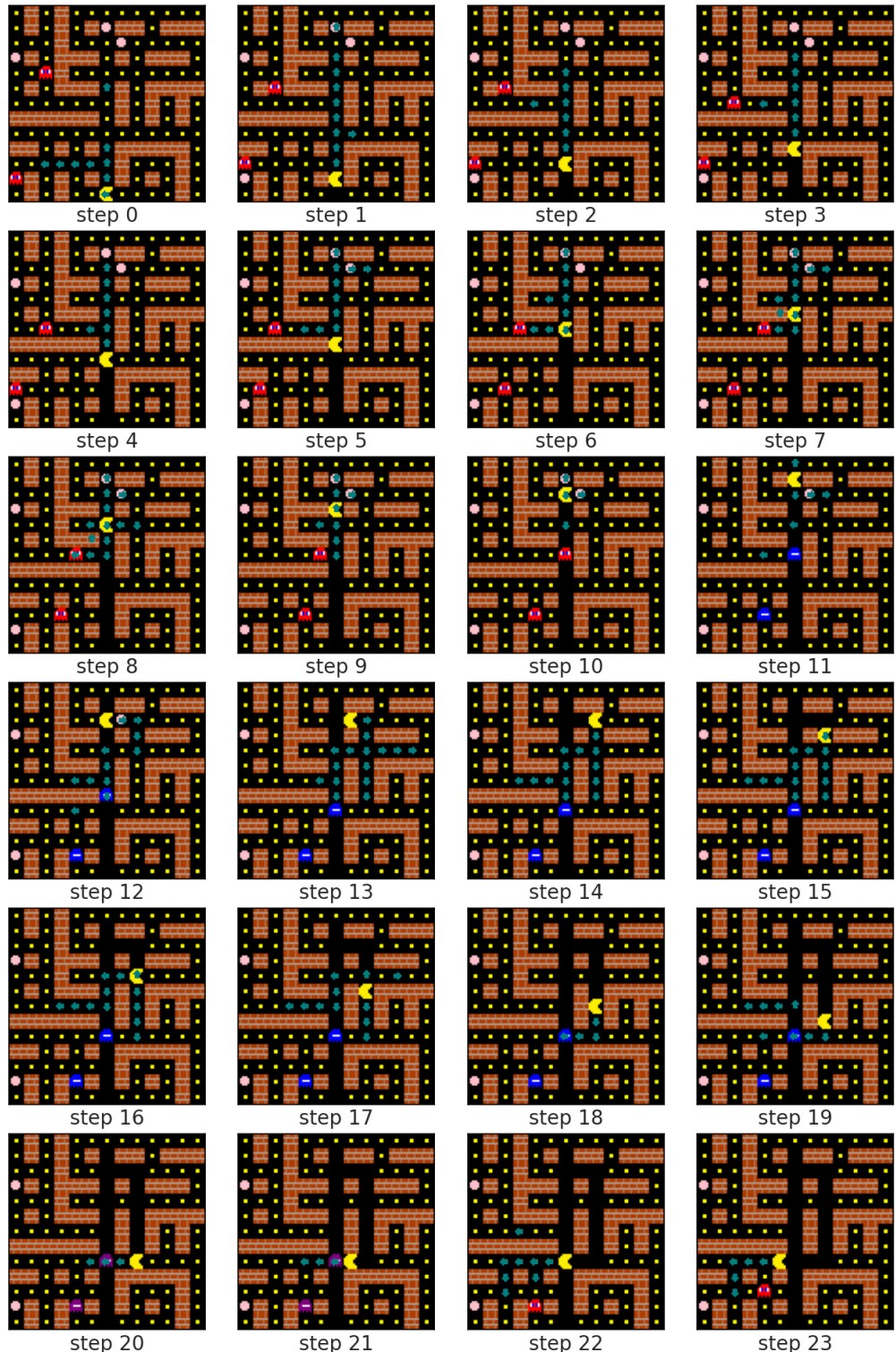

Figure 61: The DRC agent's internal plan (in terms of its square-level representations of 'Agent Approach Direction 16' as decoded from its final-layer cell state by a 3x3 probe) after its final internal tick over the first 24 steps of an episode. A teal arrow corresponds to a probe predicting that the agent expects to step onto a square from the corresponding direction over the next 16 time steps.

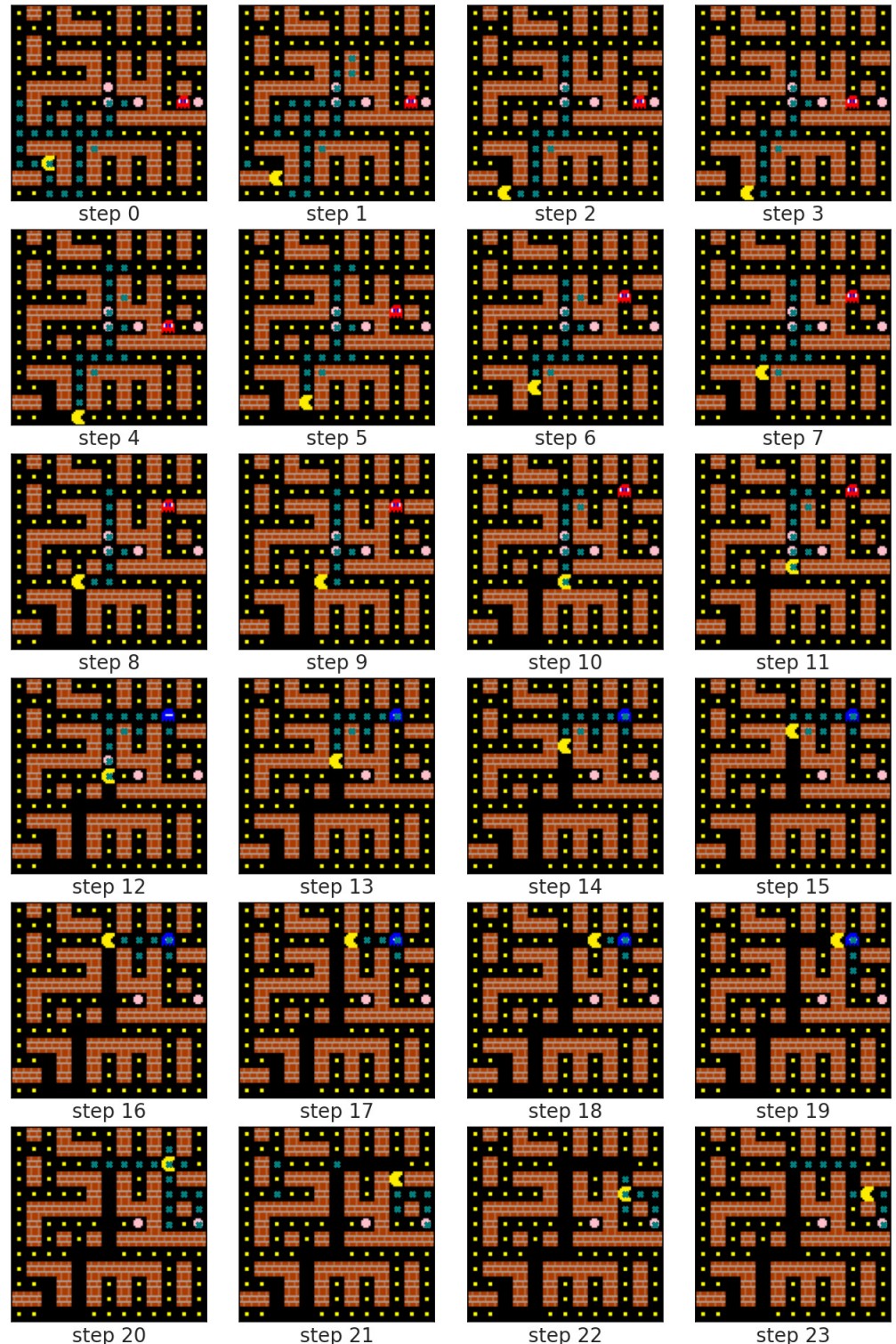

Figure 62: The DRC agent's internal plan (in terms of its square-level representations of 'Agent Approach 16' as decoded from its final-layer cell state by a 1x1 probe) after its final internal tick over the first 24 steps of an episode. A teal cross corresponds to a probe predicting that the agent expects to step onto a square over the next 16 time steps.

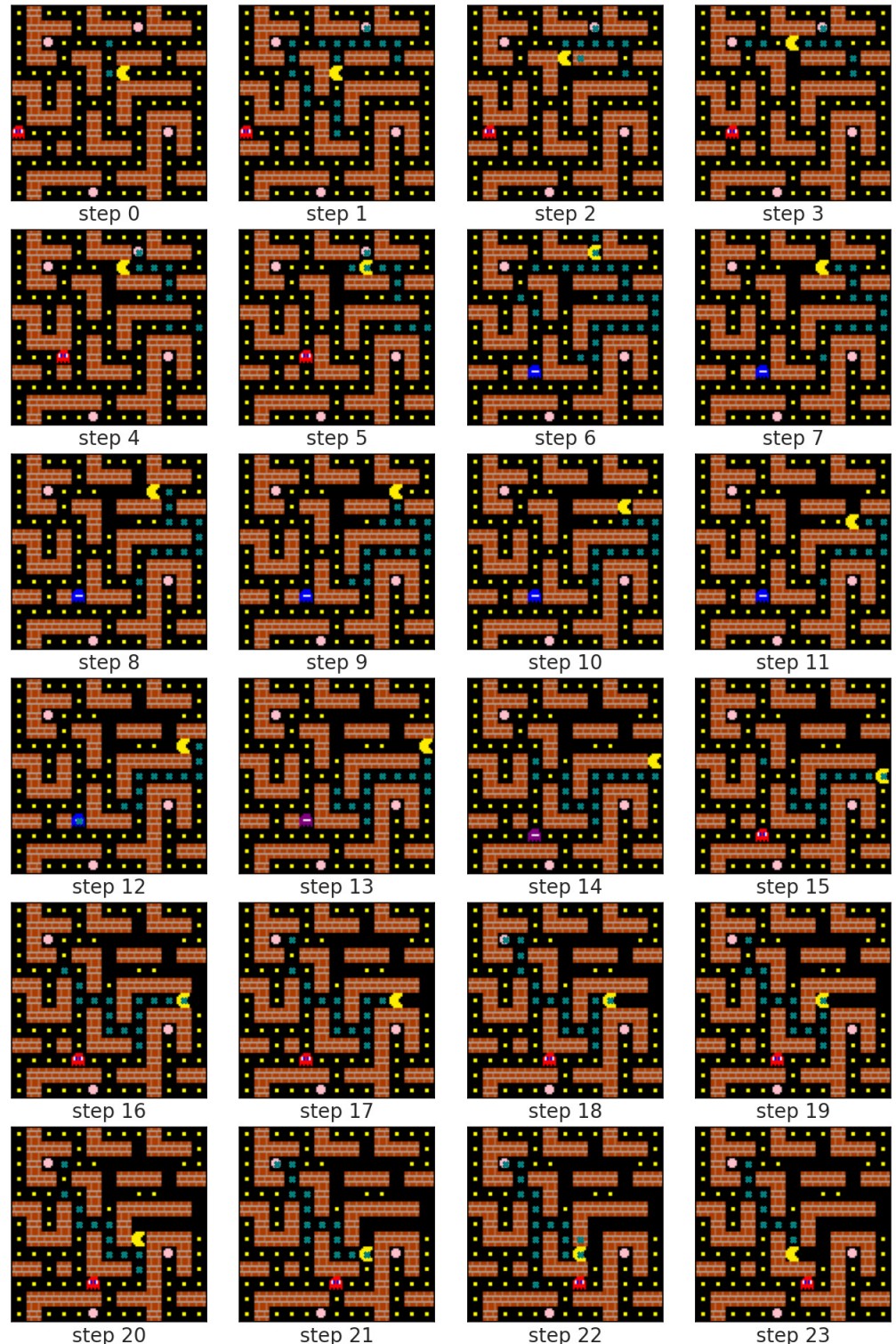

Figure 63: The DRC agent's internal plan (in terms of its square-level representations of 'Agent caption 16' as decoded from its final-layer cell state by a 1x1 probe) after its final internal tick over the first 24 steps of an episode. A teal cross corresponds to a probe predicting that the agent expects to step onto a square over the next 16 time steps.

as in Sokoban, the DRC agent's internal plans tend to corresponded to connected paths to follow. Second, again as in Sokoban, the agent's internal plans seem to iteratively develop.

However, there are also important respects in which the agent's internal planning in Mini PacMan seems different from the planning of the DRC agent in Sokoban. First, unlike in Sokoban where the DRC agent seems to plan to a fixed horizon – the end of the episode –the agent here does does not seem to have a fixed planning horizon. Rather, it seems to often plan paths towards a 'target' such as a pill or, when ghosts are edible, an edible ghost (the blue/purple sprites). We hypothesise that this may explain why the probing macro F1 scores are relatively low despite the qualitative evidence of planning, since it means the probing target is only correlate of what the agent is truly planning in terms of. Similarly, unlike in Sokoban, the agent seems to actively alter large parts of its plans, and considers multiple plans in parallel. Note that these two points imply that the concepts we probe for are mere correlates of the 'true' concepts the agent is planning in terms of.

