# OpenReview forum: "Interpreting Emergent Planning in Model-Free Reinforcement Learning"
_ICLR.cc/2025/Conference — ICLR 2025 Oral_

### Official Review · Reviewer_SDx9 · 2024-10-31

**Soundness:** 3
**Presentation:** 3
**Contribution:** 4
**Rating:** 8
**Confidence:** 4

**Summary:**

The authors conduct a series of experiments that mechanistically interpret learned neural network weights of reinforcement learning (RL) agents using deep repeated convLSTM (DRC) to determine whether they are internally planning on an implicitly learned model. The agent network is probed for specific, predetermined concepts in the Sokoban domain. The results evidence that these agents do learn spatially local concepts and reason about them in a manner resembling parallelized bi-directional search.

**Strengths:**

- This is a novel and interesting use of mechanistic interpretability.
- Provides solid empirical evidence that model-free RL agents can implicitly learn to plan under certain conditions.
- Performed experiments are highly relevant and results are well analyzed. all points made about the results are clear and it does not seem like any outstanding phenomena were overlooked.
- The discovery that this is reminiscent of bi-directional search is powerful, and may have implications on the future of model-based RL.
- visualizations of plans are easily interpretable and highly informative.

**Weaknesses:**

- Does not dive into the effect of probing different layers, even though such results are displayed.
- Only tested domain is Sokoban. analyzing second domain that is fundamentally different from grid domains is highly recommended to show that this is not a domain-specific phenomenon.
- Does not provide any background on probing, even though this is a central part of the experimentation.
- No theoretical explanation as to why planning behavior emerges.

**Questions:**

- Is it possible that algorithms other than DRC that also exhibit emergent planning will be more similar other planning algorithms (as opposed to bi-directional search)?
- Can you explain or conjecture why parallelized bi-directional search is the algorithm that emerges in this case?
- What about the size of the model and the agent's capacity to learn the world model? Is there a tradeoff between the model size and the quality of concepts that are learned?
- It seems like DRC is a perfect fit for problems like Sokoban with spatially local properties.
- Is it possible to perform many computational ticks to arrive at a final plan and then act blindly according to that plan? would this yield high accuracy? what is the horizon to which such an agent can plan, and does this also have anything to do with the model size?

---

> ### Author Response · Authors · 2024-11-22
>
> Thank you for your appreciation of our work and thoughtful feedback which has been incredibly useful. We are pleased you find our application of mechanistic interpretability to RL planning to be convincing, and to yield informative insights.
>
> **New Results and Revisions to the Submission**: Firstly, we would like to direct the reviewer’s attention towards the [global comment](https://openreview.net/forum?id=DzGe40glxs&noteId=iMHfi5Drw6) which summarises the major changes we have made to the submission, including the addition of several new results.
>
> **Probing different layers**: We have amended Appendix A.1 to include illustrations of the agent’s plan at all layers in the same example levels, and included relevant discussion of these examples.
>
> **Results on Other Domains**: To further improve the robustness of our findings, we are currently investigating a DRC agent’s planning capabilities on Mini Pacman. We expect to be able to add some results in this regard by the end of the rebuttal period. Although Mini Pacman is still grid-based, it differs from Sokoban in its non-local transition dynamics and the fact that the environment contains other (hard-coded) agents that the RL agent has to account for in its planning.
>
> **Background on probing**: We have added a sentence to the bottom of the paragraph in which we introduce linear probes to clarify how linear probes operate:
>
>  “As a linear classifier, a linear probe will compute a logit $l_k= w^T_kg$ for each class $k$ by projecting the associated activations $g \in \mathbb{R}^d$ along a class-specific vector $w_k \in \mathbb{R}^d$.”
>
> We have also added a reference to Belinkov (2022) [1] for interested readers. We hope that these changes are satisfactory to you. We are sorry that we can not add more detailed background due to the space limitations.
>
> **Lack of Theoretical Explanation** The primary goal of this paper is to empirically ascertain whether a model-free agent could indeed learn to internally plan. Theoretical explanation of this phenomenon is orthogonal to our work and would be a very exciting avenue for future work.
>
> **Other Model-Free Agents Might Implement Other Planning Algorithms Than Bidirectional Search**: It is likely that different designs of model-free RL agents might plan differently. We are currently working on interpreting a ResNet agent similar to that studied by Guez et al (2019) [2]. Given the training time of this agent, we believe it is unlikely we will be able to provide the results of applying our methodology to this agent during the rebuttal period. We apologise for this, but expect that we will have a more confident answer to your question once we have successfully interpreted it.
>
> **Why Parellized Bi-Directional Search Emerges**: We expect the reason that parallelised bidirectional planning emerges is because it allows for especially rapid plan formation at the start of an episode. This is important as it reduces the likelihood of the agent making early mistakes that make levels unsolvable. We discuss this more in a new appendix, Appendix A.2.10. Further evidence of bidirectional search being especially useful in Sokoban can be seen in the fact that one of the most capable handcrafted (i.e. not relying on deep RL) Sokoban agents uses a method that is similar to bidirectional planning [3].
>
> **Effect of model size on quality of (internal) world model** This is a very interesting question. We are currently investigating DRC agents of different sizes and hope to be able to provide results before the end of the rebuttal period.
>
> **Suitability of DRC for Sokoban** DRC agents are indeed highly suitable for Sokoban, and the prior positive evidence that DRC excels at Sokoban informed our choice of interpreting this agent.
>
> **Performing Many Ticks to Create a Plan and Then Acting Blindly** We do not have a definitive answer at hand to this question. We are fairly sure that for the DRC agents we are interpreting, this is not possible. There are at least two reasons for this:
> - The agent’s plans frequently contain transient, minor errors at individual layers. We show examples of this in Appendix A.1.
> - We believe an ‘empty’ observation would be too OOD for the agent to handle over a large number of timesteps. For example, in Appendix  A.2.6 we show instances of the agent planning based on an observation in which the agent itself is not present. While the agent’s plan can be decoded and observed to be improving over initial timesteps, if run long enough, it can sometimes result in the breakdown of agent’s representations (and plan). We think this suggests that any capacity to form plans “blind” is not robust enough to perfectly guide blind action across many episodes.
> However, we believe it may be possible to train an agent to act in such a way, given the DRC agent does seem to be capable of creating a plan upfront.
>
> We again thank you for your thoughtful comments. We welcome further discussion that would help us improve the paper.

---

> ### Author Response · Authors · 2024-11-22
>
> [1] [Belinkov (2022) Probing Classifiers: Promises, Shortcomings, and Advances](https://direct.mit.edu/coli/article/48/1/207/107571/Probing-Classifiers-Promises-Shortcomings-and)
>
> [2] [Guez et al. (2019) An Investigation of Model-Free Planning](https://arxiv.org/abs/1901.03559)
>
> [3] [Shoham & Elidan (2022) Solving Sokoban with forward-backward reinforcement learning](https://arxiv.org/abs/2105.01904)

---

> ### Author Response · Authors · 2024-11-24
> **Request for Response on Authors' Response**
>
> Respected reviewer, thank you for your initial positive review. We highly appreciate that. As promised in our initial comments, we have now included preliminary results analyzing additional agents (including a standard ConvLSTM agent) and an additional environment (Mini Pacman). This is detailed in our new top-level comment.
>
> As the discussion period will end in 2 days, we would greatly appreciate if you could review our response and let us know if you have any further questions.

---

> > ### Comment · Reviewer_SDx9 · 2024-11-26
> >
> > This reviewer appreciate the authors' very detailed response and is impressed by the new results provided.
> >
> > A suggestion regarding the visualization of probing at different layers: show the same transition for different layers side-by-side. the current, separated visuals make it hard to see the difference between the layers.
> >
> > While this work is highly detailed and impressive, I hesitate to update the paper rating to full marks until the authors ground this phenomenon in some theoretical framework and/or demonstrate it in a more complex, non-grid environment.

---

> > > ### Author Response · Authors · 2024-11-29
> > > **Conveying Thanks To The Reviewer**
> > >
> > > Thank you for your kind comments. We deeply appreciate your engagement throughout the discussion period, and shall incorporate your suggestion regarding the visualisation of plans across layers.

---

### Official Review · Reviewer_LpHt · 2024-11-03

**Soundness:** 2
**Presentation:** 3
**Contribution:** 3
**Rating:** 8
**Confidence:** 4

**Summary:**

This paper provides the first mechanistic evidence (non-behavioural) that model-free reinforcement learning agents can learn to plan.  The authors do this by studying a Deep Repeated ConvLSTM (DRC) agent playing Sokoban. While previous work showed that DRC agents exhibit planning-like behaviors, this paper demonstrates that they may actually perform internal planning.

There are three main steps in the methodology: Firstly they use linear probes to probe for planning-relevant concepts in the agent's representations. They then look at how plans form within these representations and finally look at the causal relationship of this planning by intervening on the agent behaviour.

Using this methodology, they claim that the DRC agent have an internal representation of planning concepts and can form plans through an algorithm that is like a parallel bidirectional search, planning forward and backwards. It then evaluates and adapts its plans. There is further evidence from the fact that the agent develops planning capabilities that correlate with improved performance when given extra "thinking time".

All of this is done within the Sokoban environment.

**Strengths:**

The paper's main strength is in its rigorous approach to demonstrating mechanistic evidence of planning in model-free agents. The probing experiments are well-motivated and well-designed. The authors then build on this foundation through interventional experiments that demonstrate these representations causally influence the agent's behavior.

The authors also show that the emergence of these planning capabilities correlates with improved performance when given extra computation time, connecting their mechanistic findings to previously observed behavioral results. The ablation studies also validate their methodological choices and demonstrate the robustness of the findings.

**Weaknesses:**

The lack of detail in the main part of the paper on what a concept means (which is left to the appendix) means that this important point is hard to follow. If reading the appendix is necessary to understand the paper, then the particular detail should not be in the appendix. The same goes for most sections of the appendix which should not be seen as appendices, but actually necessary parts of the paper to understand it as a whole.

One significant methodological weakness is the lack of statistical rigor in the empirical evaluation. The authors run only 5 random seeds for their experiments and perform no statistical significance testing, making it difficult to assess the reliability of their results. For example, when comparing performance between different probe types or intervention strategies, it's unclear whether the observed differences are statistically meaningful. The paper would be significantly strengthened by proper statistical analysis, including hypothesis tests, confidence intervals, and effect size calculations. The error bars in figure 4 are also surprisingly small and regular which is itself surprising. However, given that there is no code provided, it is impossible to know how reliable these results are..

Another major limitation is the narrow scope of the investigation. The paper focuses exclusively on a specific architecture (DRC) in a single environment (Sokoban), which means that it's impossible to know how well these methods or results generalise. The DRC architecture is somewhat atypical, with its multiple computational ticks per timestep, and it's unclear whether more conventional architectures could learn similar planning capabilities. It does not seem all that surprising that an architecture of this type might develop an intrinsic world-model.

Additionally, while Sokoban is a well-established planning benchmark, it has a very specific structure that may make it particularly amenable to the type of planning discovered. The authors do not discuss how their findings might extend to other environments or architectures. While the causal aspects of the paper strengthen their arguments, it would be particularly interesting to see negative results where planning is not found, using the same techiques.

**Questions:**

Have you tested whether more conventional architectures (without multiple computational ticks) can learn similar planning capabilities?
Can you provide statistical significance tests for your key comparisons, particularly where differences appear small relative to standard deviations?
In cases where the agent forms "almost correct" plans, what prevents it from finding the optimal solution? Is this a systematic failure mode?
Can you provide more analysis of the correlation between concept representation and additional compute benefit?
What led you to choose the specific concepts (Agent Approach Direction and Box Push Direction) to probe for? Did you investigate other potential planning-relevant concepts?
What happens with stronger or weaker interventions in the causal aspects of the experiments?
How computationally expensive is your probing methodology? Would it be feasible to apply this analysis in real-time during training?
Can you look at how the planning concepts emerge through the training process?
Will code be made available to reproduce the results shown here?

---

> ### Author Response · Authors · 2024-11-22
>
> Thank you for your insightful comments, which we have found to be of great help in refining and improving the paper. We are pleased that you appreciate our mechanistic approach to investigating the limits of model-free training.
>
> **New Results and Revisions to the Submission**: Firstly, we would like to direct the reviewer’s attention towards the [global comment](https://openreview.net/forum?id=DzGe40glxs&noteId=iMHfi5Drw6) which summarises the major changes we have made to the submission, including the addition of several new results.
>
> **Improvements in Discussion of “Concepts”**:  We have revised the paper to try and ensure that it is clear to a reader what a concept is without reading the Appendix.
> - We have amended the first paragraph in Section 2.4 (in which the notion of a concept is introduced) to make clear that multi-class concepts are mappings from inputs/parts of inputs to classes.
> - We have added a sentence to the end of the first paragraph in Section 3.1 (in which square-level concepts are introduced) to make it clear that square-level concepts assign classes to individual squares.
>
> **Importance of The Appendix** We agree that many parts of the Appendix strongly augment the paper. However, given space constraints and the nature of the paper, we sadly are unable to include these sections of the Appendix in the main paper. We have amended the main text to make clearer references to relevant sections of the Appendix:
> - Section 5 now contains detailed references to sections in the Appendix in which we provide further examples of each type of plan formation
> - Section 6.1 now provides detailed references to the intervention experiments described in the Appendix
>
> **Further Results To Broaden Scope** In our paper, we aimed to answer the question of whether model-free RL agents can plan or not. As has been noted by reviewer E9uN, an affirmative result in a single agent-environment pair here is sufficient to achieve this.However, we agree that the applicability of our paper would be improved by applying our methodology to alternate agents and environments. To help further improve the robustness of our findings, we are currently investigating a DRC agent’s planning capabilities on Mini Pacman. We expect to be able to add some results in this regard by the end of the rebuttal period. We believe Mini Pacman to be an interesting environment to study as, unlike Sokoban, its transition dynamics are not entirely spatially-localised.
>
> We also believe that our interpretability approach can generalise to other architectures. As such, we are also looking to give some preliminary results regarding a ResNet agent trained on Sokoban. However, a relatively large ResNet is required to get good performance on Sokoban (as shown in the original DRC paper [1]) which we have found is very time-consuming to train (training is estimated to require over 10 days on an A100 GPU). Hence, it may be challenging to include the results in time for the rebuttal period, though we will try our best.
>
> **Statistical Significance of Intervention Results** The table below shows the means and p-values when doing a t-test for the difference in means (between success rates for trained and random probes) being statistically significant. All interventions using trained probes are statistically significantly more successful than the respective intervention with random probes (at a 1% significance level).
>
> Probe	| Trained (%)  Random (%) | Trained (%)  Random (%) | Trained (%)      Random (%)
> ---------|-----------------------------------|-----------------------------------|------------------------------
> Layer  	| Layer 1  	    Layer 1   | Layer 2            Layer 2       | Layer 3              Layer 3
> AS   	| 94.6  (0.0031)    33.7        | 90.1 (0.0064)     29.8  	    | 98.8 (0.0030)     27.8
> BS   	| 56.2  (0.0042)    31.5        | 72.7 (0.0068)     30.9  	    | 80.6 (<0.0001)     4.1

---

> ### Author Response · Authors · 2024-11-22
>
> **Statistical Significance of  Probing Results** The below table shows, for each layer , the average macro F1 and the p-value for the difference in means (between 1x1 and 3x3 probes) being statistically significant. All differences are different by a statistically significant margin at a 1% significance level. This is consistent with the fact that these concepts are easier to predict for a 3x3 probe than a 1x1 probe (as evidenced by the large increase in baseline performance when moving from the 1x1 to 3x3 baseline).
>
> Probe	| 1x1                     3x3        | 1x1                   3x3            | 1x1                        3x3
> ---------|----------------------------------|-----------------------------------|----------------------------------
> Layer   | Layer 1  	    Layer 1   | Layer 2                 Layer 2  | Layer 3                Layer 3
> AS   	| 0.8024 (<0.0001) 0.8847  | 0.8560 (<0.0001)   0.8889 | 0.8516 (<0.0001) 0.8957
>
> All probes trained on the agent’s cell state activations achieve macro F1 scores that are statistically significantly different from the macro F1 scores achieved by the respective baseline probe at the 1% significance level (all with p-values <0.0001). For both 1x1 and 3x3 probes, the macro F1 scores achieved at each layer are different by a statistically significant margin (at the 1% significance level) for the macro F1 scores achieved at other layers.
>
> **Small Error Bars In Figure 4** This is a consequence of 1x1 and 3x3 probes having a minimal number of parameters (160 and 1440 respectively) and being trained on a large dataset. Specifically,  the datasets consist of >100k transitions, each with 64 labelled grid squares. As such, this dataset contains >6400k labelled examples.
>
> **What explains “almost correct plans”?** In cases of almost correct plans, the agent often (1) represents a plan without the relevant mistakes at an alternate layer (though sometimes with different mistakes) or (2) fixes the mistakes at a later time step. Appendix A.1 has been augmented to now include examples of the agent’s plan at each layer in the same level that demonstrate point (1).  We hypothesise that “almost correct plans” are best viewed as intermediate steps of the agent’s internal plan formation process.
>
> **Additional Analysis of Correlation Between Concepts and Compute Benefit** We have added two new relevant sections to the Appendix. The new sections are:
> - Appendix C.3, in which we show that the emergence during training of agent’s concept representations at all layers (i.e. not just at the final layer as previously shown) is correlated with additional compute benefit.
> - Appendix C.4, in which we show that the emergence during training of the agent’s ability to iteratively refine its plan when given additional compute (i.e. the amount by which the agent’s plan becomes more correct when given 15 additional internal ticks of compute) is correlated with additional compute benefit.
>
> **Choice of Concepts** We chose these concepts as they seemed natural for planning in a grid-based environment with localised transition dynamics. We discuss alternate square-level concepts in Appendix D.4. We decided to study these concepts specifically since (1) subsequent Sokoban states differ only in agent and box locations, and (2) boxes move off of squares (captured by Box Push Direction) when the agent moves onto squares (captured by Agent Approach Direction).
>
> **Negative Results With Alternate Concepts** We have added Appendix D.5 which briefly investigates if the agent plans by directly representing the actions it plans to take in N time steps. Appendix D.5 shows that, even when using “global” linear probes that receive as input the entirety of the agent’s cell state, we cannot accurately predict the actions the agent will take in N time steps.  This is despite “global” linear probes having many more parameters (i.e. 64x more than 1x1 probes) than the probes we use to predict square-level concepts.

---

> ### Author Response · Authors · 2024-11-22
>
> **Stronger and Weaker Interventions** We have added results in Appendix B.2 and B.3. In Appendix B.2, we investigate alternate interventions in Agent-Shortcut and Box-Shortcut levels. We detail experiments in which we:
> - Scale the intervention vector by an “intervention strength”. We find that too low or too high of an intervention strength reduces success rates.
> - Intervene upon between 0 and 3 squares (as opposed to only 1 square) as part of the “directional” intervention. We find that intervening on additional squares is helpful for low intervention strengths but not for high strengths.
> - Intervene on none of the squares on the short route. We find that we can still sometimes successfully steer the agent in Box-Shortcut levels but not Agent-Shortcut levels.
> We have also added Appendix B.3 in which we perform interventions in a new set of levels in which we intervene to make the agent act optimally when it otherwise wouldn’t.
>
> **Computational Cost of Methodology** Our probes have few parameters and so are inexpensive to train.  It took ~30 minutes on an RTX3090 to train probes on the main training dataset from the paper (>100k transitions). However, we have trained 1x1 probes to a moderate degree of accuracy (i.e. macro F1 scores that are 0.02-0.1 lower than in the paper) on ~3000 transitions in less than a minute on a RTX3090.
>
> **Could We Apply Our Methodology During Training?** Yes. We could collect transitions and labelled Sokoban boards with a FIFO buffer during training. We could then continuously train probes on the FIFO buffer.
>
> **How Do Concepts Emerge?** We have added two new Appendices:
> - Appendix C.1, in which we plot the macro F1 achieved when training 1x1 probes to predict the concepts over the first 50 million transitions of training. Appendix C.1 provides evidence that these concepts emerge early in training.
> - Appendix C.2, in which we plot, for the checkpoints of the agent taken over the first 50 million transitions of training, the increase in macro F1 when probing the agent before and after the agent is given 15 extra internal ticks of computation prior to acting. We show that the agent’s ability to iteratively refine the plans it uses these concepts to form emerges early in training.
>
> **Reproducibility** We will release the code to reproduce our results in the camera-ready version. The code was not uploaded earlier because we are still conducting new experiments (e.g., those in the appendix), and the codebase is undergoing rapid changes.
>
> We again thank you for your detailed review. We would be happy to receive additional comments you have that could aid in improving the paper even more.
>
> [1] [Guez et al. (2019) An Investigation of Model-Free Planning](https://arxiv.org/abs/1901.03559)

---

> > ### Comment · Reviewer_LpHt · 2024-11-24
> > **Reply to rebuttal**
> >
> > I think that the authors for the extremely thoughtful and detailed replies. I have upgraded my score accordingly.

---

> > > ### Author Response · Authors · 2024-11-24
> > > **Thank you for revising your score**
> > >
> > > Thank you for reading through and responding positively to our rebuttal. We highly appreciate that. As promised in our initial comments, we have now included preliminary results analyzing additional agents (including a standard ConvLSTM agent) and an additional environment (Mini Pacman). This is detailed in our new [top-level comment](https://openreview.net/forum?id=DzGe40glxs&noteId=QBUbCZ4YJf).

---

### Official Review · Reviewer_E9uN · 2024-11-03

**Soundness:** 3
**Presentation:** 3
**Contribution:** 3
**Rating:** 8
**Confidence:** 3

**Summary:**

This paper investigates the internal behavior of a Deep RL algorithm, namely, DRC by Guez et al. (2019). The purpose of this work is to verify whether this model-free RL agent is capable of planning, even if it does not rely on an explicit model of the environment. This hypothesis is tested in a discrete environment called Sokoban. The paper contains three analyses: testing for important concepts in the neural network activations with linear probes, investigating how these concepts evolve during internal RNN iterations and during episode steps, and observing whether the policy can be influenced by providing a bias to these activations using the concepts above.

**Strengths:**

The paper addresses an interesting topic for the RL and planning communities. Understanding whether some form of planning-like behavior is present in model-free RL would greatly help in merging concepts that are now discussed separately and it can help in creating more capable hybrid agents.

The paper adopts an interesting technique that I did not previously encountered in the RL literature. They apply linear probes to the activations of the policy network with specific concepts that are symbolic representations of the future behavior of the policy. The definition of behavior-dependent concepts is an interesting concept that provides useful insights about the policy. The authors also confirm that forcing the internal representation related to the desired concepts can influence the policy in the expected way.

The paper is well written and it is easy to follow.

**Weaknesses:**

1. This paper studies one algorithm, DRC, in only one class of environments, Sokoban. This strongly limits the applicability of some of the insights, which cannot be immediately applied to different algorithms and environments. For example, although the idea of behavior-dependent concepts is interesting, the proposed concepts are strongly dependent on Sokoban (there is one feature for each cardinal direction and for each direction in which the box can be pushed by the agent). Nevertheless, considering that the intended purpose of the paper is to understand whether any model-free RL agent is capable of a planning-like behavior, a positive answer can be answered even by looking at a single agent and environment class.

2. The paper often tends to be inclined towards a positive answer that DRC agents can plan, and omits to discuss alternative explanations for the observed behaviors. For example, an LSTM that is trained for N iterative steps is likely to provide more accurate answers after N interative steps at test time. Moreover, advancing throughout the episodes makes the concepts gradually easier to predict. Therefore, Figure 6 apparently represents the expected behavior of the network and it is not necessarily an instance of "plan formation", as suggested. Similarly, the fact that the arrows evolve as in Figure 1 does not necessarily imply that the agent is evaluating then refining an hypothetical policy. In some parts, the paper seems to over-interpret the meaning of a small number of empirical observations. For example, the statements in lines 373-376 or 425-427 are not sharp implications as stated in the text. Question (1) is also related to this point.

**Questions:**

(1) The policy network contains come convolutional layers. Thus, in my understanding, the fact that the activations in the policy network linearly correlate with the future movements of the agent can also be explained from the fact that the CNN can learn to encode the spatial gradients of the value function with respect to actions and neighboring states. If true, this hypothesis can be an alternative explanation for the presence of the concepts in the policy network.

(2) How an intervention is performed in practice? A quick description is given in line 462, but I believe a more precise description is required.

---

> ### Author Response · Authors · 2024-11-22
>
> Thank you for your thoughtful review. We are glad that you find our paper interesting and have found your comments very helpful for improving the paper.
>
> **New Results and Revisions to the Submission**: Firstly, we would like to direct the reviewer’s attention towards the [global comment](https://openreview.net/forum?id=DzGe40glxs&noteId=iMHfi5Drw6) which summarises the major changes we have made to the submission, including the addition of several new results.
>
> **Generalizability to other algorithms and environments**: As the reviewer has noted, a single affirmative result is sufficient for answering whether model-free RL agents can plan or not. However, we agree that results in additional settings could help improve the robustness of our findings. We are currently investigating a DRC agent’s planning capabilities on Mini Pacman (a grid-based environment with non-local transition dynamics), and expect to be able to add results in this regard by the end of the rebuttal period.
>
> We also believe that our interpretability approach can generalise to other convolutional architectures. As such, we are also looking to give some preliminary results regarding ResNet architecture on Sokoban. However, a relatively large ResNet is required to get good performance on Sokoban (as shown in the original DRC paper [1]) which we have found is very time-consuming to train (training is estimated to require over 10 days on an A100 GPU). Hence, it may be challenging to include the results in time for the rebuttal period, though we will try our best.
>
> **Details on how representation for intervention is computed**: In Section 2.4, we now explain that probes learn a vector for each concept class:
>
> “As a linear classifier, a linear probe will compute a logit $l_k= w^T_kg$ for each class $k$ by projecting the associated activations $g \in \mathbb{R}^d$ along a class-specific vector $w_k \in \mathbb{R}^d$”
>
> We have reworded the first paragraph of Section 6.1 to make our interventions clearer:
>
> “Recall that a 1x1 probe projects activations along a vector $w_k \in \mathbb{R}^{32}$ to compute a logit for class $k$ of some multi-class concept $C$. We thus encourage the agent to represent square $(x,y)$ as class $k$ for concept $C$ by adding $w_k$ to position $(x,y)$ of the agent's cell state $g_{x,y}$: $g_{x,y}$ ←  $g_{x,y} + w_k$.”
>
> **Inclination Towards Positive Answer That DRC Agent Can Plan**: At a high level, we find the most plausible explanation for the phenomenon we study to be that the agent is engaging in planning. This is because alternative explanations seem less capable of simultaneously accounting for both (1) the behavioural evidence of planning presented in the original DRC paper [1], and subsequent work [2] and (2) the internal evidence of planning that we provide.
>
> To show that, consistent with the original DRC paper, the agent we study exhibits behavioural evidence of planning,  we have added Appendix E.5 in which we show the agent solves additional levels when given extra compute. Section 5 has been amended to clarify that the behavioural evidence of planning supports the conclusions we draw: “When considered alongside the agent's planning-like behaviour, the evidence in this section indicates the agent uses the concepts we study to perform search-based planning”
>
> To provide additional support for the claim that the representations we uncover are linked to an internal planning mechanism, we have also added Appendices A.2.6-A.2.9 in which we investigate how these representations relate to capabilities commonly associated with planning: adapting and generalising to OOD scenarios. For instance, Appendix A.2.6 shows how the agent appears to be capable of generalising and forming plans in terms of these representations in levels with more boxes and targets than it saw during training. We believe alternative explanations are less able to explain the agent’s ability to form plans in OOD scenarios.

---

> ### Author Response · Authors · 2024-11-22
>
> **Softening of language and Inclusion of Additional Examples**:
> > In some parts, the paper seems to over-interpret the meaning of a small number of empirical observations. For example, the statements in lines 373-376 or 425-427 are not sharp implications as stated in the text.
>
> Despite contending that the most likely hypothesis is that the agent is indeed planning, we also believe it is possible that alternative hypotheses we have not thought of could explain both the observed behaviour and the internal mechanisms we uncover. We have adjusted the language accordingly throughout the paper to better acknowledge this. A discussion of alternative hypotheses put forward by the reviewer is provided at the end of our comment.
>
> We have amended the issue you raise regarding lines 373-376 by making our language more reflective of the level of evidence we provide. We now say “Figures 1 (A)-(B) show examples in which the agent appears to…”. We have also added additional examples of the agent forming plans in a manner suggestive of search in Appendices A.2.1-A.2.5.
>
> Regarding the issue you raise about lines 425-427, our intention was to say that a failure in our intervention experiments would falsify the hypothesis that the agent planned using these concepts. We have also, in Appendix B.1, added additional examples of the agent forming an alternate plan in response to interventions. We have also added further intervention results to support the conclusion that the interventions influence the agent’s behaviour in a manner that is consistent with the hypothesis that the agent uses the concepts for planning:
> - In Appendix B.2, we show that our interventions are largely robust to (1) scaling the probe vectors before we add them to the - agent’s cell state and (2) increasing the number of squares intervened upon as part of the “Directional” intervention
> - In Appendix B.3 we perform intervention experiments on an alternate set of levels in which we intervene to steer the agent to act optimally when it otherwise wouldn't.
>
> **Alternative Hypothesis 1**:
>  > For example, an LSTM that is trained for N iterative steps is likely to provide more accurate answers after N interative steps at test time. Moreover, advancing throughout the episodes makes the concepts gradually easier to predict. Therefore, Figure 6 apparently represents the expected behavior of the network and it is not necessarily an instance of "plan formation", as suggested.
>
> We thank you for bringing this alternate explanation to our attention.
>
> With respect to the first point you raise, we note the recurrent network is not trained to predict the consequences of future actions. Instead, it is trained to output a return-maximising action and a value estimate. As such, we do not believe that there is an a priori reason that it should be able to better predict the consequences of future actions when given extra time.
>
> We have revised the relevant section of our paper to address the second point you raise. Figure 6 now shows that the agent’s internal plan becomes iteratively more accurate when it is forced to perform 15 additional internal ticks of computation prior to acting (e.g. when the agent is forced to remain stationary for 5 steps prior to acting). This revised experimental setting removes the confounder that the increase in F1 might be due to the concepts becoming easier to predict as the agent acts in the environment.

---

> ### Author Response · Authors · 2024-11-22
>
> **Alternative Hypothesis 2**:
> > The policy network contains come convolutional layers. Thus, in my understanding, the fact that the activations in the policy network linearly correlate with the future movements of the agent can also be explained from the fact that the CNN can learn to encode the spatial gradients of the value function with respect to actions and neighboring states. If true, this hypothesis can be an alternative explanation for the presence of the concepts in the policy network
>
> We interpret this alternative hypothesis as suggesting that the agent learns to estimate and store either (1) the entire value function v(agent location) where the box locations are fixed, or (2) a value function v(agent location, box locations) that accounts for box locations. Under this hypothesis, the agent could estimate the current value by directly looking up its current location (and potentially the locations of boxes). The agent could then use the value of neighbouring locations to determine the optimal next action.
>
> We think this is an interesting hypothesis. However, we are unconvinced by it for the following reasons.
> - First, we do not think v(agent location) is sufficient to form plans of the type we show the agent does. This is because v(agent location) is limited in that it only enables planning up to the point of pushing the first box (as it assumes box locations are static). An agent that planned using such a value map would be unable to, at the start of episodes, form plans to push all boxes to targets as our probes show the agent does.
> - Similarly, we do not think the agent’s planning mechanism can be explained by the agent having learned v(agent location, box locations). For one, this is because the dimension of this value map is prohibitively large (with four boxes and 64 locations, the dimensionality of this value map would be of order 64^5). Additionally, the hypothesis that the agent’s plans are a consequence of it representing v(agent location, box locations) is inconsistent with evidence found in the original DRC paper [1] that the agent can generalise to levels with a different number of boxes. It is also inconsistent with a new appendix, Appendix A.2.7, in which we show that the agent can generalise to form plans in levels with different numbers of boxes and targets than seen during training.
>
> That said, we acknowledge the possibility that the agent partially learns these value maps and employs them as heuristics to guide the formation of plans. Indeed, we suspect that searching for evidence of partial value maps within the agent’s cell state could be a good foundation for future work that aims to reverse-engineer the planning algorithm the agent uses. However, we believe these value maps alone are insufficient to account for the full scope of the agent's planning behaviour.
>
> We would be excited to discuss this hypothesis or alternative hypotheses further.
>
> We again thank you for your insightful feedback. We would welcome further comments you have that could aid in improving the work.
>
> [1] [Guez et al. (2019) An Investigation of Model-Free Planning](https://arxiv.org/abs/1901.03559)
>
> [2] [Garriga-Alonso et al. (2024) Planning behavior in a recurrent neural network that plays Sokoban](https://openreview.net/forum?id=T9sB3S2hok&referrer=%5Bthe%20profile%20of%20Adri%C3%A0%20Garriga-Alonso%5D(%2Fprofile%3Fid%3D~Adri%C3%A0_Garriga-Alonso1))

---

> > ### Author Response · Authors · 2024-11-24
> > **Request for Response on Authors' Response**
> >
> > Respected reviewer, we have given a detailed response to your comments (and in the global response). In particular, as detailed in the new [top-level comment](https://openreview.net/forum?id=DzGe40glxs&noteId=QBUbCZ4YJf), we have added additional results analyzing different agents (including a generic ConvLSTM agent) and on an additional enviroment that could be of high interest to you.
> >
> > As the discussion period will end in 2 days, we would greatly appreciate if you could review our response and let us know if you have any further questions. If we have successfully addressed your concerns, we request that you please revise your score accordingly.

---

> > > ### Author Response · Authors · 2024-11-29
> > > **Request for Response on Authors' Response**
> > >
> > > Dear Respected Reviewer,
> > >
> > > Thank you again for your detailed and insightful review. As the extended discussion period ends in two days, we would greatly appreciate it if you could review our response and let us know if you have any further questions. If we have successfully addressed your concerns, we request that you please revise your score accordingly.

---

> > > > ### Comment · Reviewer_E9uN · 2024-12-01
> > > >
> > > > I thank you the authors for their specific comment, focused to the points I raised. I also apologize for my late reply.
> > > >
> > > > I now regard Weakness 1 as being fully addressed.
> > > >
> > > > Regarind Weakness 2, the new Figure 6 gives futher evidence to support the authors' claim. I do not believe that this excludes alternative explanations for the planning-like behaviours, but further analisys can be deferred to later studies.
> > > >
> > > > Given the above, I have increased my score.

---

> ### Author Response · Authors · 2024-12-03
> **Conveying Thanks To The Reviewer**
>
> We greatly appreciate your thoughtful comments. The points you have raised have been of great help in improving the paper.

---

### Official Review · Reviewer_CqaJ · 2024-11-04

**Soundness:** 4
**Presentation:** 4
**Contribution:** 3
**Rating:** 8
**Confidence:** 4

**Summary:**

The paper investigates whether planning -- a typically “model based” ability -- can emerge within the internal representation of “model free” agents. Specifically, the authors apply concept-based interpretability methods to identify planning-relevant concepts, whether they emerge, if they support planning, and if they can be intervened on to change the behavior of the agent. This is done for a “deep repeated ConvLSTM” (DRC) agent architecture trained on Sokoban tasks.

**Strengths:**

**[Originality]**
The investigation of emergent planning using a concept-based interpretability approach is original and interesting.

**[Quality]**
The paper is of high scientific quality, the experiments are rigorous, hypothesis driven and the conclusion (in the affirmative) is well backed-up.

**[Clarity]**
The paper is clearly written with relevant information adequately provided.

**[Significance]**
Provides a greater understanding about what is possible with model-free training alone.

**Weaknesses:**

The setting the authors investigate is ultimately restricted to a single agent architecture (DRC) trained using a single type of model-free RL algorithm (IMPALA) in a single environment type (Sokoban). The author is candid about this limitation. Nevertheless, repeating the analysis over more environments could make the general results more convincing, and investigating other architecture and/or learning rules would make the insights more generally applicable (e.g. could it be that some algorithms give rise to planning while others do not?).

Further, the paper does not address the *consequence* of planning, i.e. why is planning useful at all? For instance, [Guez 2019] investigates *data efficiency* and *generalization* as signs for planning. [Wan 2022] shows that even model-based approaches do not necessarily exhibit adaptability to local change. Thus, while it is nice to see a concept-based notion of planning, investigating whether this *leads* to things such as data efficiency, generalization and adaptability can shed greater light for the usefulness of having these representations at all.


[Guez 2019] Guez, Arthur, et al. "An investigation of model-free planning." International Conference on Machine Learning. PMLR, 2019.

[Wan 2022] Wan, Yi, et al. "Towards evaluating adaptivity of model-based reinforcement learning methods." International Conference on Machine Learning. PMLR, 2022.

**Questions:**

1. Figure 4 demonstrates the quality of the probes. However, for completeness, would it be possible to provide additional metrics such as precision, recall, class confusion matrices, etc.? The reason is that the results in this work depend heavily on the probes, and therefore it would be good to be fully transparent about the behavior of the probes and its implication on the results.

2. L447: “we [intervene] by adding the representation of NEVER to cell state position on the short path”. Could the authors describe in more details how the representation is computed for interventions (ideally as equations / pesudocodes)?
    - To my understanding, the authors train linear probes $f: \mathbb{R}^d \rightarrow \mathbb{R}^{|C|}$, with $d$ being the internal DRC representation dimension, and outputting a $|C|$-dimensional logit over the number of concepts (left right etc.). The idea here is to add a $d$-dimensional vector back into the internal DRC representation to change the agent’s behaviour, but since we are mapping from $|C|$ dimension to $d$ where $|C| << d$ (many $d$ can map onto the same prediction with high probability), how is this mapping done? Would one have to optimize for a $d$ that maximizes each class probability?
    - Also, why add instead of replace? What would happen if you replace?

3. Can the authors briefly discuss if / how the method can be applied to other model-free methods  that do not have the same architecture as DRC? Is it applicable to all architectures trained with model-free objectives?

4. Spelling: L202 behavior should be capitalized

---

> ### Author Response · Authors · 2024-11-22
>
> Thank you for your detailed and encouraging review. We are glad that you found our approach interesting and have found your comments very helpful.
>
> **New Results and Revisions to the Submission**: Firstly, we would like to direct the reviewer’s attention towards the [global comment](https://openreview.net/forum?id=DzGe40glxs&noteId=iMHfi5Drw6) which summarises the major changes we have made to the submission, including the addition of several new results.
>
> **Repeating the analysis over other environments/architectures**: We are currently investigating a DRC agent’s planning capabilities on Mini Pacman, and expect to add results in this regard by the end of the rebuttal period. We believe Mini Pacman to be an interesting environment to study as it lacks Sokoban’s spatially-localised transition dynamics.
>
> We also believe that our interpretability approach can generalise to other convolutional architectures. As such, we are also looking to give some preliminary results regarding ResNet architecture on Sokoban. However, a relatively large ResNet is required to get good performance on Sokoban (as shown in the original DRC paper [1]) which we have found is very time-consuming to train (training is estimated to require over 10 days on an A100 GPU). Hence, it may be challenging to include the results in time for the rebuttal period, though we will try our best.
>
> **Planning Helps Agent in Generalization and Adaptation**: We have added additional results in Appendices A.2.6-A.2.9 that investigate the link between the internal planning mechanism we uncover and the agent’s capacity for generalisation and adaptation. Some highlights of these results are:
> - Appendix A.2.6 shows examples of the agent forming plans in OOD levels where the agent observes a Sokoban board in which it is not itself present. These results also indicate that the learned planning algorithm within DRC is not egocentric.
> - Appendix A.2.7 shows examples of the agent planning in OOD levels with 5 boxes and targets, and with 6 boxes and targets. Guez et al (2019) [1] show that, despite being trained solely on levels with 4 boxes and targets, DRC agents can generalise to solve such levels. Our examples show that the agent’s internal planning mechanism successfully produces plans in these levels, suggesting the planning mechanism that we uncovered helps the agent generalise.
> - In Appendices A.2.8 and A.2.9, we show examples of the agent adapting its plan to changes in the environment unlike anything seen during training. Specifically, in these experiments we respectively add or remove walls in the environment during an episode, and show that the agent updates its (internal) plan in response to the changes in the environment.
>
>
> These findings shed light on the usefulness of the representations we study by suggesting a potential link between the agent’s apparent planning mechanism and the agent’s capacity for generalisation and adaptability. We have added a detailed reference to these appendices in Section 5 so that readers will be made aware of the relationship between the internal planning mechanism and the agent’s generalisation and adaptation capabilities.
>
> **Additional Metrics**: We have added class-specific precision, recall and F1 tables in Appendix D.2.
>
> **Details on how representation for intervention is computed**: When computing the logit for some class k of concept C, a 1x1 probe will project the 32-dimensional vector of cell state activations along a learned 32-dimensional weight vector $w_k$. We have added a sentence at the end of Section 2.4 to make this clear:
>
> “As a linear classifier, a linear probe will compute a logit $l_k= w^T_kg$ for each class $k$ by projecting the associated activations $g \in \mathbb{R}^d$ along a class-specific vector $w_k \in \mathbb{R}^d$.”
>
>  We have reworded the first paragraph of Section 6.1 to make our interventions clearer:
>
> “Recall that a 1x1 probe projects activations along a vector $w_k \in \mathbb{R}^{32}$ to compute a logit for class $k$ of some multi-class concept $C$. We thus encourage the agent to represent square $(x,y)$ as class $k$ for concept $C$ by adding $w_k$ to position $(x,y)$ of the agent's cell state $g_{x,y}$: $g_{x,y}$  ← $g_{x,y}$ + $w_k$”

---

> ### Author Response · Authors · 2024-11-22
>
> **Reason for adding (instead of replacing) intervention vector**: Adding vectors is standard practice when seeking to steer agent behaviour with learned vectors. An example of a paper that adds linear probe vectors to a model during a forward pass is Nanda et al (2023) [2]. The rationale for this practice is to minimise the extent to which information is overwritten.
>
>
>
> **Application to other model-free architectures**: We believe that, at a high level, our methodology is general. Applying our method to a general model-free agent would involve three steps. We illustrate these using the example of a model-free agent trained on Breakout:
>
> - In the first step, we hypothesise concepts the agent could plan with, and then probe for these concepts. For instance, the Breakout agent might plan using concepts corresponding to which bricks it plans to remove over the next 10 hits of the ball.
> - In the second step, we would inspect the manner in which the agent’s concept representations develop at test-time. For instance, we might investigate whether the Breakout agent’s representations of the above concepts developed in a way that corresponded to iteratively constructing a planned hole to drill through the wall from the bottom to the top of the wall.
> - In the final step, we would investigate whether we could use the vectors from the linear probes to intervene to steer the agent in the expected way.  For instance, we could intervene on the Breakout agent to force it to drill a hole at a specific location of the wall.
>
> However, some implementation details of our methodology are specific to our experimental setting. For instance, the assumption of spatially-localised concept representations may hold in some cases (e.g. CNN-based Atari agents) but is unlikely to hold for all agents (e.g. MLP-based Mujoco agents).  In cases where it doesn’t hold, we would have to probe all of the agent’s activations at a specific layer rather than using spatially-localised probes as in the paper.
>
> > Spelling: L202 behavior should be capitalized
>
> This has been fixed.
>
> We again thank you for your helpful comments and would welcome further comments that could help us improve our paper further.
>
> [1] [Guez et al. (2019) An Investigation of Model-Free Planning](https://arxiv.org/abs/1901.03559)
>
> [2] [Nanda et al (2023) Emergent Linear Representations in World Models of Self-Supervised Sequence Models](https://aclanthology.org/2023.blackboxnlp-1.2.pdf)

---

> ### Author Response · Authors · 2024-11-24
> **Request for Response on Authors' Response**
>
> Respected reviewer, we have given a detailed response to your comments below (and in the global response). In particular, as detailed in the new [top-level comment](https://openreview.net/forum?id=DzGe40glxs&noteId=QBUbCZ4YJf), we have added additional results analyzing different agents (including a generic ConvLSTM agent) and on an additional enviroment that could be of high interest to you.
>
> As the discussion period will end in 2 days, we would greatly appreciate if you could review our response and let us know if you have any further questions. If we have successfully addressed your concerns, we request that you please revise your score accordingly.

---

> > ### Comment · Reviewer_CqaJ · 2024-11-28
> >
> > I thank the authors for their very comprehensive response, I believe all of my questions have been addressed.
> >
> > As a final note, I deeply appreciated the author's example above regarding "Application to other model-free architectures". I think the Breakout example as well as the method's assumptions (e.g. spatially localized concepts) can be added to the appendix section and referenced to in the methods, e.g. Section 3.1. They were helpful for me to gain an appreciation for how one could apply the method elsewhere if someone is interested in building on this work.
> >
> > All in all, I think this is a very strong work with an excellent degree of scientific rigour. I have increased my score accordingly.

---

> > > ### Author Response · Authors · 2024-11-29
> > > **Conveying Thanks To The Reviewer**
> > >
> > > We deeply appreciate your kind comments, and your engagement throughout the discussion period. We shall, as you suggest, add the discussion regarding "Application to other model-free architectures"  as an Appendix.

---

### Author Response · Authors · 2024-11-22
**Global Comment 1**

We are grateful to all reviewers for the insightful reviews. We have attempted to address the specific issues each reviewer has raised in individual comments.  In this global comment, we will summarise the main changes and additions made to the paper.

**Important New Results**
We have added several new results in the appendix that we give details of later in this comment. Some key results that we would like to highlight are:

* In Appendices A.2.6 to A.2.9, we have added examples of agent planning in OOD levels. Specifically:
  * Appendix A.2.6 (Figure 18\) shows examples of the agent forming plans in OOD levels where the agent observes a Sokoban board in which it is not itself present.
  * Appendix A.2.7 (Figure 19\) shows examples of the agent planning in OOD levels with 5 boxes and targets, and with 6 boxes and targets. The agent was trained with 4 boxes and 4 targets.
  * In Appendices A.2.8 and A.2.9 (Figures 20 and 21\) we respectively add or remove walls in the environment *during* an episode, and show that the agent updates its (internal) plan in response to the changes in the environment.
* In Appendix B.2 (Figures 28-31), we provide ablations for intervention results:
  (a) varying the number of squares intervened upon.
  (b) varying the values of intervention strength parameter $\\alpha$.
  (c) performing directional intervention without performing short-route intervention.
* In Appendix B.3 (Figures 32-34), we perform interventions on a new set of levels. These new levels are constructed to test whether we can intervene to steer the agent to act optimally when it otherwise would not.
* In Appendix D.3 (Figure 40), we give results for larger probes of sizes 5x5 and 7x7. The performance differential between these much larger probes and our 1x1 probes remains small, validating the hypothesis that agent’s representations are localised.

**Main Text**

We have made the following major changes to the main text:

* \[**Updated Results/Figures**\]
  * Figure 6 now shows that the agent’s plans iteratively improve when the agent is forced to remain stationary for the first 5 steps of episodes. Previously, we showed this was the case when the agent performed actions over the first 5 steps of episodes. This removes the potential confounding effect that the improvement in F1 could be due to the concepts getting easier to predict across ticks. Figure 6 also now only shows results for the final layer for consistency with the rest of section 5\. Results for other layers have been moved to Appendix C.3.
  * Figure 7 has been split into two figures now: Figures 7 and 8 to help improve clarity and to have a style consistent with other figures.
  * Figure 9 (previously Figure 8\) has been amended to have a style consistent with other figures.
* \[**Clarity**\] We have improved our explanations regarding (1) what multi-class and square-level concepts are (Sections 2.4 and 3.2), (2) how linear probes predict classes (Section 2.4),  and (3) how we perform interventions (Section 6.1).
* \[**References To Appendices**\] We have added detailed references to new and existing appendices.

These changes have been highlighted in blue in the revised paper attached to this submission. To generate space for these changes, sentences have been reworded for greater brevity.

(continued in "Global Comment 2)

---

> ### Author Response · Authors · 2024-11-22
> **Global Comment 2**
>
> **Appendix**
>
> We have made several additions and organisational changes to the Appendix \- the length of the appendix has grown from 18 pages to 51 pages. A summary of the changes to the Appendix is as follows.
>
> * Appendix now contains a Table of Contents to help with browsing of the appendix.
> * Revisions/additions to existing appendices
>   * \[Appendix A\] Rather than containing disconnected examples, Appendix A.1 now provides examples of the agent’s plan at all layers on the same levels. Appendices A.2.1-A.2.5 now contain additional examples of types of planning.
>   * \[Appendix B\] Appendix B.1 now contains additional examples of the agent’s plan after interventions. Appendix B.2 now contains results when intervening using an intervention strength parameter, and in the absence of a “short-route” intervention.
> * New Appendices
>   * \[Appendix A\] We have added Appendices A.2.6-A.2.9 (in which we provide examples of the agent forming plans in OOD scenarios) and Appendix A.2.10 (in which we discuss links with the relevant literature). We have added Appendices A.3.1 (which explores test-time plan improvement at all layers) and A.3.2 (which provides evidence of compute being used for search).
>   * \[Appendix B\] We have added Appendix B.3 in which we intervene to steer the agent to act optimally when it otherwise wouldn’t.
>   * \[Appendix C\] A new appendix in which we provide results regarding the emergence of concept representations (C.1) and plan refinement capabilities (C.2) during training, and investigate the correlation between planning-like behaviour and concept representations (C.3) and plan refinement capabilities (C.4).
>   * \[Appendix D\] We have added Appendices in which we provide additional class-specific metrics for the probes detailed in the main paper (D.2), consider 5x5 and 7x7 probes (D.3) and apply global probes to show that the agent does not linearly represent which action it will take in specific future time steps (D.5) .
>   * \[Appendix F\] We have added Appendices in which we link our characterisation of planning to definitions of planning (F.1), and in which we show that the agent we study exhibits behavioural evidence of planning (F.5).
>
> **Next Steps**
>
> We are currently working on the following experiments
>
> * Interpreting a DRC agent trained to play Mini Pacman. We expect to have results regarding this agent ready before the end of the rebuttal period.
> * Interpreting Sokoban-playing DRC agents of different sizes. We expect to have results regarding this agent ready before the end of the rebuttal period.
> * Interpreting a Sokoban-playing ResNet agent. Training is taking a long time so we are uncertain if we will be able to provide results before the end of the rebuttal period.
>
> We commit to releasing the code to reproduce our results in the camera-ready version.

---

### Author Response · Authors · 2024-11-24
**Additional Results Regarding (1) Mini PacMan and (2) Alternate DRC Agents That Enhance the Generalisability of Our Paper**

Since posting our initial comments, we have made the following additions regarding preliminary results (that we will continue to work on in preparation for the camera-ready version of the paper) to our paper that we believe enhances the generalisability of our findings:
- **Additional DRC Agents** We have now added Appendix G (Figures 46-54), in which we provide preliminary evidence suggesting that DRC agents of different sizes also engage in planning. Specifically, we show that these alternate agents represent planning-relevant concepts, that the plans these agents form iteratively improve when given extra compute, and [UPDATE] that these plans can be intervened upon to steer the agent.
    - *Of particular interest, we provide evidence indicating that a DRC agent that performs only a single tick per step also engages in planning. As it only performs a single tick per step, this agent is a generic ConvLSTM agent*. This suggests that
        - internal planning we uncover is not merely a consequence of the special structure that DRC agent possesses, but could be a broader phenomenon.
       - At the same time, this also **shows that our proposed approach is general** and not specific to DRC.

- **Additional Environment** We have also now added Appendix H (Figures 55-57), in which we provide our preliminary results regarding interpreting a DRC agent trained in an alternate environment: Mini PacMan. We find evidence indicative of this agent representing planning-relevant concepts and using them for planning, albeit in a different way to in Sokoban.  This indicates that internal planning is not limited to environments whose transition dynamics are fully spatially-localised.

---

### Meta-Review · Area_Chair_2uWC · 2024-12-21

**Metareview:**

The paper uses concept-based interpretability to investigate if a mode-free RL algorithm plans over a set of concepts that are implicitly encoded in the learnt internal representation of the agent. The results suggest that the algorithm is performing planning in the Sokoban environment.

**Additional Comments On Reviewer Discussion:**

The authors provided detailed responses to the concerns raised by the reviewers. There was a consensus among the reviewers that the paper should be accepted.

---

### Decision · Program_Chairs · 2025-01-22

Accept (Oral)